# Cell cycle arrest enhances CD8+ T cell effector function by potentiating glucose metabolism and IL-2 signaling

Cell cycle-inhibiting chemotherapeutics are widely used in cancer treatment. Although the primary aim is to block tumor cell proliferation, their clinical efficacy also involves specific effector CD8+ T cells that undergo synchronized proliferation and differentiation. How CD8+ T cells are programmed when these processes are uncoupled, as occurs during cell cycle inhibition, is unclear. Here, we show that activated CD8+ T cells arrested in their cell cycle can still undergo effector differentiation. Cell cycle-arrested CD8+ T cells become metabolically reprogrammed into a highly energized state, enabling rapid and enhanced proliferation upon release from arrest. This metabolic imprinting is driven by increased nutrient uptake, storage and processing, leading to enhanced glycolysis in cell cycle-arrested cells. The nutrient sensible mTORC1 pathway, however, was not crucial. Instead, elevated interleukin-2 production during arrest activates STAT5 signaling, which supports expansion of the energized CD8+ T cells following arrest. Transient arrest in vivo enables superior CD8+ T cell-mediated tumor control across models of immune checkpoint blockade, adoptive cell transfer and therapeutic vaccination. Thus, transient uncoupling of CD8+ T cell differentiation from cell cycle progression programs a favorable metabolic state that supports the efficacy of effector T cell-mediated immunotherapies.

Upon encountering cognate antigens, naive CD8+ T cells initiate a tightly regulated program of clonal expansion and differentiation to generate effector T cells that are central to antiviral and antitumor immunity[1]. This process involves rapid proliferation and acquisition of cytolytic function, including perforin-mediated and granzyme-mediated killing and production of inflammatory cytokines such as interferon gamma (IFNγ) and tumor necrosis factor (TNF), which are essential for pathogen clearance and tumor control. A key early component of this response is the autocrine and paracrine production of interleukin (IL)-2, which acts as a critical driver of T cell proliferation and effector differentiation by supporting expansion of antigen-specific clones and promoting the survival of effector T cells through STAT5 signaling[2–5].

The magnitude and quality of the CD8+ T cell response dictate the formation of a competent pool of effector T cells, capable of trafficking into peripheral tissues and infiltrating the tumor microenvironment (TME), where they must overcome antigenic persistence, metabolic constrains and suppressive signals[6,7]. Recent advances have uncovered key transcriptional and metabolic programs that govern CD8+ T cell activation and differentiation. For instance, single-cell RNA sequencing has revealed dynamic transcriptional programs that guide naive CD8+ T cells through distinct differentiation states, from early activated to terminally differentiated effector cells. These transitions are regulated by key transcription factors, such as ID2, T-bet, Eomes and Blimp-1, which balance effector function and memory formation[8]. Concurrently, metabolic reprogramming, particularly the shift toward glycolysis, enhanced mitochondrial biogenesis and lipid metabolism, enables the bioenergetic demands of clonal expansion and effector differentiation[9]. Within the TME, however, persistent antigen exposure

✉ e-mail: r.arens@lumc.nl

and inhibitory cues such as signaling by programmed death-1 (PD-1) and its ligand PD-L1 drive T cell exhaustion, impairing cytotoxicity and proliferative capacity[10]. The therapeutic potential of enhancing CD8+ T cell responses has been highlighted by advances in immunotherapy. Immunotherapies including checkpoint blockade, adoptive T cell transfer and neoantigen-targeted vaccines seek to reinvigorate or amplify CD8+ T cell responses, underscoring the therapeutic importance of understanding how T cells integrate proliferative and functional cues[11].

Cell cycle-inhibiting chemotherapeutics are widely used in cancer treatment for their ability to curb tumor cell proliferation. However, emerging data suggest that the clinical efficacy of these agents often depends on the activity of effector CD8+ T cells, rather than solely on direct cytotoxic effects against tumor cells[12–14]. The fundamental question that arises from experimental and clinical settings in which cell cycle inhibitors are used is how programming of CD8+ T cells develops when proliferation and differentiation are decoupled as occurs during cell cycle arrest[7,15].

Here, we show that transient uncoupling of cell cycle progression from differentiation enables CD8+ T cells to acquire metabolic features that enhance their proliferation and effector function. These traits bolster the efficacy of various T cell-mediated immunotherapies, revealing an unappreciated layer of regulation in CD8+ T cell biology. Our findings suggest that controlled modulation of cell cycle dynamics could be leveraged to improve the design and efficacy of immunotherapeutic interventions.

## Results

### Enhanced CD8+ T cell proliferation after temporal cell cycle inhibition

To uncouple CD8+ T cell proliferation from differentiation, we developed a reductionistic assay that allowed strict control of cell cycle progression (Fig. 1a). Human and mouse CD8+ T cells were activated with CD3 and CD28 agonistic antibodies ex vivo to mimic antigenic stimulation. After activation, the CD8+ T cells were allowed to experience normal cell cycle progression (non-arrested setting), or were 'arrested' using cell cycle inhibitors that acted in distinct phases of the cell cycle; that is, hydroxyurea (HU), which arrests cells in S phase[16], the cyclin-dependent kinase 1 (CDK1) inhibitor RO-3306 (ref. 17), which inhibits G2-to-M progression, the topoisomerase I inhibitor topotecan, which arrests cells in G1 phase[18], and the CDK4/CDK6 inhibitors ribociclib and palbociclib, which prevent G1-to-S progression[19] (Fig. 1b). As expected, all inhibitors effectively arrested the cell cycle of mouse and human CD8+ T cells after activation (Fig. 1c,d and Extended Data Fig. 1a). Next, cell cycle arrest was terminated by removal of the inhibitor, thereby permitting CD8+ T cells to undergo cell cycle progression after initial blockade (hereafter named 'released' CD8+ T cells). Released CD8+ T cells displayed an increase in cell division when compared to non-arrested cells that proliferated the same time, as visualized by cell proliferation tracing dyes and by determining the percentage of dividing cells and calculating the average number of cell divisions (division index) of non-arrested and released conditions (Fig. 1c–e and Extended Data Fig. 1a). Enhanced proliferation of released CD8+ T cells was observed for all tested inhibitors, indicating that this effect was neither drug specific nor restricted to inhibition of a particular phase of the cell cycle. HU consistently provided the best overall CD8+ T cell survival compared to other cell cycle inhibitors (Extended Data Fig. 1b).

The enhanced proliferation after transient arrest suggested altered CD8+ T cell differentiation. Therefore, we examined key effector markers. ID2 expression was induced in both arrested and non-arrested cells and further increased after release (Fig. 1f). Arrested cells initially showed low granzyme B, but upregulated it strongly upon release, exceeding levels in non-arrested cells (Fig. 1f). While CD62L and CD127 were only modestly reduced, arrested cells upregulated CD69 (Fig. 1g

and Extended Data Fig. 1c–e). EOMES was induced during arrest and further increased after release, paralleling granzyme B and CXCR3 expression. PD-1 and LAG-3 were moderately upregulated during arrest, with higher expression in non-arrested and released cells (Fig. 1g and Extended Data Fig. 1c,d). Arrested cells also exhibited blast formation, although less prominently than non-arrested or released cells (Extended Data Fig. 1f–h).

We next assessed the impact of temporal cell cycle arrest on CD8+ T cell responses in vivo by vaccinating mice with the HPV16 E7$_{43–63}$ long peptide in combination with HU or topotecan treatment. Although E7-specific CD8+ T cell responses were initially lower in cell cycle inhibitor-treated mice, the peak response exceeded that of untreated vaccinated mice (Fig. 1h, Extended Data Fig. 2a and Supplementary Fig. 1). This reflected a steeper expansion of vaccine-induced CD8+ T cells following the blockade (Fig. 1h,i and Extended Data Fig. 2a), suggesting that cell cycle arrest programmed cells for rapid proliferation. The expansion effect was also evident in circulating KLRG1+ CD8+ T cells, marking antigen-reactive effector cells (Fig. 1h and Extended Data Fig. 2a)[20]. Total CD8+ T cell numbers were unaffected, indicating a specific effect on proliferating antigen-specific cells (Extended Data Fig. 2b). Despite enhanced expansion, CD8+ T cells did not display traits of exhaustion, as PD-1 expression remained unaltered over time (Extended Data Fig. 2c). Together, these data indicate that enforced cell cycle arrest promotes differentiation of CD8+ T cells into effector cells with enhanced proliferative capacity.

### Altered metabolism and differentiation during cell cycle arrest

To interrogate the mechanisms underlying enhanced cell cycle progression after temporal cell cycle arrest, we characterized the transcriptional activity in unstimulated, arrested, released and non-arrested CD8+ T cells. Gene expression profiling showed that all four conditions showed distinct transcriptomic profiles, indicated by their segregation in principal component analysis ($q < 0.05$; Fig. 2a). From the transcriptome dataset, significant differentially expressed genes were selected ($q < 0.05$) and ingenuity pathway analysis (IPA) was performed to characterize the underlying molecular pathways. Differentially regulated pathways included cell cycle regulation (G1/S checkpoint regulation), cellular metabolism (for example, glycolysis, cholesterol biosynthesis) and T cell-specific differentiation (for example, type 1/2 helper T (T$_H$1/T$_H$2) pathway; Fig. 2b and Extended Data Fig. 3a). Based on these results, we performed gene-set enrichment analysis (GSEA) of Molecular Signatures Database (MSigDB) hallmark gene sets including cell cycle regulation (G2M checkpoint), metabolic pathways (that is, glycolysis, cholesterol biosynthesis, oxidative phosphorylation, fatty acid metabolism) and signaling pathways (IL-2–STAT5, MTORC1, PI3–AKT–MTOR signaling; Fig. 2c,d and Extended Data Fig. 3b,c). The metabolism-related gene sets were particularly upregulated in released CD8+ T cells, indicating that released cells adjusted their cellular metabolism.

We next profiled individual differentially expressed genes. Notably, arrested CD8+ T cells already exhibited enhanced expression of glycolysis-related and cholesterol biosynthesis-related genes compared to unstimulated cells (Fig. 2e and Extended Data Fig. 3d), indicating activation of glucose metabolism and cholesterol biosynthesis despite their non-proliferative state. In addition, transcription of glycolysis-related genes, such as *Pkm*, *Aldoa* and *Mtor*, and certain cholesterol synthesis-related genes, such as *Fdft1*, *Fdps* and *Mvd* was increased in released CD8+ T cells compared to non-arrested and arrested cells (Fig. 2e and Extended Data Fig. 3d). Transcripts of genes implicated in PI3K–AKT–mTOR signaling were also increased in arrested CD8+ T cells compared to unstimulated conditions, and many of these genes elevated to higher levels in released cells compared to non-arrested cells (Fig. 2e and Extended Data Fig. 3d). Correspondingly, expression levels of genes from the mitogen-activated protein kinase (MAPK)–c-MYC pathway, a pathway interconnected with both mTOR

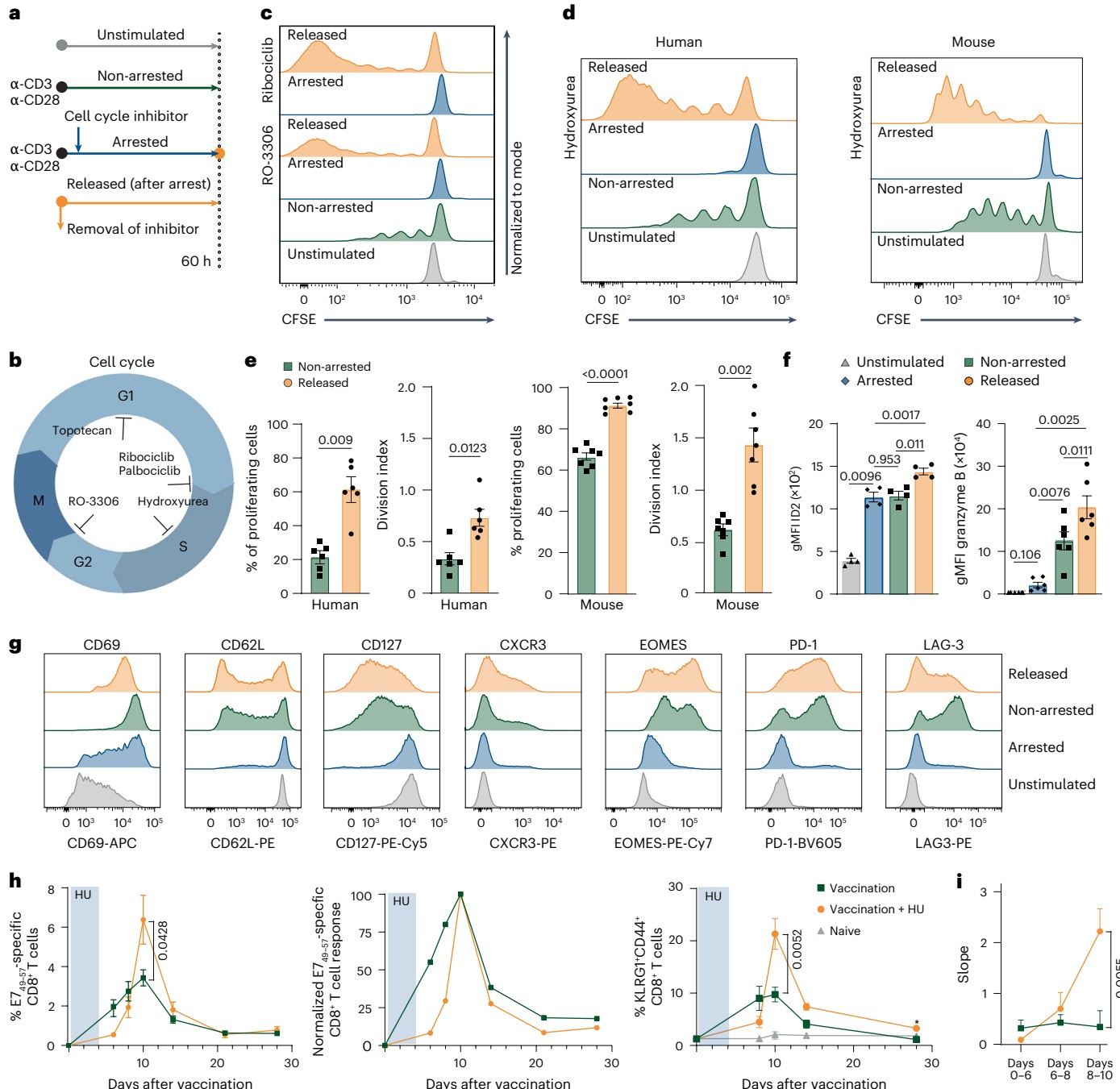

**Fig. 1 | Enhanced CD8⁺ T cell proliferation and effector function after temporal cell cycle inhibition. a**, Experimental setup. Experiments were performed with isolated CD8⁺ T cells derived from peripheral blood mononuclear cells from healthy donors or from splenocytes of naive mice. **b**, Schematic overview of cell cycle inhibition with different cell cycle inhibitors. **c**, Representative proliferation plots of unstimulated human CD8⁺ T cells, or ex vivo-stimulated human CD8⁺ T cells that were left untreated (non-arrested), arrested in the cell cycle using RO-3306 or ribociclib (arrested) or released from cell cycle arrest (released). **d**, Representative proliferation plots of unstimulated and ex vivo-stimulated human and mouse CD8⁺ T cells under the same conditions as in **c**, treated with HU. **e**, Percentage of proliferating cells and division index (mean ± s.e.m.) of ex vivo-stimulated human ($n = 8$) and mouse ($n = 7$) CD8⁺ T cells that were left untreated (non-arrested) or released after HU arrest. **f**, Expression (geometric mean fluorescence intensity (gMFI) ± s.e.m.) of ID2 (left, $n = 4$ mice)

and granzyme B (right, $n = 6$ mice) in unstimulated and ex vivo-stimulated mouse CD8⁺ T cells that were left untreated (non-arrested), arrested with HU or released from HU-induced arrest. **g**, Representative histograms of CD69, CD62L, CD127, CXCR3, EOMES, PD-1 and LAG-3 expression in ex vivo-stimulated mouse CD8⁺ T cells treated temporally with HU or left untreated. **h**, Naive mice were vaccinated with E7 SLP/CpG on day 0 and treated with HU for 4 consecutive days ($n = 5$) or left untreated ($n = 6$). Left, percentages of circulating E7$_{49–57}$-specific CD8⁺ T cells over time (mean ± s.e.m.). Middle, normalized response relative to the peak. Right, percentages of KLRG1⁺CD44⁺ CD8⁺ T cells over time (mean ± s.e.m.). **i**, Slope of the E7$_{49–57}$-specific CD8⁺ T cell response depicted in **h**. Statistical comparisons were performed with two-sided paired $t$-test (**e**), repeated-measures analysis of variance (ANOVA) with Sidak's multiple-comparison test (**f**) or two-sided unpaired $t$-test (**h** and **i**); $P$ values are shown on the graphs.

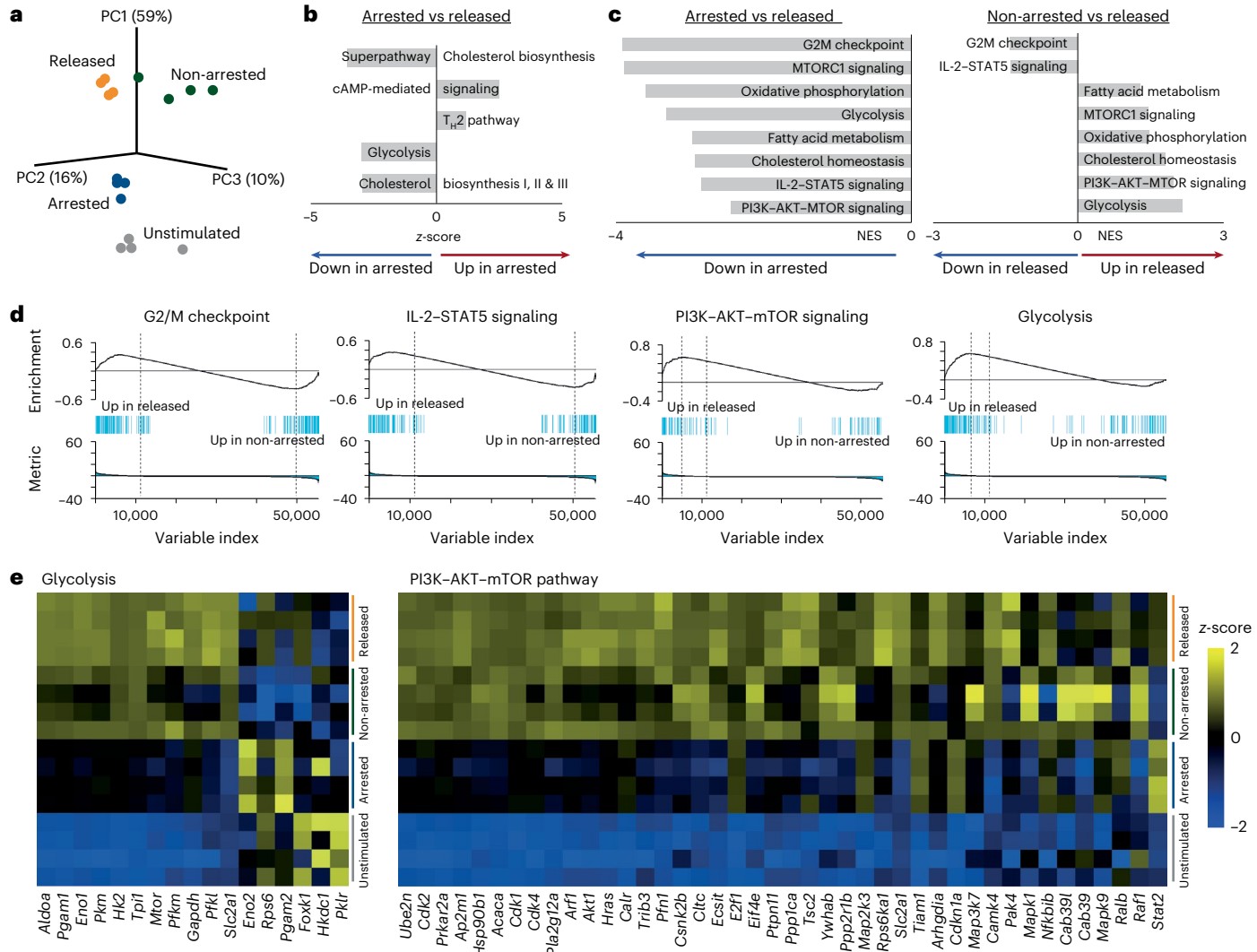

**Fig. 2 | Cell cycle arrest induces transcriptional remodeling of CD8⁺ T cell metabolism.** Transcriptomic analysis of unstimulated mouse CD8⁺ T cells, or ex vivo-stimulated mouse CD8⁺ T cells that were left untreated (non-arrested), arrested in the cell cycle using HU or released from HU-induced arrest (released; $n$ = 4). **a**, Three-dimensional principal component analysis plot. **b**, IPA of differentially expressed genes (FDR < 0.05) comparing arrested to released cells. The top five significantly enriched pathways with the most positive and negative $z$-scores are shown. Statistical analysis was performed using two-sided unpaired $t$-test with Benjamini–Hochberg correction. **c**, Normalized enrichment scores (NES) of eight selected mouse hallmark gene sets (from MSigDB; FDR < 0.05) based on GSEA. **d**, GSEA enrichment plots after GSEA analysis of the same hallmark gene sets. Genes are ranked on the $x$ axis by $\log_2$(fold change) in expression between released and non-arrested cells. Vertical bars represent individual genes within each gene set; the enrichment score is plotted on the $y$ axis. **e**, Heat maps of differentially expressed genes associated with glycolysis (left) and the PI3K–AKT–mTOR-pathway (continued in Extended Data Fig. 3d). Each row represents an individual sample.

signaling[21] and regulation of glucose metabolism[22], were elevated in released cells as well (for example, *Map2k1*, *Map2k2*, *Mapk3* and *Myc*; Extended Data Fig. 3d).

We next examined T$_H$1/T$_H$2-related transcripts. *Il2ra*, *Il2rb*, *Ccr4* and *Ccr5* were upregulated in arrested CD8⁺ T cells relative to unstimulated cells, indicating induction of a differentiation program during arrest. Expression of these transcripts further increased in non-arrested and released cells (Extended Data Fig. 3d). *Il2* mRNA was highest in arrested cells, whereas *Runx3* and *Havcr2* (Tim3) were most highly expressed in released cells, consistent with more advanced effector differentiation. The DNA damage response (DDR) pathway was not differentially regulated, reflecting the absence of key DDR transcripts in either arrested or released cells (Extended Data Fig. 4). Together, these data indicate that temporal cell cycle inhibition of activated CD8⁺ T cells enhances expression of genes involved in glycolysis, cholesterol biosynthesis and effector differentiation.

To assess whether the transcriptional programs of the arrested and released CD8⁺ T cells resemble those of resting and reactivated memory CD8⁺ T cells, we compared our mRNA-sequencing dataset to a recently published dataset profiling lymph node and tissue-resident memory CD8⁺ T cells in both resting and reactivated states[23]. Using the same EdgeR pipeline (false discovery rate (FDR) < 0.05), we identified 1,519 overlapping differentially expressed genes (Extended Data Fig. 5). While certain genes were similarly upregulated (such as *Il2*) or downregulated in both arrested and reactivated cells, many changes, including those in glycolytic and effector genes (*Pkm*, *Aldoa*, *Pgam1*, *Gapdh*, *Eno1*, *Gzmb*), were shared between the released and reactivated conditions. These findings indicate that arrested CD8⁺ T cells largely mirror the transcriptional profile of resting memory T cells, whereas released CD8⁺ T cells acquire gene expression patterns characteristic of reactivated memory T cells, suggesting that memory-like features including reactivation properties are already imprinted during the arrested state.

## Arrested CD8+ T cells stockpile nutrients and increase glycolysis

To complement the transcriptomic analysis and further define metabolic adaptations induced by cell cycle inhibition and release, we performed intracellular metabolite profiling of CD8+ T cells by mass spectrometry. Arrested CD8+ T cells showed elevated levels of hexose (including glucose) and several amino acids, such as glutamine and aspartate (Fig. 3a), indicating active nutrient accumulation during arrest. Released CD8+ T cells also exhibited increased hexose levels, but their amino acid levels were reduced compared to non-arrested cells (Fig. 3a).

The differences in intracellular glucose and amino acid levels prompted us to examine the underlying metabolic pathways. To link these changes to functional adaptations, we selected key nutrient transporters and metabolic enzymes for single-cell validation by spectral flow cytometry, focusing on molecules involved in amino acid uptake, glycolysis and the pentose phosphate pathway[24,25]. Expression of the amino acid transporter CD98 was upregulated in arrested cells and remained elevated in non-arrested and released cells, suggesting that arrested cells acquired the ability to take up and stockpile amino acids by upregulating transporter expression (Extended Data Fig. 6a,b). Expression of the glucose transporter GLUT1 was also upregulated in arrested cells, but further increased upon release, indicating enhancement of glucose metabolism. These higher levels coincided with the increased levels of PKM and ALDOA, key enzymes in glycolysis, and with enhanced uptake of the glucose analog 2-NBDG, and increased G6PD, the rate-limiting enzyme of the pentose phosphate pathway (Fig. 3b,c and Extended Data Fig. 6c). Notably, released cells exhibited the highest levels of GLUT1, PKM, ALDOA, G6PD and 2-NBDG uptake exceeding those of non-arrested cells, indicating superior glucose metabolic activity following release from arrest.

Increased PKM expression was specifically observed during the early S phase in HU-arrested cells, which displayed only low DNA content as determined by FxCycle staining (Extended Data Fig. 6d). Because HU blocks progression beyond the S phase, no cells advanced into G2/M under arrested conditions. Upon release from HU, however, cells progressed through the cell cycle and displayed elevated PKM levels in both early and late S phases as well as in G2/M. Notably, PKM expression remained low in the G0/G1 phase across arrested, non-arrested and released conditions, highlighting a cell cycle-linked regulation of PKM that is associated with DNA replication and mitotic entry. Consistent with our ex vivo findings, expression of CD98, G6PD, PKM and GLUT1 was enhanced in vaccine-elicited CD8+ T cells residing in blood and lymph nodes following transient cell cycle inhibition with either HU or palbociclib (Fig. 3d,e and Extended Data Fig. 6e).

As arrested cells do not undergo energy-intensive cell cycle progression yet continue to take up glucose, we interrogated whether glucose was stockpiled as glycogen rather than used for energy[26]. While unstimulated cells negligibly stored glycogen, arrested CD8+ T cells accumulated glycogen (Fig. 3f). This accumulation depended on glucose uptake, as GLUT1 inhibition with WZB117 prevented glycogen storage during arrest (Fig. 3g). Although non-arrested CD8+ T cells also stored glucose, released cells rapidly depleted their glycogen stores (Fig. 3f), consistent with their elevated proliferation and increased glycolytic activity. Restraining glycogen breakdown by selective inhibition of glycogen phosphorylase[27] using CP91149 impaired proliferation in a dose-dependent manner, indicating that cell-intrinsic glycogenolysis and glycolytic activity supports proliferation following transient cell cycle arrest (Fig. 3h). In line with this, WZB117 reduced proliferation of both non-arrested and released cells, underscoring the critical role of glucose metabolism to support proliferation (Fig. 3i). Notably, a shorter period of cell cycle arrest (12 h), permitting less time for stockpiling nutrients such as glucose, enhanced proliferation upon release but to a lesser extent (as compared to 60 h), highlighting

the functional relevance of metabolic preconditioning during arrest (Extended Data Fig. 6f).

Next, we investigated the impact of cell cycle arrest on energy production in the tricarboxylic acid (TCA) cycle in mitochondria by assessing levels of SDHA and ATP5a, and mitochondrial reactive oxygen species (by MitoSOX). Despite being non-proliferative, the activity of the TCA cycle and mitochondria increased in arrested CD8+ T cells. Compared to non-arrested cells, released cells displayed no substantial alterations in expression of the TCA cycle-related enzymes, but mitochondrial reactive oxygen species was further enhanced, indicating increased mitochondrial activity (Fig. 4a,b).

Based on our transcriptomic analysis, we next examined fatty acid metabolism and cholesterol biosynthesis. Fatty acid metabolism is linked to T cell differentiation and long-term function, whereas cholesterol biosynthesis is critical for proliferating CD8+ T cells by supporting membrane biogenesis and signaling[28]. BODIPY-labeled FL-C16 staining was similar between unstimulated and arrested cells (Extended Data Fig. 6g,h), suggesting no difference in uptake of palmitate fatty acids. However, arrested cells showed signs of enhanced fatty acid processing capacity compared to unstimulated cells, as CPT1a, an essential enzyme for beta oxidation of long-chain fatty acids, was increased and further elevated upon release (Fig. 4c,d). Consistent with the transcriptomic data, arrested CD8+ T cells also upregulated FDFT1 (squalene synthase), a key enzyme in cholesterol biosynthesis, both ex vivo and in vivo (Fig. 4c–e and Extended Data Fig. 6i). FDFT1 expression was further increased in released CD8+ T cells and exceeded levels observed in non-arrested cells. Restricting FDFT1 using zaragozic acid impaired the proliferation of non-arrested cells but not of released cells, which may be related to their higher FDFT1 levels (Fig. 4e). Inhibition of the rate-limiting enzyme HMG-CoA reductase with atorvastatin suppressed proliferation in both non-arrested cells and released CD8+ T cells (Fig. 4e).

Together, these data show that cell cycle-arrested CD8+ T cells display enhanced cholesterol metabolism, elevated glucose metabolism, increased mitochondrial activity and greater TCA cycle engagement. Moreover, when given sufficient time to accumulate nutrients during arrest, these cells become metabolically primed for enhanced proliferation upon release.

## CD8+ T cell proliferation after cell cycle inhibition is partially mTOR independent

We next investigated the molecular mechanisms driving glucose metabolism in CD8+ T cells during temporal cell cycle arrest. Phosphoproteomic analysis of released and non-arrested human CD8+ T cells revealed clusters of phosphorylation sites linked to activated glycolysis, transcription factors, FOXK1 and FOXK2, which regulate glycolysis-related genes[29], and activation of MAPK–c-MYC and JAK–STAT signaling (Fig. 5a). Consistent with the transcriptome data, DDR-associated phosphosites were detectable but did not exhibit the characteristic phosphorylation pattern indicative of DNA damage or replication stress, such as activation of core DDR proteins including ATM, ATR, PRKDC and CHEK1/CHEK2 and components of the FANC pathway (Extended Data Fig. 7a). Kinase phosphosite analysis revealed a coordinated cell cycle progression pattern concurrent with enhanced phosphorylation of multiple cell cycle-regulatory kinases (Extended Data Fig. 7b).

To evaluate directly whether transient cell cycle inhibition induces DNA damage, we assessed γ-H2AX expression, a marker of DNA double-strand breaks, in CD8+ T cells[30]. HU-arrested cells showed elevated γ-H2AX at 60 h, consistent with replication of stress-associated DNA damage caused by stalled replication forks[31], an effect that is exacerbated by prolonged HU exposure. Upon release from HU, γ-H2AX levels were reduced, likely reflecting DNA repair. In contrast, treatment with palbociclib or ribociclib did not induce detectable DNA damage, and released cells exhibited only a modest increase in γ-H2AX

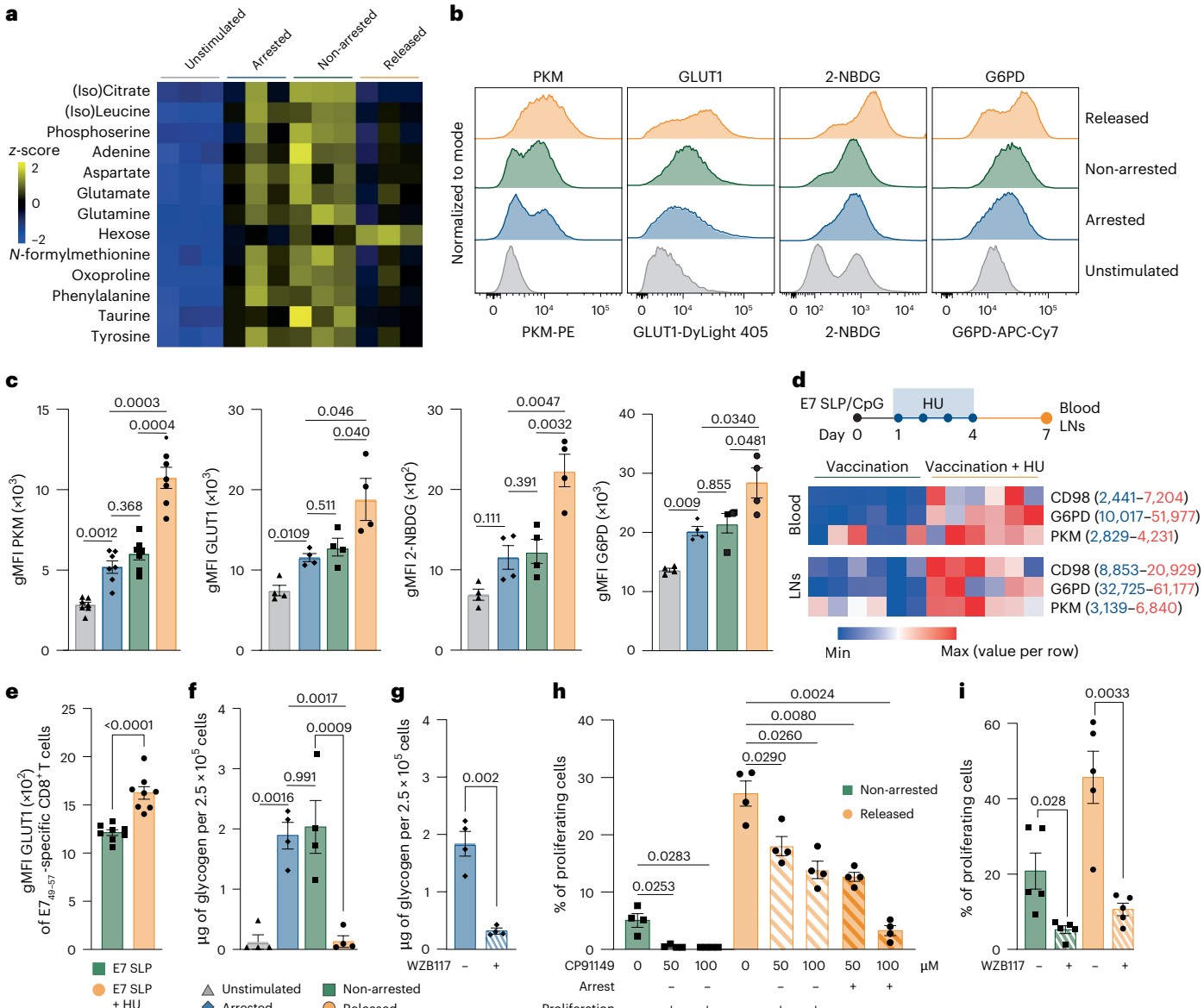

**Fig. 3 | Cell cycle-arrested CD8+ T cells stockpile nutrients and increase glucose metabolism. a**, Heat map of differentially expressed metabolites extracted from unstimulated human CD8+ T cells, or ex vivo-stimulated human CD8+ T cells or ex vivo-stimulated mouse CD8+ T cells that were left untreated (non-arrested), cell cycle-arrested with HU (arrested) or released from HU-induced arrest (released; $n = 3$ donors). z-scores are color coded. **b,c**, Representative histograms (**b**) and gMFI (± s.e.m.; **c**) of PKM ($n = 7$), GLUT1 ($n = 4$), 2-NBDG ($n = 4$) and G6PD ($n = 4$) expression in unstimulated and ex vivo-stimulated human CD8+ T cells (HU arrested, non-arrested and released from HU arrest). Each symbol represents one healthy donor ($n = 4$). **d**, Heat map of CD98, G6PD and PKM expression in E7$_{49-57}$-specific CD8+ T cells at day 7 after E7 SLP/CpG vaccination in the blood and lymph nodes (LNs) of mice treated with HU on days 1–4. Geometric mean is color coded, and marker-specific ranges are indicated ($n = 6$ mice per group). **e**, gMFI (± s.e.m.)

of GLUT1 on circulating E7$_{49-57}$-specific CD8+ T cells at day 7 after vaccination ($n = 8$ mice per group). **f**, Glycogen levels (mean ± s.e.m.) in unstimulated, and ex vivo-stimulated human CD8+ T cells (HU-arrested, non-arrested and released from HU arrest; $n = 4$ donors). **g**, Glycogen levels (mean ± s.e.m.) in HU-arrested human CD8+ T cells treated with WZB117 ($n = 4$ donors). **h**, Percentage of proliferating ex vivo-stimulated human CD8+ T cells (mean ± s.e.m.; $n = 4$ donors) that were either HU-arrested and subsequently released or left untreated (non-arrested), and the same conditions in which CP91149 was added during arrest or proliferation. **i**, Percentage of proliferating non-arrested and HU-released human CD8+ T cells (mean ± s.e.m., $n = 5$ donors) treated with WZB117. Statistical comparisons were performed using repeated-measures ANOVA with Sidak's multiple-comparisons test (**c**, **f** and **h**) and two-sided unpaired (**e**) and paired (**g** and **i**) t-tests; P values are shown on the graphs.

expression (Extended Data Fig. 8a,b). Non-arrested cells displayed low γ-H2AX expression at 24 h, which increased after 60 h of stimulation, consistent with replication-associated stress during continuous proliferation. Together, these data show that transient cell cycle inhibition and subsequent release do not cause substantial or lasting DNA damage.

FOXK1 and FOXK2 were differentially phosphorylated in released and non-arrested cells (Fig. 5a). Phosphorylation of FOXK1 at the C-terminal sites Ser441, Thr436, Ser445, Ser472 and Ser416 was

observed in released cells, while non-arrested cells showed phosphorylation of the N-terminal sites Ser213 and Ser223. In accordance with the enhanced glycolytic state, FOXK1 nuclear translocation was increased in arrested and released CD8+ T cells compared to non-arrested (Fig. 5b).

The nutrient-sensitive mammalian target of rapamycin complex 1 (mTORC1) signaling pathway is important for the translocation of FOXK1/FOXK2 (ref. 32). To determine the role of this pathway, we assessed the activity of mTOR by measuring the phosphorylation of the downstream target ribosomal protein S6 (ref. 33; RPS6, S6). Arrested

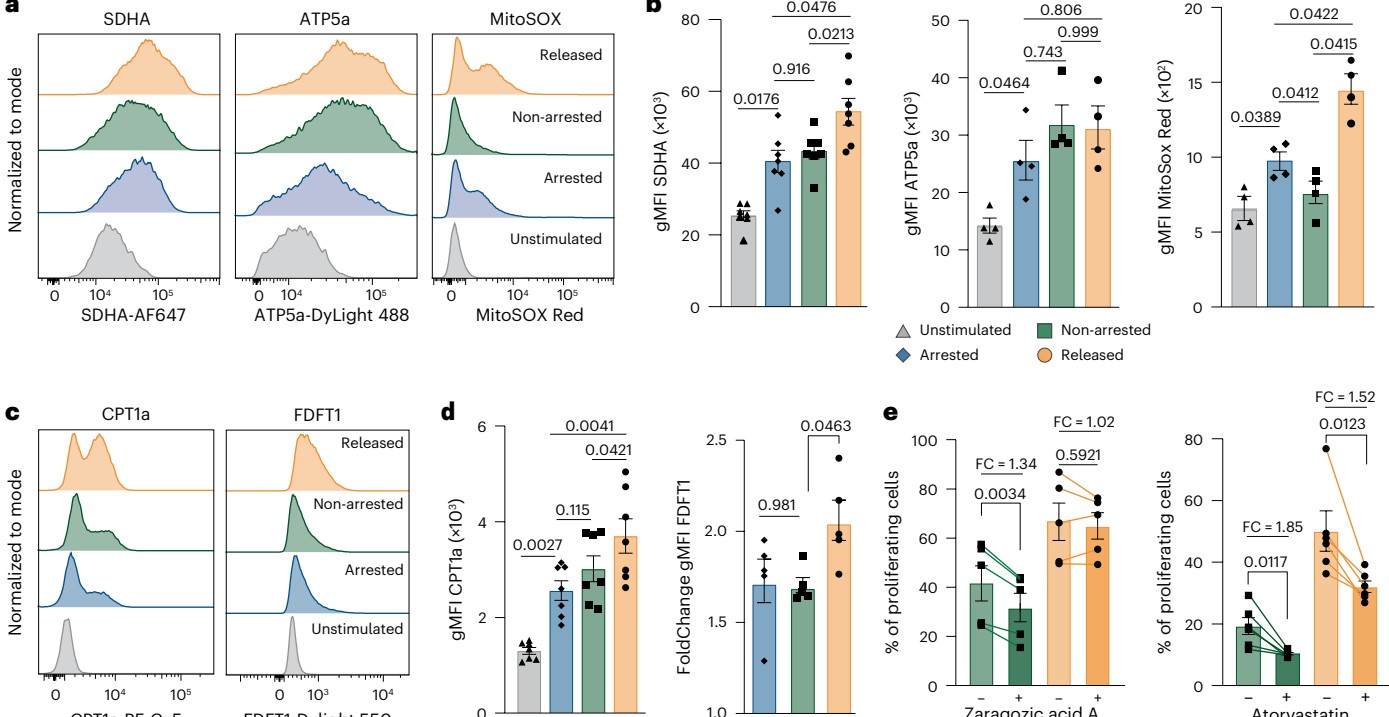

**Fig. 4 | Temporal cell cycle arrest of CD8+ T cells modulates mitochondrial activity and cholesterol metabolism. a,b,** Representative histograms (**a**) and gMFI (± s.e.m.; **b**) of SDHA (*n* = 7 donors), ATP5a (*n* = 4 donors) and MitoSox Red (*n* = 4 donors) in unstimulated human CD8+ T cells or ex vivo-stimulated human CD8+ T cells that were left untreated (non-arrested), cell cycle-arrested with HU (arrested) or released from HU-induced arrest (released). **c,d,** Representative histograms (**c**) and gMFI (± s.e.m.; **d**) of CPT1a (*n* = 7 donors) and FDFT1 (*n* = 5 donors) under the same conditions. FDFT1 gMFI is indicated as fold change

relative to unstimulated cells. **e,** Percentage (mean ± s.e.m.) of proliferating human CD8+ T cells that were ex vivo-stimulated (non-arrested and released after HU arrest) and treated with zaragozic acid (left graph, *n* = 5) or atorvastatin (right graph, *n* = 6) during proliferation to inhibit cholesterol biosynthesis. Lines indicate individual donors. Statistical comparisons were performed using repeated-measures ANOVA with Sidak's multiple comparisons (**b** and **d**) and two-sided paired *t*-test (**e**); *P* values are shown on the graphs.

cells showed activation of mTOR by increased levels of pS6 compared to unstimulated cells (Fig. 5c,d). Released cells, however, showed decreased levels of pS6 compared to non-arrested cells (Fig. 5c,d), a finding in agreement with the decreased RPS6 phosphorylation at Ser205 (Fig. 5a). Kinetic analysis showed that non-arrested cells remained high in pS6 levels (up to at least 120 h after stimulation), while after cell cycle arrest the released cells showed decreased levels, already noticeable 24 h after release (Extended Data Fig. 8c).

To investigate mTORC1 directly, we used CD8+ T cells lacking Raptor, a binding protein of mTORC1 and critical for its activity, obtained from CD8-Cre-*Rptor* (Raptor) mice. While non-arrested Raptor-deficient CD8+ T cells were substantially blocked in their proliferation, released Raptor-deficient CD8+ T cells still proliferated considerably (Fig. 5e,f). Notably, released Raptor-deficient CD8+ T cells showed higher PKM expression as non-arrested Raptor-deficient cells, indicating that after cell cycle blockade CD8+ T cells maintain a high glycolytic activity independent of mTOR signaling (Fig. 5g). Decreased mTORC1 dependency was further evident when CD8+ T cells were treated with the mTOR inhibitor rapamycin during proliferation. Proliferation of released mouse and human CD8+ T cells was modestly reduced (1.5-fold and 1.2-fold, respectively), whereas non-arrested cells were more sensitive to mTOR inhibition, showing 3.4-fold and 7.8-fold reductions, respectively (Fig. 5h). Rapamycin treatment during both arrest and release still allowed proliferation of mouse and human CD8+ T cells, with only 2.3-fold and 1.8-fold reductions, respectively. These results indicate that, while mTOR is critical for proliferation of non-arrested cells, released CD8+ T cells are only partially mTOR dependent, suggesting activation of alternative pathways driving cell cycle progression after transient arrest.

## IL-2-mediated cell cycle progression after cell cycle arrest

To identify pathways that could potentially bypass the mTORC1 pathway, we investigated the role of the activated MAPK–c-MYC pathway, known to be implicated in the induction of T cell proliferation and glycolysis[34]. Our transcriptomic analysis showed enhanced expression of genes of the MAPK pathway, including *Myc*, in released cells compared to non-arrested cells (Extended Data Fig. 3d). Moreover, the phosphoproteomics analysis confirmed that MAPK–c-MYC signaling is more activated in released CD8+ T cells compared to non-arrested cells (Extended Data Fig. 8d). MYC inhibition during proliferation, however, only partially inhibited the proliferation (1.2-fold) of released CD8+ T cells (Extended Data Fig. 8e), indicating involvement of MYC-driven and MYC-independent pathways.

The partial mTORC1 and MYC independence of released cells to proliferate prompted us to assess the role of IL-2, given its importance for CD8+ T proliferation[2,4], and signaling capacity via the alternate STAT5–PI3K pathway[35]. Consistent with our transcriptomic data, the percentage of IL-2-producing cells was increased in cell cycle-arrested CD8+ T cells as compared to unstimulated and non-arrested cells (Fig. 6a and Extended Data Fig. 9a). Moreover, the IL-2 production on a per-cell basis was enhanced in arrested cells (Fig. 6b), which coincided with high amounts of IL-2 in the supernatant (Fig. 6c and Extended Data Fig. 9b). Arrested cells also produced more IL-2 compared to released cells, indicating that IL-2 is consumed during proliferation, with production returning to levels comparable to those in non-arrested CD8+ T cells (Fig. 6a–c and Extended Data Fig. 9a,b). To corroborate these findings, we tracked IL-2 expression in CD8+ T cells from IL-2-GFP reporter mice. Cell cycle arrest ex vivo led to an increased frequency of GFP-hi CD8+ T cells compared to non-arrested settings, and this percentage

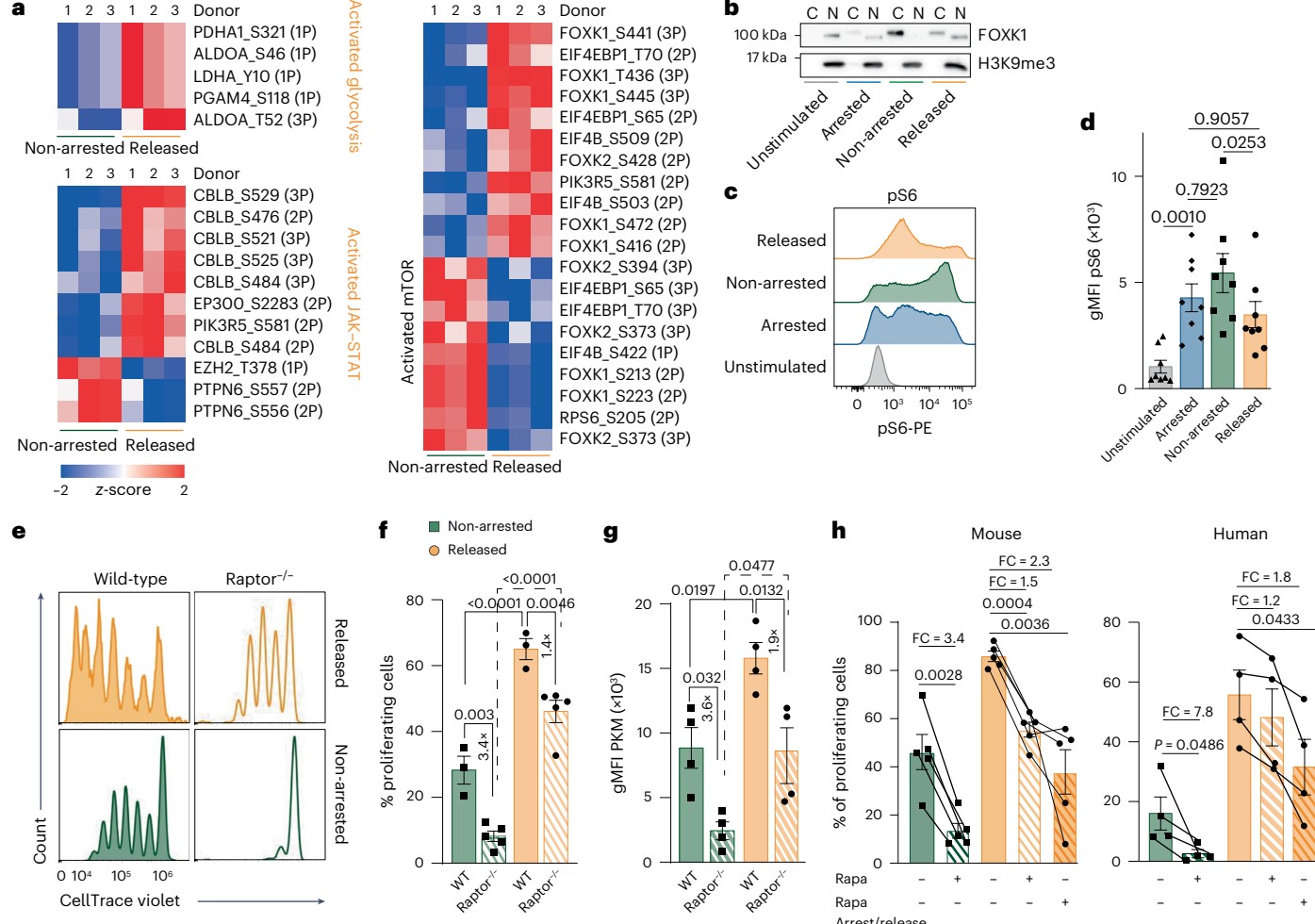

**Fig. 5 | CD8⁺ T cell proliferation after cell cycle inhibition is partially mTOR independent. a**, Heat map of hierarchical clustered *z*-scored phosphosite intensities measured by mass spectrometry in ex vivo-stimulated human CD8⁺ T cells that were either released from HU-induced arrest or left untreated (non-arrested; *n* = 3 donors). Coupled *z*-scores are based on normalized phosphosite intensities. The number of phospho groups per site is indicated in parentheses (3P indicates three or more phospho groups). **b**, Immunoblot of unstimulated human CD8⁺ T cells or ex vivo-stimulated human CD8⁺ T cells that were left untreated (non-arrested), cell cycle-arrested with HU (arrested) or released from HU-induced arrest (released). FOXK1 expression was assessed in the cytosolic (C) and nuclear (N) fractions. H3K9me3 was used as the loading control. **c,d**, Representative histograms (**c**) and gMFI (± s.e.m.; **d**) of phosphorylated S6 (pS6) in unstimulated and ex vivo-stimulated human CD8⁺ T cells (HU-arrested, non-arrested and

released from HU arrest; *n* = 8). **e**, Representative CellTrace Violet plots of ex vivo-stimulated (non-arrested and released from HU arrest) CD8⁺ T cells derived from wild-type (WT) and mTORC1-deficient (*Raptor⁻/⁻*) mice. **f**, Percentage of proliferating ex vivo-stimulated CD8⁺ T cells (mean ± s.e.m.) from WT (*n* = 3) and *Raptor⁻/⁻* (*n* = 5) mice that were non-arrested or released from HU arrest. **g**, PKM expression (gMFI ± s.e.m.) in ex vivo-stimulated CD8⁺ T cells from WT and *Raptor⁻/⁻* mice (*n* = 4) under the same conditions as in **f**. **h**, Percentage of proliferating ex vivo-stimulated mouse and human CD8⁺ T cells (mean ± s.e.m.) that were either released from HU arrest (released) or non-arrested, and treated with rapamycin either only during proliferation or during both HU arrest and release. Lines indicate individual mice or donors (*n* = 4). Statistical comparisons were performed using repeated-measures ANOVA with Sidak's multiple comparisons (**d**, **f**, **g** and **h**); *P* values are shown on the graphs. FC, fold change.

decreased upon release from arrest (Fig. 6d,e). To determine whether the enhanced IL-2 production observed during ex vivo cell cycle arrest also occurred in vivo, we vaccinated IL-2^GFP reporter mice with E7 peptide in the presence or absence of HU treatment. IL-2^GFP expression was markedly increased in E7-specific CD8⁺ T cells during HU treatment compared to their counterparts in untreated mice or after HU withdrawal (Fig. 6f), confirming our ex vivo findings. The frequency of IFNγ-producing CD8⁺ T cells also increased during arrest but remained lower than in non-arrested or released CD8⁺ T cells (Extended Data Fig. 9c). A substantial fraction of IL-2-producing CD8⁺ T cells coexpressed TNF (Extended Data Fig. 9d,e), indicating enhanced cytokine polyfunctionality, which together with elevated autocrine IL-2 levels on a per-cell basis are hallmarks of memory T cells with superior expansion potential[2].

Consistent with enhanced IL-2 transcription, expression of cREL, a key IL-2-regulating transcription factor[36], was substantially increased

in arrested cells compared to unstimulated cells, and remained higher compared to non-arrested and released cells (Extended Data Fig. 9f). NFAT, another critical regulator of IL-2 expression[37], was also increased in arrested cells compared to unstimulated cells, but its expression was comparable to non-arrested cells and lower than in released cells (Extended Data Fig. 9f).

Next, we assessed the capacity of CD8⁺ T cells to respond to IL-2 by examining expression of the high-affinity IL-2 receptor CD25 (IL-2Rα). Notably, a substantial fraction of the arrested cells already showed expression of CD25, indicating that IL-2 responsiveness is established during arrest (Fig. 6g,h and Extended Data Fig. 9g). While CD25 expression on non-arrested cells increased with each cell division, the faster-cycling released cells started to down-modulate CD25 proliferation, and this down-modulation was profoundly evident for human CD8⁺ T cells (Fig. 6g,h).

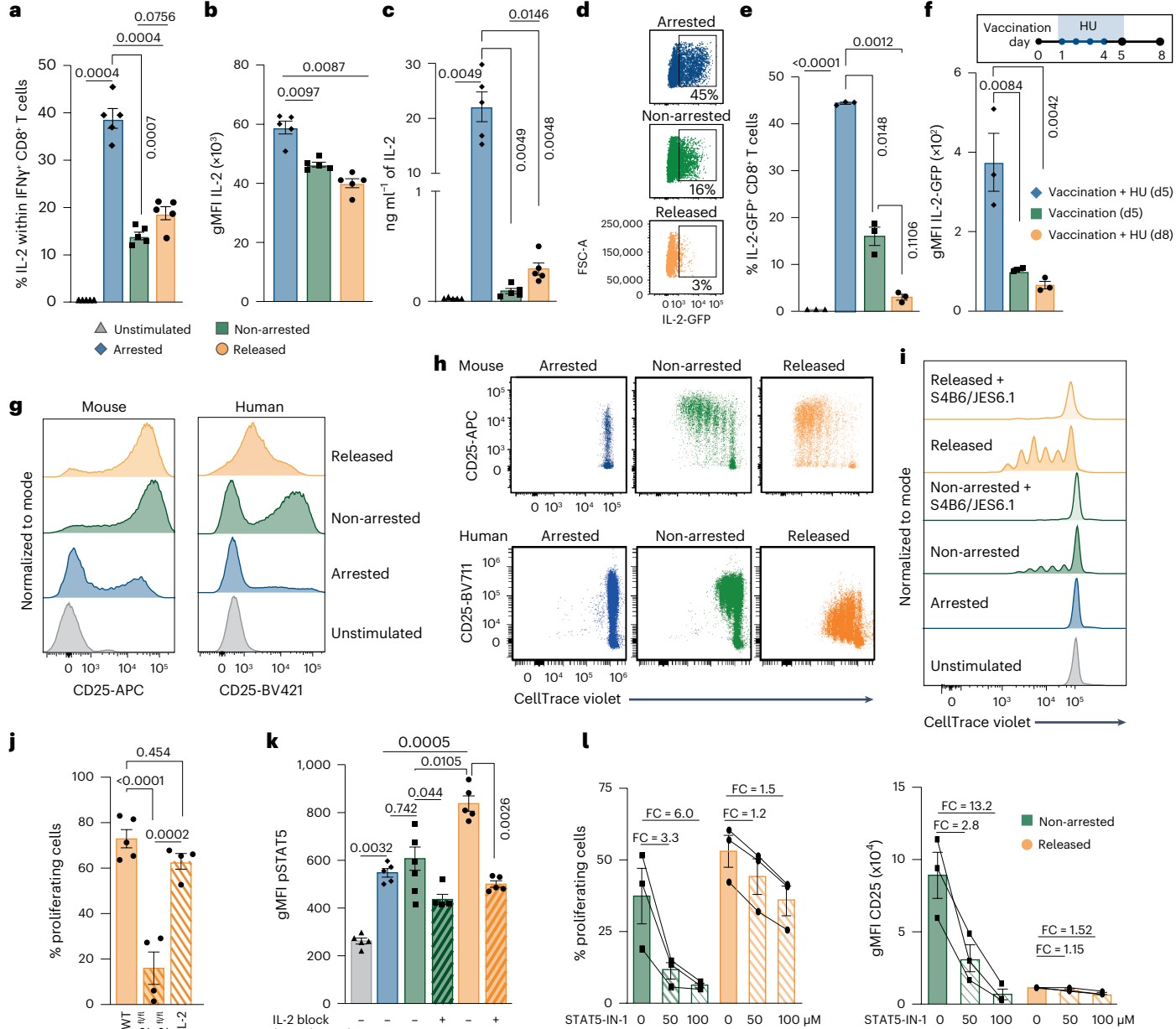

**Fig. 6 | IL-2 production during cell cycle arrest drives enhanced CD8+ T cell proliferation. a**, Percentage of IL-2-producing cells (mean ± s.e.m.) among unstimulated IFNγ+ mouse CD8+ T cells, or ex vivo-stimulated IFNγ+ mouse CD8+ T cells that were left untreated (non-arrested), arrested in the cell cycle using HU or released from HU-induced arrest (*n* = 5 mice). **b**, gMFI (± s.e.m.) of IL-2 in IL-2+IFNγ+ mouse CD8+ T cells under the same conditions as in **a** (*n* = 5 mice). **c**, IL-2 concentration (mean ± s.e.m.) in supernatants of unstimulated mouse CD8+ T cells, or ex vivo-stimulated mouse CD8+ T cells that were non-arrested, HU-arrested or released from HU-induced arrest (*n* = 5 mice). **d**, Representative flow cytometry plots of ex vivo-stimulated IL-2GFP+ CD8+ T cells from IL-2GFP reporter mice under ex vivo-stimulated conditions (HU-arrested, non-arrested and released). **e**, Percentage of IL-2GFP+ CD8+ T cells (mean ± s.e.m.) under the same conditions as in **d** and in unstimulated conditions (*n* = 3 mice). **f**, IL-2GFP reporter mice were vaccinated with E7 SLP/CpG and treated with HU for 4 consecutive days (days 1–4). IL-2GFP expression (gMFI ± s.e.m.) was measured in E7₄₉–₅₇-specific CD8+ T cells from spleens at days 5 and 8 after vaccination (*n* = 3

mice). **g**, Representative histograms of CD25 expression in mouse and human CD8+ T cells that were unstimulated or ex vivo-stimulated and non-arrested, HU-arrested or released. **h**, Representative flow cytometry plots of CellTrace Violet versus CD25 expression of HU-arrested, non-arrested and released mouse and human CD8+ T cells ex vivo. **i**, Representative CellTrace Violet plots of mouse CD8+ T cells under the same conditions as in **g**, with IL-2-neutralizing antibodies (S4B6 and JES6.1) added during proliferation. **j**, Percentage of proliferating CD8+ T cells (mean ± s.e.m.) from WT and IL-2fl/fl mice that were ex vivo stimulated and released from HU arrest. IL-2 was supplemented during proliferation (*n* = 5 WT mice, *n* = 4 IL-2fl/fl mice). **k**, pSTAT5 levels (gMFI ± s.e.m.) in mouse CD8+ T cells under the same conditions as in **i** (*n* = 5 mice). **l**, Percentage of proliferating cells (left) and CD25 expression (right; gMFI ± s.e.m.) of ex vivo-stimulated human CD8+ T cells (non-arrested and HU-released) with or without addition of STAT5-IN-1 supplemented during proliferation (*n* = 3 donors). Statistical comparisons were determined by repeated-measures ANOVA with Sidak's multiple comparisons (**a**–**c**, **e**, **f**, **j** and **k**); *P* values are shown on the graphs.

We then examined the extent to which enhanced IL-2 production and signaling contribute to the increased proliferation observed in released CD8+ T cells. Abrogation of IL-2 signaling with neutralizing

antibodies (S4B6 and JES6.1) inhibited proliferation in both released and non-arrested CD8+ T cells (Fig. 6i), indicating a requirement for IL-2-driven proliferation. Furthermore, IL-2-deficient CD8+ T cells

showed limited proliferation upon release, which was restored by exogenous IL-2 supplementation (Fig. 6j). IL-2 downstream signaling as determined by quantification of phosphorylated STAT5 (pSTAT5) levels was already induced in arrested CD8+ T cells compared to nonresponding unstimulated cells (Fig. 6k). pSTAT5 levels further increased during release in an IL-2-dependent manner to levels exceeding those in the non-arrested cells, indicating that IL-2 downstream signaling is increased after cell cycle blockade (Fig. 6k). Pharmacological inhibition of IL-2 signaling by STAT5-IN-1, a potent and selective STAT5 inhibitor (IC$_{50}$ 47 μM) at a concentration of 50 μM or 100 μM, resulted in stronger inhibition of non-arrested than released mouse and human CD8+ T cells (Fig. 6l and Extended Data Fig. 9h–j). These data indicate that while both cell populations rely on IL-2 signaling, released CD8+ T cells exhibit stronger IL-2 signaling. Collectively, these findings demonstrate that cell cycle-arrested CD8+ T cells produce elevated levels of IL-2, both ex vivo and in vivo, leading to enhanced downstream IL-2 signaling and supporting of IL-2-dependent proliferation upon release from arrest.

## Temporal cell cycle blockade improves immunotherapy

To evaluate the potential therapeutic relevance of our findings, we assessed the impact of transient cell cycle arrest in the immunogenic MC-38 tumor model. HU treatment induced increased expression of GLUT1, PKM, G6PD and CD98 in blood-circulating memory/effector CD8+ T cells of MC-38 tumor-bearing mice (Fig. 7a–c). Phenotypically, these cells exhibited high KLRG1 and CD43$^{1B11}$ expression (Extended Data Fig. 10a,b), and these cells were also increased in the tumor, which is shown to correlate with improved tumor control[38] (Fig. 7d).

Tumor-specific CD8+ T cell responses, detected by reactivity to the MuLV p15E (M8) tumor antigen using major histocompatibility complex class I tetramers, increased to higher levels over time in the blood of HU-treated mice compared to non-treated mice (Fig. 7a,e). During arrest, these tumor-specific CD8+ T cells exhibited a Ki-67$^{lo}$ profile, but Ki-67 expression rapidly increased upon HU withdrawal, and this correlated with increased GLUT1, PKM, G6PD and CD98 expression (Fig. 7a,e and Extended Data Fig. 10a). A similar increase in Ki-67$^{hi}$ CD8+ T cells was observed in the lymph nodes after treatment (Fig. 7f). Furthermore, metabolic profiling of M8-specific CD8+ T cells in the lymph nodes revealed enhanced glucose uptake (GLUT1), glucose metabolism (PKM, G6PD) and cholesterol biosynthesis (FDFT1) after HU treatment compared to M8-specific CD8+ T cells of untreated mice (Fig. 7f).

Next, we evaluated whether transient cell cycle blockade affects CD8+ T cell metabolism in a clinical setting[39]. In biopsy samples from individuals with stage II/III breast cancer, the frequency of GLUT1-expressing CD8+ T cells increased in the majority of individuals following intermittent ribociclib treatment compared to baseline levels during letrozole monotherapy (Fig. 7g–i). To further explore therapeutic implications, we assessed the impact of transient cell cycle arrest across multiple in vivo immunotherapy models, including adoptive T cell transfer, immune checkpoint blockade and therapeutic vaccination.

To assess efficacy of transient cell cycle arrest in the context of adoptive T cell transfer, we transferred ex vivo HU-treated or untreated OT-I cells into EG7 tumor-bearing mice. While mice that did not receive OT-I cells succumbed to tumor challenge, transfer of untreated OT-I cells delayed tumor outgrowth. Notably, transfer of OT-I cells treated with HU ex vivo (60 h before transfer) resulted in complete EG7 tumor regression (Fig. 7j).

In the MC-38 model, provision of HU in vivo delayed tumor progression in a CD8+ T cell-dependent manner (Fig. 7k and Extended Data Fig. 10c). Furthermore, combining transient HU treatment with αPD-L1 immune checkpoint blockade led to greater inhibition of MC-38 tumor growth and prolonged survival compared to either treatment alone (Fig. 7k and Extended Data Fig. 10d). Lastly, we assessed whether transient cell cycle blockade could enhance therapeutic vaccination. Administration of HPV E7 peptide vaccines together with HU treatment substantially improved survival in tumor-bearing mice, suggesting that transient cell cycle arrest augments the antitumor capacity of vaccine-elicited CD8+ T cells (Fig. 7l).

## Discussion

Cell cycle inhibitors are widely used in cancer therapy. We show that transient arrest of activated CD8+ T cells promotes highly proliferative effector cells ex vivo and in vivo. This enhanced proliferation is driven by metabolic reprogramming during arrest, when CD8+ T cells stockpile nutrients and rewire key pathways, acquiring an energetic state that enables rapid proliferation upon release. Arrested cells also produce elevated IL-2, which is critical for driving this post-arrest expansion.

In particular, glucose and cholesterol metabolism were enhanced after cell cycle blockade. Glycolysis is essential for proliferation and differentiation of naive T cells, as this metabolic pathway essentially provides a fast route for ATP production[40,41]. During cell cycle blockade, the intake of glucose via the glucose transporter is already increased resulting in stockpiling glucose in the form of glycogen. This, together with higher levels of PKM and ALDOA, supports enhanced proliferation after the arrest. Cholesterol biosynthesis, essential for proliferation through its role in membrane biogenesis and lipid raft-mediated signaling[28,42], was elevated during cell cycle arrest and further increased upon release. Consistent with their higher FDFT1 expression, released

**Fig. 7 | Temporal cell cycle arrest synergizes with immunotherapy.**
**a**,**b**, Uniform manifold approximation and projection (UMAP) analysis of blood CD44+CD8+ T cells from untreated ($n = 8$) and HU-treated (day 4–7, $n = 15$) MC-38 tumor-bearing mice (day 11 after challenge). After downsampling to 40,000 cells per group, UMAP embedding was performed. **a**, Contour plots (blue indicates low density; red indicates high density). **b**, Expression overlays of CD98, GLUT1, PKM and G6PD. **c**, Expression levels (mean ± s.e.m.) of CD98, GLUT1, PKM and G6PD in blood-derived KLRG1+CD8+ T cells from untreated ($n = 8$) and HU-treated (days 4–7, $n = 15$) MC-38 tumor-bearing mice on day 7 and 11 after tumor challenge. **d**, Frequency (mean ± s.e.m.) of intratumoral KLRG1+CD43$^{1B11+}$ CD8+ T cells in untreated ($n = 5$) and HU-treated (days 4–7, $n = 6$) MC-38 tumor-bearing mice at day 15 after tumor challenge. **e**, Percentage (mean ± s.e.m.) of M8-specific CD8+ T cells (left) and Ki-67 expression (gMFI, right) in blood from untreated and HU-treated (days 4–7) MC-38 tumor-bearing mice ($n = 5$). **f**, Heat map of metabolic markers and Ki-67 in M8-specific CD8+ T cells from lymph nodes of untreated ($n = 5$) and HU-treated (days 4–7, $n = 6$) MC-38 tumor-bearing mice. Color scale reflects geometric mean expression; individual marker ranges are indicated. **g**–**i**, GLUT1 expression in human tumor biopsy samples. **g**, Clinical treatment scheme. **h**, Representative image illustrating multiplex immunofluorescence post-treatment samples (CD8, blue; CD3, red; keratin, green; GLUT1, yellow). **i**, Quantification of GLUT1+CD8+ T cells (as a percentage of total CD8+ T cells) in paired baseline (letrozole) tumor biopsy samples and post-ribociclib surgical resections, collected 3–12 days after the last ribociclib dose. Individual participants are connected by lines. Box plots show the median (center) and 25th–75th percentiles (box), whiskers extend to the minimum/maximum values, and all individual points are displayed ($n = 11$). **j**, Tumor volume (mean ± s.e.m.) over time in EG7 tumor-bearing mice receiving untreated or HU-pretreated OT-I cells, followed by vaccination with OVA SLP/CpG one day after transfer (untreated, $n = 8$; OT-I groups, $n = 11$). **k**, Kaplan–Meier survival plot of MC-38 tumor-bearing mice left untreated ($n = 18$), treated with HU (days 4–7; $n = 22$), treated with αPD-L1 (day 10/14/17; $n = 9$) or treated with a combination of HU and αPD-L1 ($n = 10$). **l**, Top, schematic of TC-1 tumor challenge with therapeutic vaccination and HU. Bottom, Kaplan–Meier survival plot of TC-1 tumor-bearing mice either untreated ($n = 6$) or treated with E7 SLP/CpG ($n = 8$), HU ($n = 7$) or both ($n = 7$ per group). Statistical analysis included a two-sided unpaired $t$-test (**c**–**e**), two-sided paired $t$-test (**h**), Kruskal–Wallis and Dunn's multiple-comparisons test (**j**) and log-rank Mantel-Cox test (**k** and **l**); $P$ values are shown on the graphs. TILs, tumor-infiltrating lymphocytes.

cells showed greater resilience to functional inhibition, highlighting FDFT1-driven metabolic priming as a mechanism supporting robust proliferation after transient cell cycle blockade.

While our study focused on CD8[+] T cells, it is important to consider that other immune cells and tumor cells may also undergo metabolic rewiring upon cell cycle blockade. However, a critical distinction is that CD8[+] T cells produce substantial amounts of IL-2 during arrest, a feature not shared by most other cell types, and this cytokine appears central to their enhanced proliferative response upon release. In this respect, Munitic et al.[43] demonstrated that discontinuous stimulation is similar to continuous stimulation in driving proliferation of naive CD4[+] T cells, involving rapid production of IL-2 by responsive G1 T cells upon restimulation. This suggests that CD8[+] T cells may uniquely exploit temporal arrest to stockpile nutrients, reprogram metabolism and prime for IL-2-driven expansion, whereas other cells may primarily experience growth restriction.

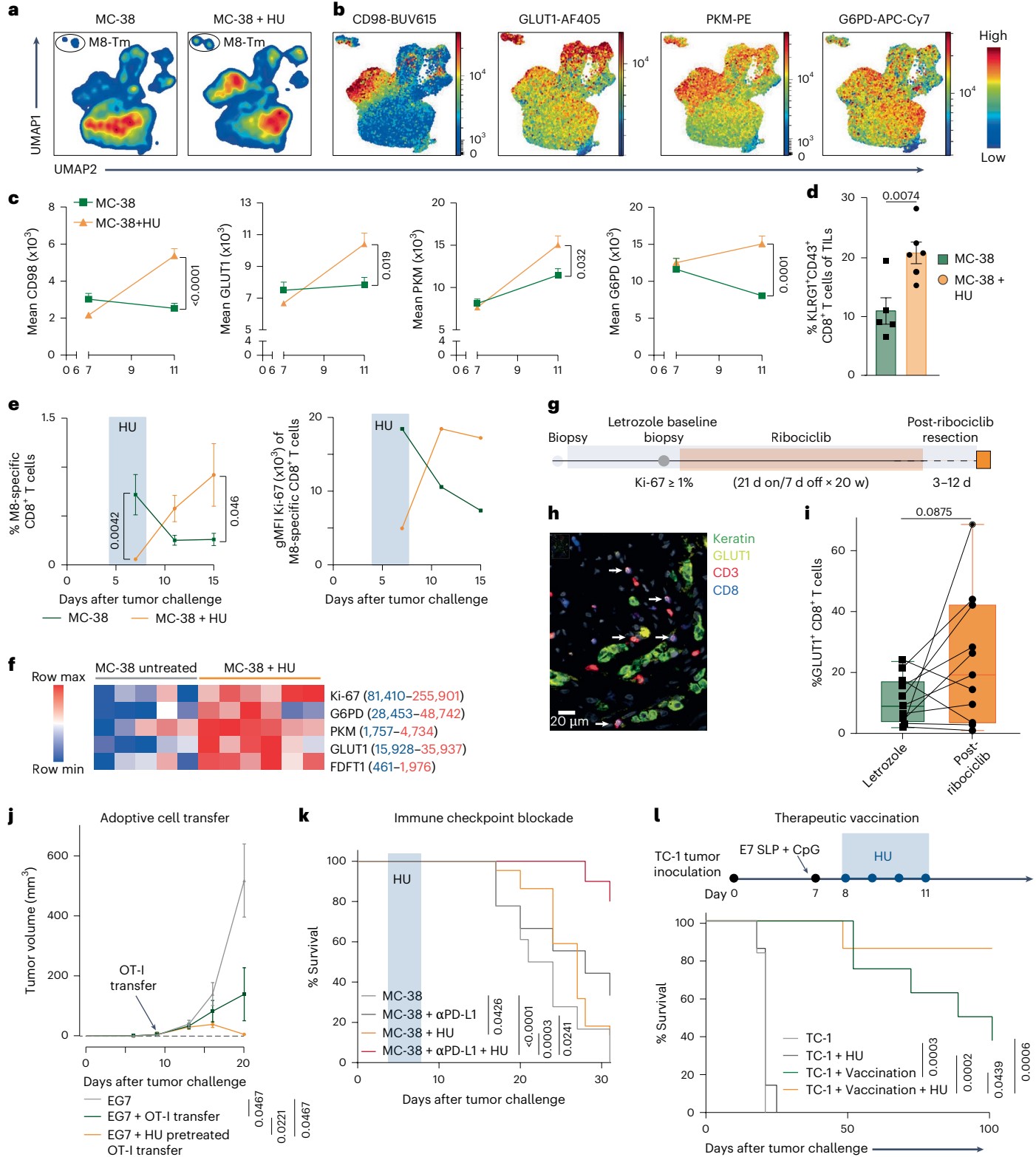

IL-2 signaling is linked to cell cycle progression in CD8[+] T cells. IL-2 promotes degradation of the CDK inhibitor p27[Kip1], leading to increased CDK2 activity and enabling progression through G1 phase. This relationship is bidirectional: IL-2 regulates CDK expression and activity[44,45], while CDKs in turn modulate IL-2 production[46,47]. Beyond its role in cell cycle control, IL-2 also fuels metabolic programming, for example, by inducing upregulation of GLUT1 expression to support glucose uptake[48]. This establishes a synergistic feedback loop whereby metabolically primed CD8[+] T cells upregulate IL-2 receptor expression, thereby increasing their sensitivity to IL-2 signaling. Together, these mechanisms highlight the intricate cross-talk between IL-2 signaling, cell cycle machinery and metabolic priming, which jointly may contribute to the enhanced proliferative capacity of CD8[+] T cells following transient cell cycle arrest. However, the dual role of IL-2 in supporting both effector T cells and regulatory T cells poses a potential limitation, as IL-2-driven regulatory T cell expansion could suppress antitumor immunity. Therapeutic strategies that selectively bias IL-2 signaling toward effector T cells, such as IL-2 muteins or IL-2–antibody complexes, may help overcome this hurdle and improve clinical outcomes[49].

The rapid expansion of effector CD8[+] T cells following transient cell cycle blockade is central to the improved tumor control observed in synergy with immune checkpoint blockade, ACT and cancer vaccines. Previous studies have shown that CDK4/CDK6 inhibitors can promote CD8[+] T cell immunity[50–54]. The work by Deng et al.[50] reported that CDK6 inhibition modulated NFAT activity and its downstream targets, while Heckler et al.[54] demonstrated a specific effect of CDK4/CDK6 inhibition on upregulation of MXD4 resulting in Myc inhibition and elevated memory formation. Additionally, CDK4/CDK6 inhibition was described to target retinoblastoma corepressor 1 (RB1)[51], highlighting multiple underlying mechanisms. Our findings demonstrate that the beneficial effects of transient cell cycle blockade extend beyond CDK4/CDK6 inhibition and are driven by a broader metabolic reprogramming of CD8[+] T cells during arrest.

These insights position temporal cell cycle inhibition, for example with HU or CDK4/CDK6 inhibitors, as a versatile strategy to enhance T cell function in cancer therapy. Moreover, the induction of metabolically primed CD8[+] T cells may serve as a predictive biomarker for responses to chemoimmunotherapy.

## Online content

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

**Floortje J. van Haften** [1,9], **Tetje C. van der Sluis** [1,2,9], **Hanna S. Hepp** [1], **Nils Mülling** [1], **Reza Nadafi**[1], **Bharath Sampadi**[3,4], **Suzanne van Duikeren** [1], **J. Shirin Mostert** [1], **Rosemarijn van der Sterre**[1], **Peter A. van Veelen** [4], **Graham A. Heieis**[5], **Dominique M. B. Veerkamp**[1], **Thomas H. Wesselink**[1], **Ward Vleeshouwers** [1], **Macha Beijnes**[1], **Iris N. Pardieck**[1], **Eralin L. F. van Horssen** [1], **Anne F. de Groot**[6], **Manon van der Ploeg**[7], **Judith R. Kroep**[6], **Noel F. C. C. de Miranda** [7], **Sabina Y. van der Zanden** [3], **Jacques Neefjes** [3], **Hailiang Mei** [8], **Alfred C. O. Vertegaal**[3], **Bart Everts** [5], **Sjoerd H. van der Burg** [2] **& Ramon Arens** [1]✉

¹Department of Immunology, Leiden University Medical Center, Leiden, the Netherlands. ²Department of Medical Oncology, Oncode Institute, Leiden University Medical Center, Leiden, the Netherlands. ³Department of Cell and Chemical Biology, Leiden University Medical Center, Leiden, the Netherlands. ⁴Center for Proteomics and Metabolomics, Leiden University Medical Center, Leiden, the Netherlands. ⁵Center for Infectious Diseases, Leiden University Medical Center, Leiden, the Netherlands. ⁶Department of Medical Oncology, Leiden University Medical Center, Leiden, the Netherlands. ⁷Department of Pathology, Leiden University Medical Center, Leiden, the Netherlands. ⁸Department of Biomedical Data Sciences, Sequencing Analysis Support Core, Leiden University Medical Center, Leiden, the Netherlands. ⁹These authors contributed equally: Floortje J. van Haften, Tetje C. van der Sluis. ✉e-mail: r.arens@lumc.nl

## Methods

### Ethics statement

The research outlined in this study complies with all relevant ethical regulations. Animal experiments were approved by the local and national committees under the permit numbers AVD116002015271, AVD11600202013796, AVD11600202417987 and AVD1160020186804, and performed according to the recommendations and guidelines set by the Leiden University Medical Centre (LUMC) and by the Dutch Act on Animal Experimentation and EU Directive 2010/63/EU. Maximal permitted tumor sizes were not exceeded. The NEOLBC trial (NCT03283384) was conducted in accordance with the Declaration of Helsinki and approved by the Medical Ethical Committee of the LUMC.

### Mice

C57BL/6J mice were obtained from Charles River (L'Arbresle, France) and Janvier labs (Le Genest-Saint-Isle, France), and maintained in the animal facility of LUMC. OT-I mice (B6.Cg-PtprcaTg(TcraTcrb)1100Mjb), IL-2$^{GFP}$-reporter mice (B6.Il2em1Lumc; generated by the Transgenic Facility Leiden; Supplementary Fig. 2), IL-2$^{fl/fl}$ Cre-ERT2 (ref. 55) and Raptor$^{fl/fl}$ (B6.Cg-Rptortm2.1LexTg(Itgax-cre)1-1Reiz/J mice crossed in house with C57BL/6-Tg(Cd8a-cre)1Itan) mice were bred and maintained in the LUMC animal facility. All mice were housed in individual ventilated cages under specific pathogen-free conditions, and were kept on 12-h light–dark cycles and ad libitum water and chow (SDS RM3; DS801203G10R), at 20–22 °C and 45–65% humidity. In all experiments, mice were used at 6–12 weeks of age.

### Ex vivo CD8$^+$ T cell assays

Spleens from naive mice were collected, red blood cell lysis was performed, and CD8$^+$ T cells were negatively enriched with magnetic beads according to the manufacturers' instructions (BD Bioscience or Miltenyi Biotec). For all assays with human CD8$^+$ T cells, buffy coats (Sanquin) from healthy donors were used. First, lymphocytes were isolated by using LeucoSep tubes (Greiner Bio-one) and Ficoll. From the lymphocyte fraction, CD8$^+$ T cells were negatively enriched with magnetic beads according to the instructions of the manufacturer (Miltenyi Biotec). Enriched CD8$^+$ T cells were labeled with either 5 μM CFSE (Invitrogen) or 5 μM CellTrace Violet (Thermo Fisher Scientific) and plated in a 96-well flat-bottom plate at a cell concentration of $1 \times 10^5$ cells per well. Mouse CD8$^+$ T cells were stimulated with 1 μg ml$^{-1}$ plate-bound anti-CD3 (BD Biosciences, clone 145-2C11) and 2 μg ml$^{-1}$ soluble anti-CD28 (BD Biosciences, clone 37.51). Human CD8$^+$ T cells were stimulated in a flat-bottom plate with 1 μg ml$^{-1}$ plate-bound anti-CD3 (eBioscience, clone OKT3) and 2 μg ml$^{-1}$ Ultra-LEAF anti-CD28 (BioLegend, clone CD28.2). To obtain IL-2 deficient CD8$^+$ T cells, IL-2$^{fl/fl}$ Cre-ERT2 mice mice were treated orally with 200 mg per kg body weight tamoxifen (Merck) dissolved in corn oil (Merck), for 5 consecutive days. Three days after the final treatment, animals were euthanized and spleens were obtained and processed further like spleens from naive mice. Both mouse and human cells were stimulated in IMDM (Gibco), which was supplemented with 10% FCS (Greiner), 100 IU ml$^{-1}$ penicillin–streptomycin (Gibco), 2 mM L-glutamine (Gibco) and 50 μM 2-mercaptoethanol at 37 °C with 5% CO$_2$. One hour after stimulation, cells were treated with different cell cycle inhibitors: 250 μM HU (Sigma-Aldrich), 5 μM RO-3306 (Sigma-Aldrich), 4 μM palbociclib (SelleckChem) or 16 μM ribociclib (BioVision). To remove the cell cycle inhibitors, CD8$^+$ T cells were washed using three centrifugation steps in conjunction with replenishing the medium. All inhibitors were titrated to fully inhibit the cell cycle progression of mouse and human CD8$^+$ T cells after activation. Pictures were obtained using an EVOS FL Auto Imaging system (Thermo Fisher).

IL-2-neutralizing antibodies, 25 μg ml$^{-1}$ S4B6 (BioXCell) and 25 μg ml$^{-1}$ JES6.1 (BioXCell) were directly added after activation in non-arrested conditions or after removal of the cell cycle inhibitor in released conditions. To prevent glucose uptake, cells were incubated with 40 μm of the GLUT1 inhibitor WZB117 (MedChemExpress) dissolved in dimethylsulfoxide. To inhibit glycolysis, 1 mM 2-deoxyglucose (Sigma-Aldrich) was used. Glycogenolysis was inhibited by blocking glycogen phosphorylase with different concentrations (50 μM, 100 μM) of CP91149 (SelleckChem). Cholesterol biosynthesis was inhibited by blocking HMG-CoA reductase with 10 μM atorvastatin calcium (Sigma-Aldrich) or by blocking squalene synthase with 40 μM zaragozic acid A (Santa Cruz Biotechnology). To inhibit mTOR signaling, 250 nM rapamycin (Calbiochem) was used. MYC signaling was inhibited by 50 μM 10058-F4 (Sigma-Aldrich). STAT5 was inhibited by different concentrations of STAT5-IN-1 (MedChemExpress). Glycogen was quantified in $1 \times 10^6$ ex vivo-stimulated human CD8$^+$ T cells per condition using the glycogen assay kit (Abcam), following the manufacturer's instructions. Cells were heated for 10 min at 95 °C, and optical density was measured at 570 nm using a SpectraMax i3x reader (Molecular Devices). Inhibitors were directly added after activation in non-arrested conditions, during cell cycle arrest and during release or only after removal of the cell cycle inhibitor in released conditions as indicated in figures and/or legends.

### Fractionation and immunoblot

Ex vivo-stimulated mouse CD8$^+$ T cells were washed in PBS and lysed in buffer (50 mM Tris-Hcl pH 8.0, 150 mM NaCl, 5 mM MgCl$_2$, 0.5% NP40, 2.5% glycerol, protease inhibitors, 10 mM $N$-ethylmaleimide and 10 μM MG132). Lysates were vortexed, incubated on ice for 10 min and centrifuged (10 min, 15,000$g$, 4 °C) to separate cytosolic (supernatant) and nuclear (pellet) fractions. Fractions were washed and mixed with SDS sample buffer (2% SDS, 10% glycerol, 5% β-mercaptoethanol, 60 mM Tris-HCl (pH 6.8) and 0.01% bromophenol blue) before SDS–PAGE and immunoblotting. Primary antibodies used were Foxk1 (1:1,000 dilution; CST, 12025S) and H3K9me3 (1:1,000 dilution; Abcam, ab8898). The secondary antibody used was horseradish peroxidase (HRP)-conjugated anti-rabbit IgG (1:5,000 dilution; Invitrogen, G21234). Blots were imaged on an AMERSHAM Imager 600 and quantified using ImageJ.

### RNA sequencing

Splenic mouse CD8$^+$ T cells were isolated from ten mice and stimulated ex vivo with or without HU with three technical replicates per condition. For each replicate, live CD8$^+$ T cells were subjected to fluorescence-activated cell sorting separately (BD FACSAria), and RNA was isolated using NucleoSpin XS columns (Macherey-Nagel). In the released and non-arrested states, we gated on proliferating cells, using CFSE. RNA concentrations were measured by using the Qubit 4 Fluorometer and the Qubit RNA HS Assay kit. RNA quality was determined with the Agilent 2100 Bioanalyzer by using the RNA 6000 Nano Kit.

Differentially expressed genes were selected ($q < 0.05$) and used for pathway analysis with IPA (version 20.0). Differential expression analysis and GSEA was performed in Qlucore Omics Explorer (version 3.7), using the trimmed mean of log expression ratios method. For GSEA, eight hallmark mouse gene sets were manually selected from the MSigDB database (v2022).

For comparison of the arrested/released CD8$^+$ T cells with the reactivation program of memory CD8$^+$ T cells, the datasets from Low et al.[23] are used (Gene Expression Omnibus, accession number GSE147908).

### Extraction and analysis of polar metabolites

Human CD8$^+$ T cells were enriched and activated in the presence or absence of 250 μM HU according to the ex vivo experimental setup. Cells were centrifuged, washed with 75 mM ammonium carbonate and extracted with 70% ethanol preheated to 70 °C. After centrifugation (-15,400$g$, 10 min, 4 °C), supernatants were collected for metabolite analysis. Total ion count was calculated as the sum of all ion counts per sample.

Polar metabolites were analyzed using General Metabolics' high-throughput, non-targeted metabolomics platform in negative ion

mode. Analyses were performed on an Agilent 1260 Infinity II LC pump coupled to a Gerstel MPS autosampler (CTC Analytics) and an Agilent 6550 Series Quadrupole TOF mass spectrometer (Agilent) equipped with a Dual AJS ESI source operating in negative mode, as described previously[56]. The mobile phase consisted of isopropanol:water (60:40, vol/vol) with 1 mM ammonium fluoride at a flow rate of 150 µl min$^{-1}$. For online mass-axis correction, two ions from the Agilent ESI-L Low Concentration Tuning Mix (G1969-85000) were used. Mass spectra were acquired in profile mode from 50–1,050 $m/z$ with a frequency of 1.4 s for 2 × 0.48 min (double injection) at the highest resolving power (4 GHz, HiRes).

## Phosphoproteomics

Samples were prepared using a previously published method[57,58]. Briefly, samples were lysed in a single pot lysis/alkylation/reduction buffer (6 M guanidine chloride (Gdmcl), 100 mM Tris (pH 8.5), 10 mM tris(2-carboxyethyl)phosphine (TCEP), 40 mM 2-chloroacetamide (CAA) and 2 µl benzonase (250 U µl$^{-1}$) per 10 ml of lysis buffer) and transferred to a 96-well plate. Lysates were stored at −80 °C until use. Reduction/alkylation was conducted by heating the samples to 95 °C for 5 min. Proteins were precipitated with acetone, and proteolytic digestion of 500 µg protein was performed in TFE-digestion buffer by adding LysC/trypsin mix (1:25 enzyme-protein ratio) to each well at 37 °C overnight. After phosphopeptide enrichment using titanium dioxide beads and cleanup, samples were dried by vacuum centrifugation. Phosphopeptide samples were resuspended in 10 µl 0.1% formic acid.

## Data-dependent acquisition mass spectrometry

Phosphopeptides were separated on a homemade analytical nano-HPLC column (50 cm × 75 µm, Reprosil-Pur C18-AQ 1.9 µm, 120 Å, Dr. Maisch) using an Ultimate 3000 nano-HPLC system (Thermo) coupled to an Exploris 480 mass spectrometer (Thermo). Peptides were loaded onto a precolumn (300 µm × 5 mm, C18 PepMap, 5 µm, 100 Å) and eluted on the analytical column with a 2–30% gradient of buffer B (80% acetonitrile, 0.1% formic acid) over 240 min at 250 nl min$^{-1}$; buffer A was 0.1% formic acid. The column tip (-10 µm) served as the electrospray source. Mass spectrometry data were acquired in data-dependent Top20 mode with HCD (30%), MS1 resolution of 120,000 (400–1,500 $m/z$, 50-ms fill) and MS2 resolution of 60,000 (fill 110 ms, first mass 120 Da, quadrupole isolation 1.2 Da). Precursors with charge 2–5 were selected, dynamically excluded for 45 s (20 ppm). FAIMS was applied at −45 V, −60 V and −75 V in standard resolution mode, and a lock mass at 445.12003 $m/z$ was used.

## ELISA

To measure IL-2 concentrations in the supernatant of ex vivo-stimulated mouse CD8$^+$ T cells, an ELISA was performed. MaxiSorp 96-well flat-bottom ELISA plates (Nunc) were coated with 1 µg ml$^{-1}$ of purified anti-mouse IL-2 (clone JES6-1A12). Supernatant was loaded onto the coated plates for 2 h at 37 °C. Next, biotinylated anti-mouse IL-2 (clone XMG1.2) was incubated for 1 h at room temperature. After incubation with Streptavidin-HRP antibody (BioLegend), TMB (Sigma-Aldrich) was used to detect HRP activity. Absorbance was measured at 450 nm with an iMark microplate reader (Bio-Rad).

## Cell cycle inhibition in vivo

To measure endogenous CD8$^+$ T cell responses, mice were subcutaneously vaccinated in the flanks with 75 µg synthetic long HPV16 E7$_{43-63}$ (GQAEPDRAHYNIVTFCCKCDS, produced in LUMC) and 20 µg CpG (InvivoGen). Vaccinated mice were treated intraperitoneally (i.p.) twice daily with 100 mg per kg body weight HU for 4 consecutive days, or daily with 2 mg per kg body weight topotecan (Accord), or daily with 150 mg per kg body weight palbociclib (MedChemExpress). To prevent glucose uptake, mice were once i.p. treated with 10 mg

per kg body weight of WZB117 (MedChemExpress) dissolved in 10% dimethylsulfoxide/90% corn oil (MedChemExpress) or with a vehicle control, one day after vaccination or one day after HU treatment was finished. HPV E7 responses were measured in the blood at different time points by using E7$_{49-57}$ tetramers (RAHYNIVTF). Mice were euthanized and spleens were collected, red blood cell lysis was performed, and cells were stained with different antibodies for further analysis by flow cytometry.

## Tumor models

Tumor cell lines MC-38 and E.G7-OVA were cultured in IMDM (Gibco) supplemented with 10% FCS (Greiner), 100 IU ml$^{-1}$ penicillin–streptomycin (Gibco) and 2 mM L-glutamine (Gibco). E.G7-OVA cultures were maintained with 400 µg ml$^{-1}$ Geneticin (G418; Life Technologies). The HPV6 E6/E7-expressing TC-1 cell line (from T. C. Wu) was cultured in IMDM supplemented with MEM non-essential amino acids (Life Technologies), 1 mM sodium pyruvate (Life Technologies) and 400 µg ml$^{-1}$ Geneticin.

For adoptive cell transfer, mice were inoculated subcutaneously in the right flanks with 1 × 10$^6$ E.G7-OVA tumor cells. Nine days after tumor inoculation, cohorts of E.G7-OVA tumor-bearing mice received ex vivo HU-pretreated and untreated OT-I cells (5 × 10$^4$) retro-orbitally, and were vaccinated with 150 µg synthetic long ovalbumin peptide (SMLVLLPDEVSGLEQLESIINFEKLTEWTS) and 20 µg CpG one day after transfer.

Mice were inoculated subcutaneously in the right flanks with 3 × 10$^5$ MC-38. Beginning 4 days later, mice received HU (100 mg per kg body weight, i.p.) twice daily for 4 consecutive days. Anti-PD-L1 (150 µg, clone MIH-5) was administered i.p. on days 10, 14 and 17. For CD8$^+$ T cell depletion, 150 µg anti-CD8 (clone 2.43) was given on day 3, and depletion was confirmed on day 4 by flow cytometry; maintenance dosing (50 µg) was performed twice weekly. Tumor growth was monitored, or mice were euthanized at defined time points for CD8$^+$ T cell analysis. Before tissue collection, mice were perfused with PBS supplemented with 2 mM EDTA, after which non-tumor-draining lymph nodes and tumors were isolated. Tumors were digested with Collagenase D (1 mg ml$^{-1}$) and DNase I (20 µg ml$^{-1}$) for 30 min at 37 °C, and single-cell suspensions were stained for flow cytometry. MC-38-specific CD8$^+$ T cells were detected using gp70/p15E major histocompatibility complex class I tetramers (KSPWFTTL).

For therapeutic vaccination, mice were inoculated subcutaneously in the right flanks with 1 × 10$^5$ TC-1 tumor cells. Seven days later, when tumors were palpable, mice were vaccinated in the left flanks with 75 µg HPV16 E7$_{43-63}$ synthetic long peptide and 20 µg CpG. HU (100 mg per kg body weight, i.p.) was administered twice daily for 4 days starting one day after vaccination. Tumors were measured at least twice weekly with calipers, and volumes were calculated as length × width × height × 0.52. Tumor growth curves and survival analyses were generated in GraphPad Prism.

## Flow cytometry

Samples were incubated with TruStain FcX PLUS antibody (anti-mouse CD16/32; BioLegend) and with protein kinase inhibitor dasatinib (Sigma-Aldrich) to block nonspecific binding to Fc receptors and T cell antigen receptor downregulation, respectively. For cell surface and viability staining, cells were resuspended in staining buffer (PBS supplemented with 1% FCS) and incubated for 30 min at 4 °C with antibodies. Metabolic assays were performed after viability staining, but before cell surface staining. To determine glucose (2-NBDG) and long-chain fatty acid (BODIPY FL-C16) uptake and mitochondrial membrane potential (TMRM), mitochondrial mass (MitoTracker Deep Red) and mitochondrial-derived superoxide (MitoSox Red), cells were stained for 15 min at 37 °C in IMDM. All antibodies for metabolic enzymes and transporters, except CD98, were first conjugated to fluorescent-labeled proteins according to the manufacturer's instructions (Abcam).

Following cell surface staining, cells were permeabilized and fixed for 45 min with the eBioscience Foxp3/Transcription factor staining buffer set (Invitrogen) and incubated with antibodies for metabolic enzymes and transporters. To measure intracellular cytokines, cells were incubated for 6 h with 5 µg ml⁻¹ brefeldin A (BD Pharmingen). Following cell surface staining, cells were fixed, permeabilized (according to instructions of Fix/Perm Kit of BD Biosciences) and subsequently stained intracellularly for expression of IL-2, IFNγ and TNF. For intracellular granzyme B staining, cell surface staining was performed, and cells were fixed and permeabilized with the eBioscience Foxp3/Transcription factor staining buffer set (Invitrogen). To perform phospo-protein staining, surface-stained cells were fixed with BD cytofix fixation buffer (BD Biosciences), followed by permeabilization with BD Phosflow Perm Buffer III (BD Biosciences) and pSTAT5 or pS6 antibody staining. For intracellular transcription factor staining, the True-Nuclear Transcription Factor Buffer Set (BioLegend) was used. For the analysis of the cell cycle phases, the Click-iT Plus EdU Pacific Blue Flow Cytometry Assay Kit (Thermo Fisher) was used in combination with FxCycle Violet Stain (Thermo Fisher) according to the manufacturer's protocol. To assess PKM expression across different cell cycle phases, PKM staining was incorporated in the intracellular staining step. For the assessment of DNA damage, surface-stained cells were fixed with 4% ultrapure paraformaldehyde (Polysciences) for 15 min, permeabilized with ice-cold 100% methanol (Supelco) for 20 min, and subsequently stained for γ-H2AX (phopho-Ser139) for 1 h.

Flow cytometry experiments were performed on the BD LSR-Fortessa (BD Biosciences) and three-laser or five-laser Cytek Aurora spectral analyzers (Cytek Biosciences). Data were analyzed with FlowJo (Tree Star, version 10) and OMIQ (https://www.omiq.ai/) analysis software.

### Clinical study

Clinical samples were obtained from participants with breast cancer enrolled in the NEOLBC trial[39]. From this randomized phase II trial, we included participants from the ribociclib plus letrozole arm. All participants in the NEOLBC study started with 2 weeks of letrozole followed by a tumor biopsy (letrozole baseline). Participants with ≥1% Ki-67 expression as scored on immunohistochemistry by central pathology review at the Department of Pathology of LUMC, were randomized between standard neoadjuvant chemotherapy or received neoadjuvant letrozole at 2.5 mg daily plus intermitted treatment with ribociclib (600 mg per day on days 1–21 of a 4-week cycle for 20 weeks for five cycles) followed by surgery. Surgical resection was performed 3–12 days after the last ribociclib dose. For the GLUT1 expression study, we analyzed paired tumor letrozole baseline biopsy samples and post-ribociclib surgical resections with a minimum of 25 CD8⁺ T cells per mm².

### Multispectral immunofluorescence imaging

Formalin-fixed paraffin-embedded tumor sections were stained with a multiplex immunohistochemistry protocol using Vectra (Akoya Biosciences) as described[59]. The panel included anti-pan-cytokeratin, anti-CD8α, anti-CD3ε and anti-Glut1 antibodies and DAPI. Each antibody was paired with a specific Opal fluorophore in an optimized staining sequence to maximize intensity and specificity. Image processing was performed using inForm (v2.4) and analyzed with QuPath (v0.3.1). T cells (CD3⁺) were classified as CD8⁺ or CD8⁻, and Glut1 expression was scored (Glut1⁺/Glut1⁻).

### Statistical analysis and experimental design

GraphPad Prism v10.2.3 was used for all statistical analyses. Blinding was not performed for the in vivo experiments; however, for selected in vitro assays, the experimenter acquiring the data was blinded to treatment, yielding results consistent with the non-blinded experiments. The researcher responsible for staining and analyzing clinical samples was blinded to sample identity. Mice were randomized before the start of each experiment, and in tumor studies they were additionally randomized based on tumor size. No animals or data points were excluded from the analyses. Sample sizes for in vivo experiments (4–12 mice per group) were determined using G*Power or power and sample size software and approved by the institutional statistician, providing 80% power at $\alpha = 0.05$. For clinical samples, normality was formally tested; for other datasets, distribution was assumed to be normal but not formally tested. Male and female animals were matched for age and sex, and cages were randomly assigned to treatment groups. Statistical parameters, including the exact $n$ (biological replicates and number of experiments) and the statistical tests used, are reported in the figure legends. Each dot represents an individual sample, and $P$ values are indicated in the figure panels. Data are representative of 2–3 independent experiments with similar results.

### Reporting summary

Further information on research design is available in the Nature Portfolio Reporting Summary linked to this article.

### Data availability

RNA-seq data have been deposited in the Gene Expression Omnibus under accession code GSE277143. Mass spectrometry proteomics data have been deposited in the ProteomeXchange Consortium via the PRIDE partner repository under dataset identifier PXD055517. All other data are available in the article and Supplementary Information. Source data are provided with this paper.

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

### Acknowledgements

We thank the Flow Cytometry Core Facility, Leiden Genome Technology Center, and the Central Animal and Transgenic Facility from LUMC for technical assistance. This work was supported by grants from the Dutch Cancer Society (KWF 11750 to R.A.), the Dutch Research Council (Gravitation Project Institute of Chemical Immunology 0021 to R.A.; Vici 724016003 to A.C.O.V.), LUMC (Gisela Thier Fellowship to T.C.S.) and the German Research Foundation (526085745 to N.M.).

### Author contributions

R.A. conceived and overall supervised the project. F.J.H., T.C.S., H.S.H., N.M., R.N., B.S., S.D., J.S.M., R.S., P.A.V., G.A.H., D.M.B.V., T.H.W., W.V., M.B., I.N.P., A.F.G., M.P. and S.Y.Z. performed experiments. F.J.H., T.C.S., H.S.H., N.M., R.N., B.S., S.D., E.L.F.H., A.F.G., M.P., H.M. and R.A. interpreted results and analyzed data. F.J.H., T.C.S. and R.A wrote the manuscript and prepared the figures. J.R.K., N.F.C.C.M., J.N., A.C.O.V.,

B.E. and S.H.B. provided scientific input and supervision. All authors reviewed and approved the manuscript.

## Competing interests

This study has been conducted by LUMC, which holds a patent on the use of T cell cycle synchronization to optimize antitumor responses (No2019548). T.C.S., S.H.B. and R.A. are named inventors on this patent. The other authors declare no competing interests.

## Additional information

**Extended data** is available for this paper at https://doi.org/10.1038/s41590-025-02407-0.

**Correspondence and requests for materials** should be addressed to Ramon Arens.

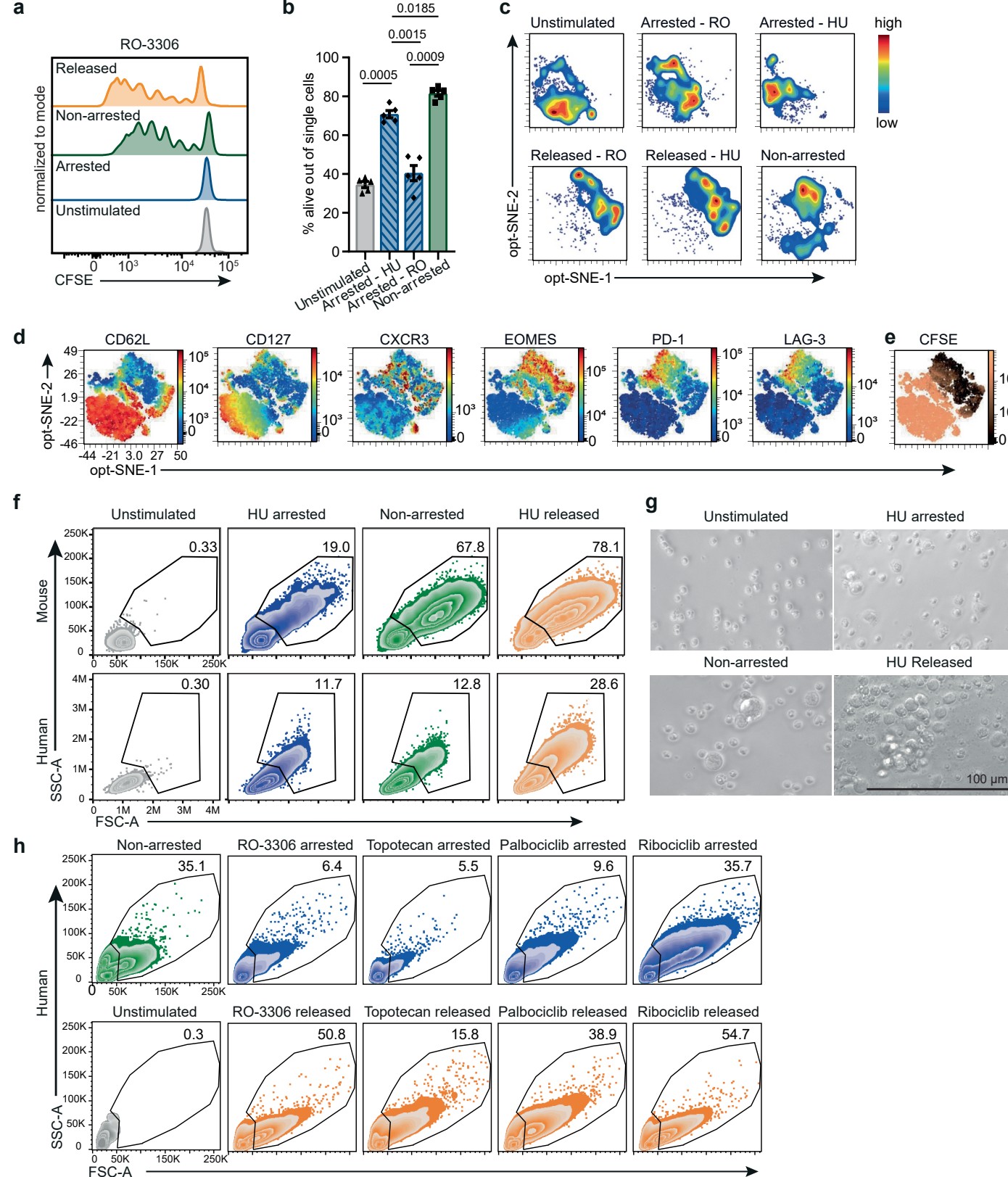

**Extended Data Fig. 1 | See next page for caption.**

**Extended Data Fig. 1 | Transient cell cycle inhibition enhances CD8⁺ T cell proliferation. a**, Representative proliferation plots of unstimulated mouse CD8⁺ T cells, or ex vivo-stimulated mouse CD8⁺ T cells that were either left untreated (non-arrested), arrested in the cell cycle using RO-3306, or released from RO-3306-induced arrest. **b**, Percentage of viable CD8⁺ T cells treated or not with HU and RO-3306, measured as Zombie-NIR⁻ cells within the singlet gate ($n=5$). Statistical comparisons were performed using RM ANOVA with Sidak's multiple comparisons; $P$ values are shown on the graphs. **c-e**, Mouse CD8⁺ T cells were left unstimulated or ex vivo-stimulated and either non-arrested, cell cycle-arrested with HU or RO-3306, or released from arrest ($n=5$). **c**, Contour plots showing the distribution of cells from each condition. **d**, Expression levels of the indicated markers overlaid on the opt-SNE map. **e**, CFSE intensity overlaid on the opt-SNE embedding, where black indicates CFSE^low (highly proliferating cells), and copper indicated CFSE^high (undivided/first cycle). **f**, Representative FSC/SSC plots of human and mouse CD8⁺ T cells that were unstimulated, cell cycle arrested with HU, non-arrested or released from arrest. Percentage of blasted cells (large, activated cells with increased FSC/SSC) is indicated. **g**, Representative microscopy images of ex vivo-stimulated mouse CD8⁺ T cells treated transiently with HU or left untreated (3 independent experiments with similar results). Representative FSC/SSC plots of human CD8⁺ T cells that were unstimulated, cell cycle arrested with RO-3306, topotecan, palbociclib or ribociclib, non-arrested or released from arrest. Percentage of blasted cells is indicated.

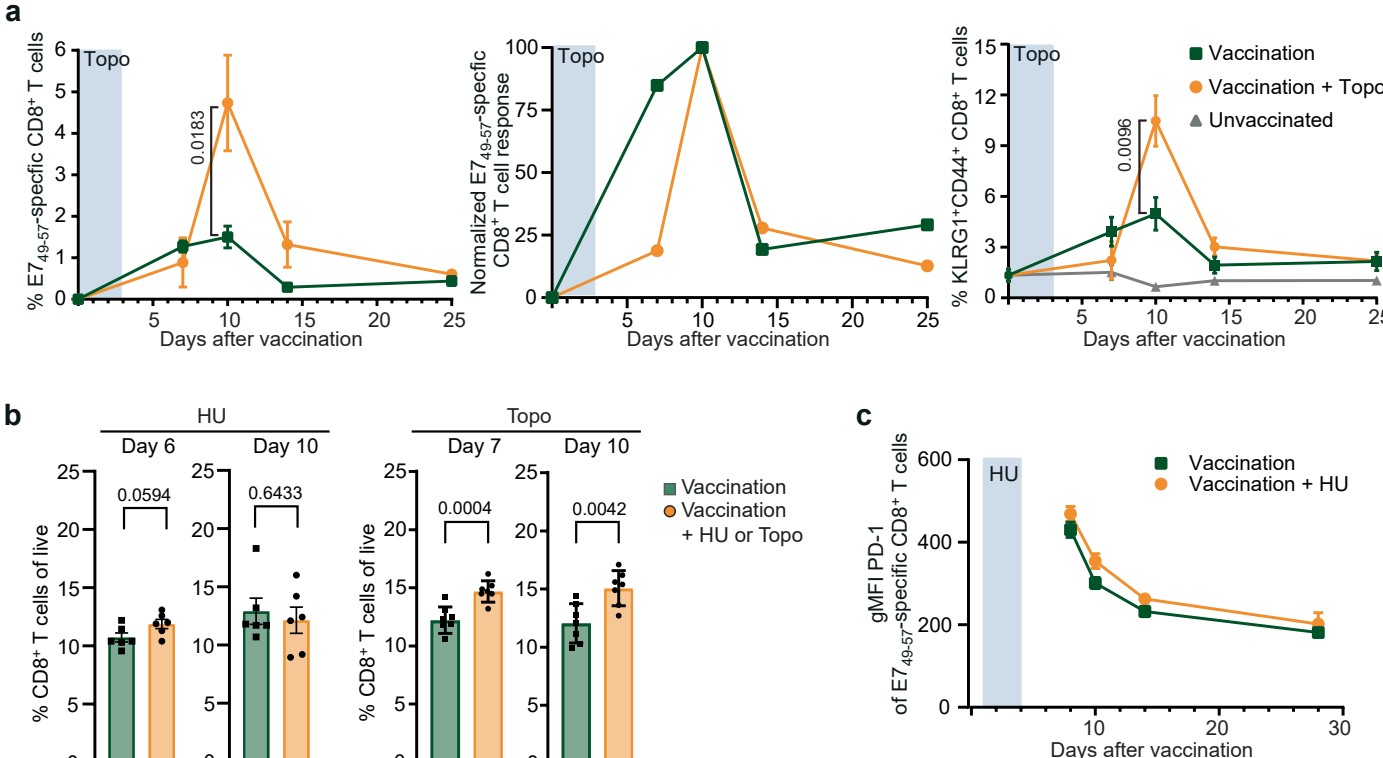

**Extended Data Fig. 2 | Transient cell cycle arrest enhanced CD8⁺ T cell proliferation *in vivo*. a**, Naïve mice were vaccinated with E7 SLP/CpG on day 0 and either left untreated or treated with topotecan (Topo) for three consecutive days (*n*=7). Left: Percentages (mean ± SEM) of circulating E7$_{49-57}$-specific CD8⁺ T cells over time. Middle: normalized values relative to the peak response. Right: Percentages (mean ± SEM) of KLRG1⁺CD44⁺ CD8⁺ T cells over time (*n*=7 per group). **b**, Percentage of total CD8⁺ T cells among live cells following E7 SLP/CpG

vaccination in mice either untreated or treated with HU (4 consecutive days, *n*=6) or Topo (3 consecutive days, *n*=7). **c**, Naïve mice were vaccinated with E7 SLP/CpG on day 0 and either left untreated (*n*=6) or treated with HU for 4 consecutive days (n=5). (PD-1 expression (geometric MFI ± SEM) on E7$_{49-57}$-specific CD8⁺ T cells was measured over time. Statistical comparisons were performed using two-sided unpaired *t*-test (**a**, **b**); *P* values are shown on the graphs.

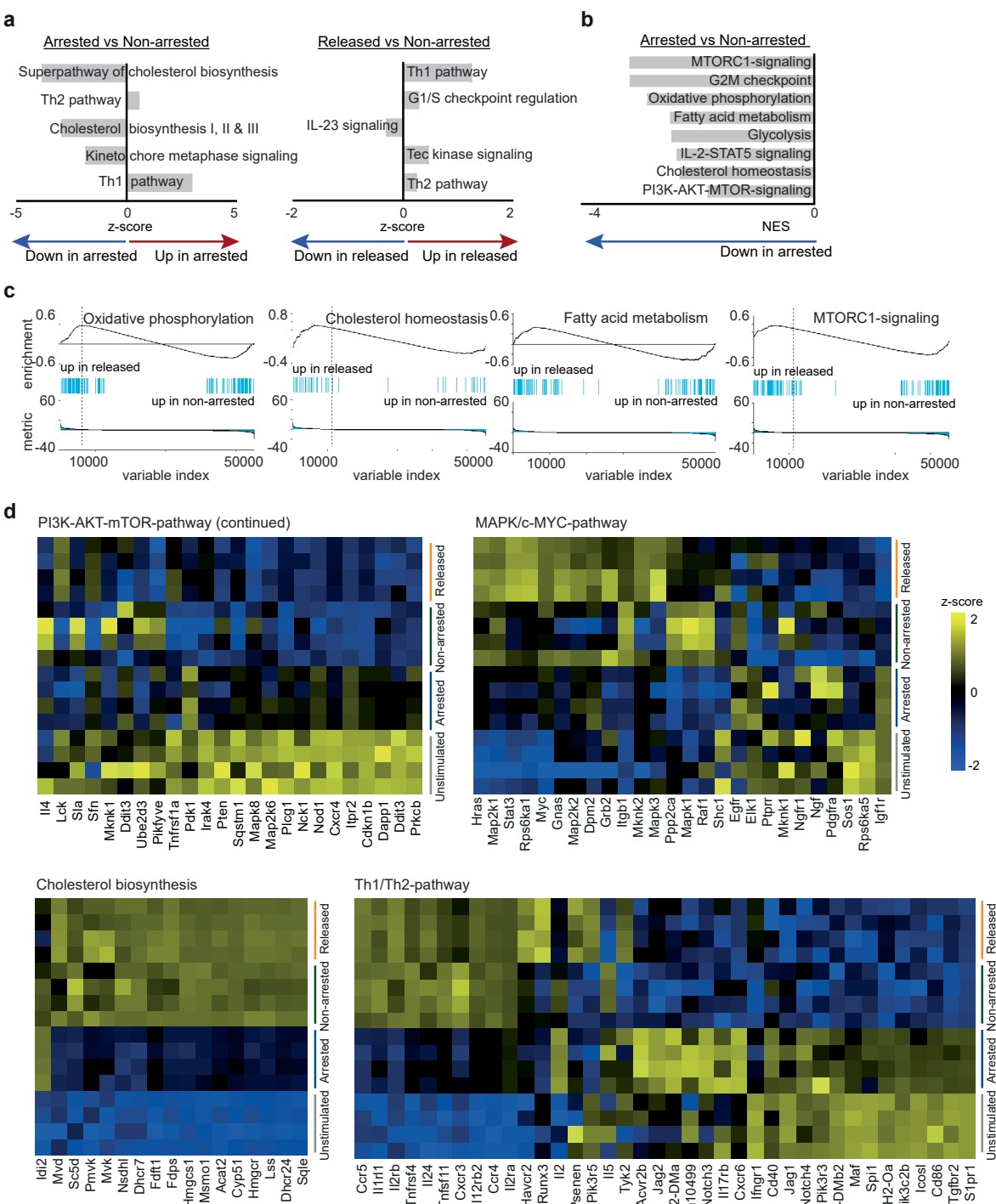

**Extended Data Fig. 3 | Transcriptional changes in CD8⁺ T cell metabolic pathways induced by cell cycle arrest.** Transcriptomic analysis of unstimulated mouse CD8⁺ T cells, or ex vivo–stimulated mouse CD8⁺ T cells that were either left untreated (non-arrested), arrested in the cell cycle using hydroxyurea (HU), or released from HU-induced arrest (released) ($n=4$). **a**, Ingenuity Pathway Analysis (IPA) of differentially expressed genes (FDR<0.05) comparing arrested or released cells to non-arrested controls. The top five significantly enriched pathways with the most positive and negative z-scores are shown. Statistical analysis was performed using two-sided unpaired t-test with Benjamini-Hochberg correction.

**b**, Normalized Enrichment Scores (NES) of eight selected mouse hallmark gene sets (from MSigDB; FDR<0.05) based on gene set enrichment analyses (GSEA). **c**, GSEA enrichment plots after GSEA analysis of the same hallmark gene sets. Genes are ranked on the x-axis by log₂(fold change) in expression between released and non-arrested cells. Vertical bars represent individual genes within each gene set; the enrichment score is shown on the y-axis. **d**, Heatmaps of differentially expressed genes within the PI3K-AKT-mTOR-pathway (continuation of Fig. 2e), the MAPK/c-MYC pathway, cholesterol biosynthesis, and the Th1/Th2 pathway. Each row represents an individual sample.

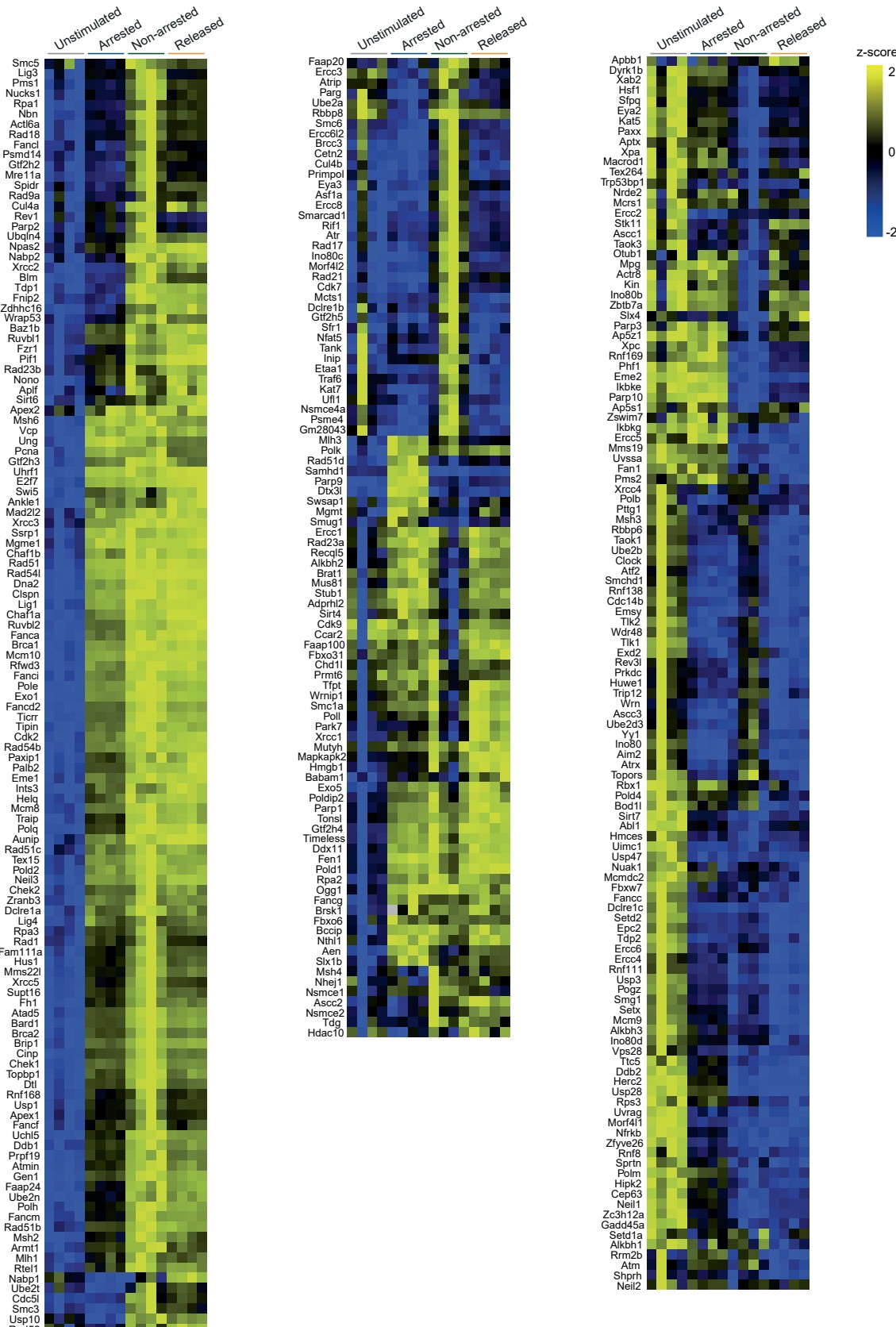

**Extended Data Fig. 4 | Transcriptomic profiling of DNA damage response and cell-cycle genes in CD8⁺ T cells during cell cycle arrest.** Transcriptomic analysis of unstimulated mouse CD8⁺ T cells, or ex vivo-stimulated mouse CD8⁺ T cells that were either left untreated (non-arrested), arrested in the cell cycle using hydroxyurea (HU), or released from HU-induced arrest (released) ($n$=4). Heatmaps display differentially expressed genes associated with the DNA damage response (DDR) pathway and cell cycle. Each column represents an individual sample.

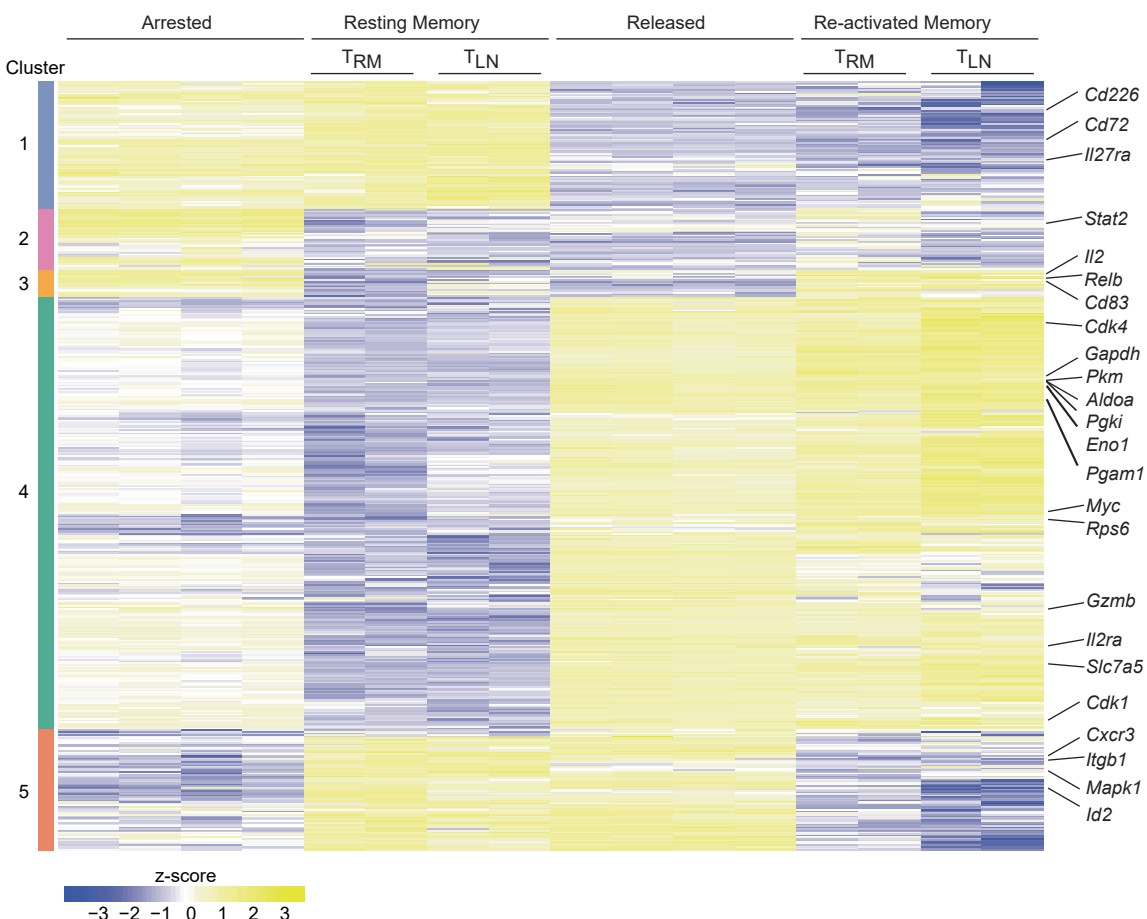

**Extended Data Fig. 5 | Transcriptomic comparison of transiently arrested CD8+ T cells and memory CD8+ T cell re-activation programs.** Heatmap of 1519 genes differentially expressed (FDR <0.05) across HU-arrested and released mouse CD8+ T cells, resting tissue-resident memory CD8+ T cells ($T_{RM}$), resting lymph node-derived memory CD8+ T cells ($T_{LN}$), and $T_{RM}$/$T_{LN}$ cells re-activated for 48 hours. Each row represents an individual gene; each column represents an individual sample. Heatmap values indicate expression z-scores based on $\log_2$ counts per million ($\log_2$CPM) as calculated using EdgeR. Clustering was performed using agglomerative hierarchical clustering with Euclidean distance.

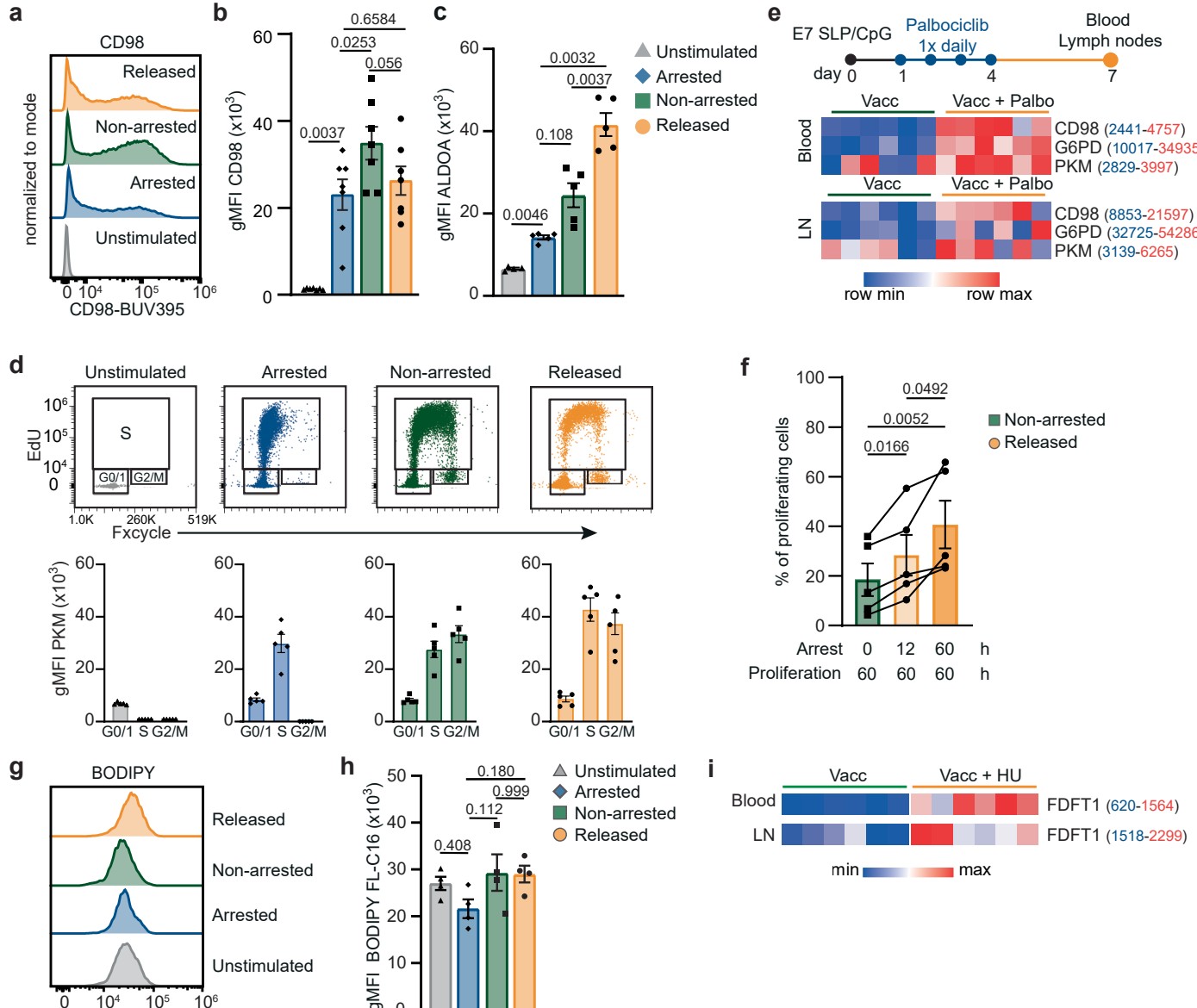

**Extended Data Fig. 6 | Temporal cell cycle arrest modulates glucose metabolism in CD8⁺ T cells. a**, Representative histograms of CD98 expression on unstimulated human CD8⁺ T cells, or ex vivo−stimulated human CD8⁺ T cells that were either left untreated (non-arrested), arrested in the cell cycle using hydroxyurea (HU), or released from HU-induced arrest (released). **b**, Geometric mean fluorescence intensity (gMFI ± SEM) of CD98 expression; each symbol represents a single healthy donor ($n=7$). **c**, ALDOA expression (gMFI ± SEM) in human CD8⁺ T cells under the same conditions as in (a) ($n=5$). **d**, Representative FXcycle *versus* Edu plots of unstimulated and ex vivo-stimulated human CD8⁺ T cells (HU arrested, non-arrested and HU released). Lower panels show gMFI of PKM expression in $G_{0/1}$, S and $G_2$/M cell cycle phases ($n=5$ donors). **e**, Heatmap showing expression of CD98, G6PD, and PKM in E7$_{49-57}$-specific CD8⁺ T cells in blood and lymph nodes (LN) on day 7 after vaccination with E7 peptide, in mice

treated with palbociclib for 4 consecutive days (days 1-4) ($n=6$ mice per group). Color scale represents geometric mean; marker-specific range is indicated for individual markers. **f**, Percentage (mean ± SEM) of proliferating human CD8⁺ T cells following ex vivo stimulation with or without HU treatment for 12h or 60h ($n=6$ donors). **g**, Representative histograms of BODIPY-FL-C16 uptake by unstimulated and ex vivo-stimulated human CD8⁺ T cells (HU arrested, non-arrested, and released after arrest). **h**, gMFI (± SEM) of BODIPY-FL-C16 uptake ($n=4$ donors). **i**, Heatmap showing relative FDFT1 expression in E7$_{49-57}$-specific CD8⁺ T cells from blood and lymph nodes (LN) of mice ($n=6$) vaccinated with E7 SLP/CpG and treated with HU or left untreated. Statistical comparisons were performed using repeated measures ANOVA with Sidak's multiple comparisons (**b, c, f, h**); $P$ values are shown on the graphs.

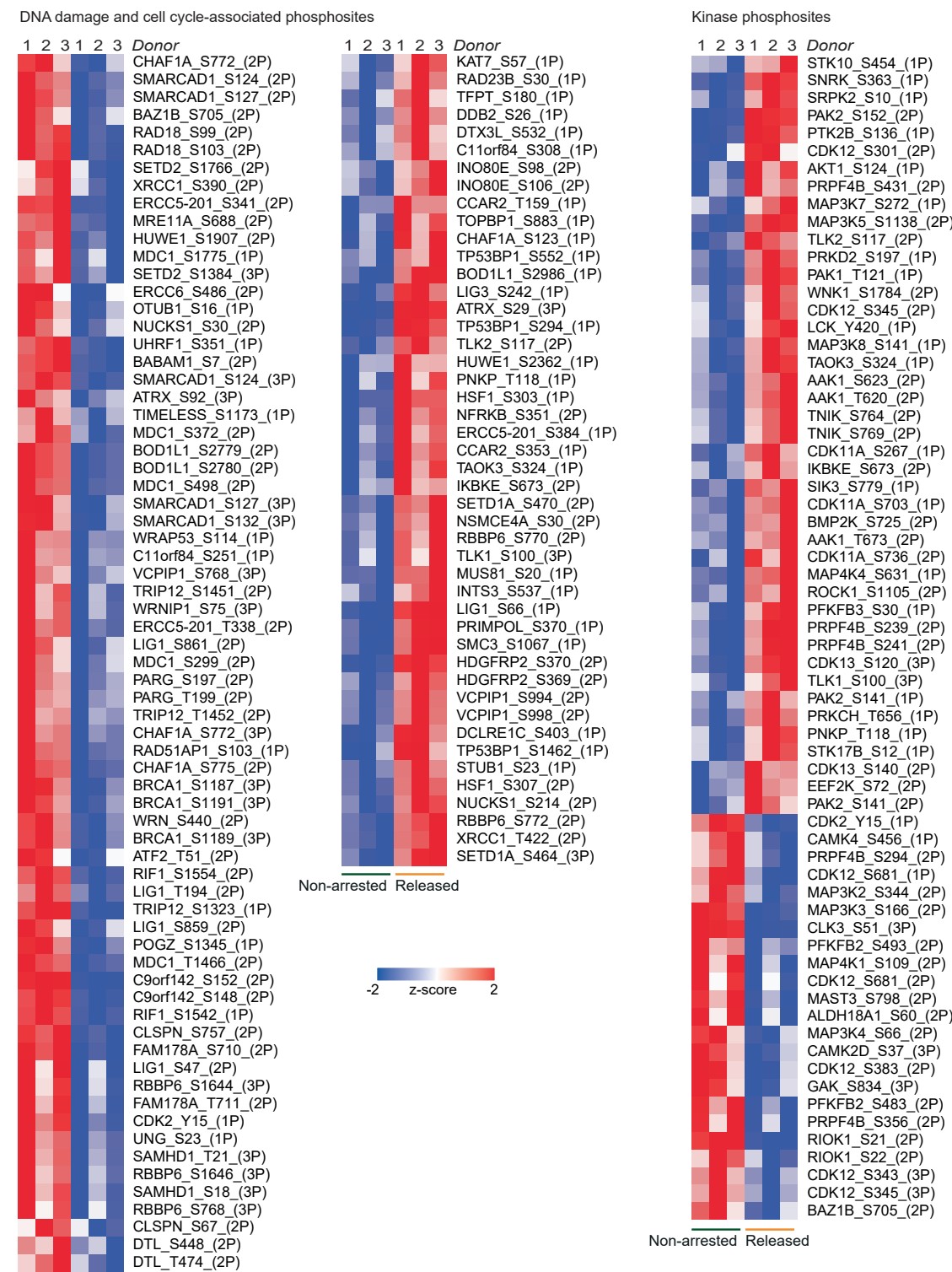

**Extended Data Fig. 7 | Phosphoproteomic analysis of DDR, cell cycle and kinase pathways in temporally arrested and non-arrested CD8⁺ T cells.**
**a**, **b**, Global heatmaps of z-scored phosphosite intensities, measured by mass spectrometry in ex vivo-stimulated human CD8⁺ T cells that were either released from HU-induced arrest or left untreated (non-arrested) (*n*=3 donors). Coupled z-scores are based on normalized phosphosite intensities. The number of phospho-groups per site is indicated in parentheses (3P indicates three or more phospho-groups). **a**, Phosphosites associated to the DNA damage response (DDR) and cell cycle pathways. **b**, Phosphosites related to kinases.

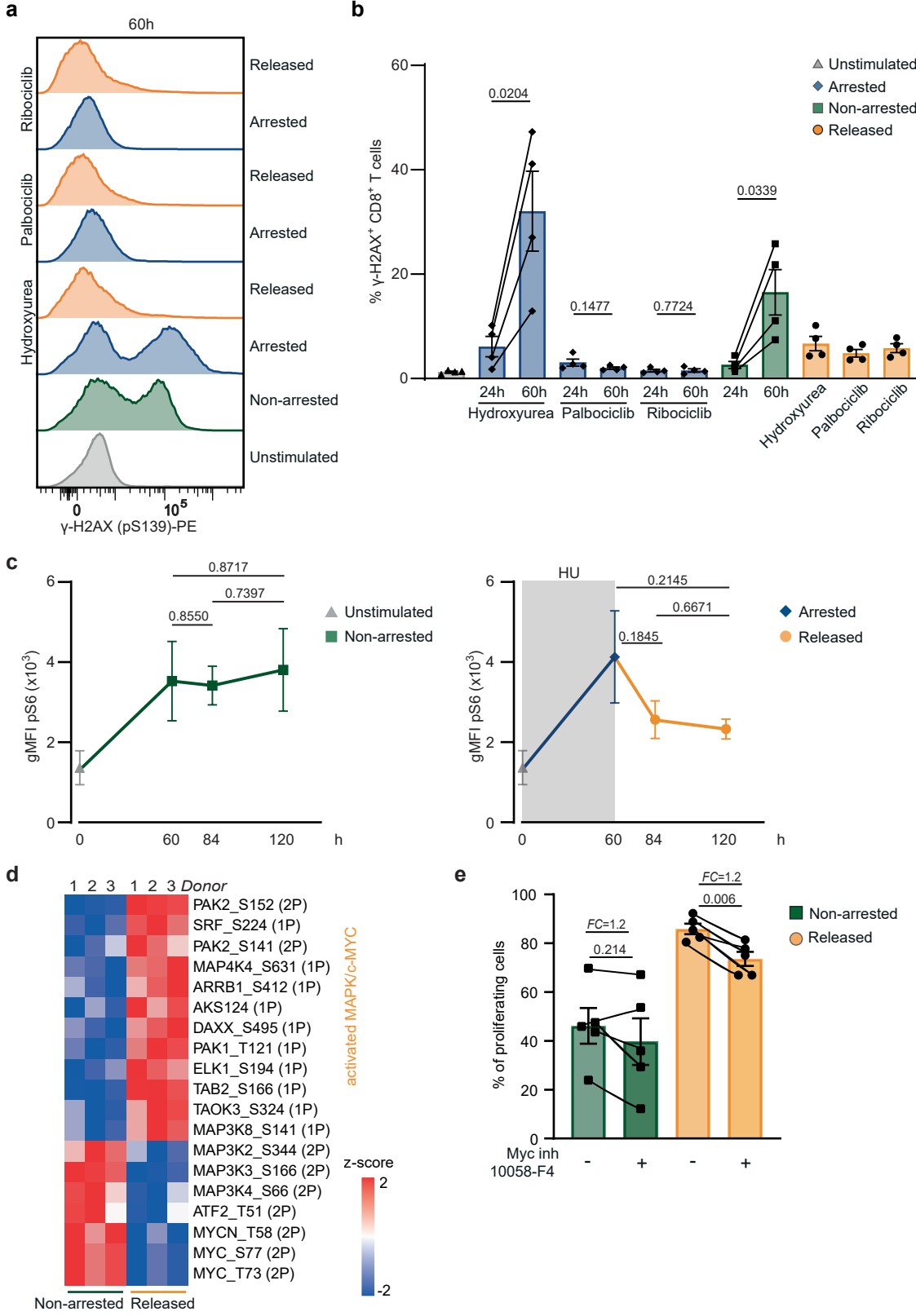

**Extended Data Fig. 8 | See next page for caption.**

**Extended Data Fig. 8 | DNA damage response and c-MYC signaling in CD8+ T cells following temporal cell cycle arrest. a**, Representative histograms of γ-H2AX expression in unstimulated human CD8+ T cells and in ex vivo–stimulated human CD8+ T cells that were either left untreated (non-arrested), cell cycle-arrested using hydroxyurea (HU), palbociclib or ribociclib, or released from arrest (released). **b**, Percentage of γ-H2AX+ CD8+ T cells (mean ± SEM) in unstimulated human CD8+ T cells, or following ex vivo stimulation with or without treatment (as described in a) for 24h or 60h. Each symbol represents an individual healthy donor (*n*=4). **c**, Kinetic analysis of pS6 levels in ex vivo-stimulated human CD8+ T cells that were either left untreated (non-arrested), arrested in the cell cycle using HU, or released from HU-induced arrest (*n*=5 donors). **d**, Heatmap of hierarchically clustered z-scored phosphosite intensities related to the MAPK/c-MYC signaling pathway, measured by mass spectrometry in ex vivo-stimulated human CD8+ T cells that were either released from HU-induced arrest or left untreated (non-arrested) (*n*=3 donors). Coupled z-scores are based on normalized phosphosite intensities. The number of phospho-groups per site is indicated in parentheses (3P indicates three or more phospho-groups). **e**, Ex vivo-stimulated mouse CD8+ T cells treated with HU and subsequently released, or left untreated (non-arrested), with or without the addition of the MYC inhibitor 10058-F4 during proliferation. Percentage of proliferating CD8+ T cells (mean ± SEM) is shown. Lines indicate individual mice (*n*=4). Statistical comparisons were performed using repeated measures ANOVA with Sidak's multiple comparisons (**c**) and two-sided paired t-test (**b**, **e**); *P* values are shown on the graphs.

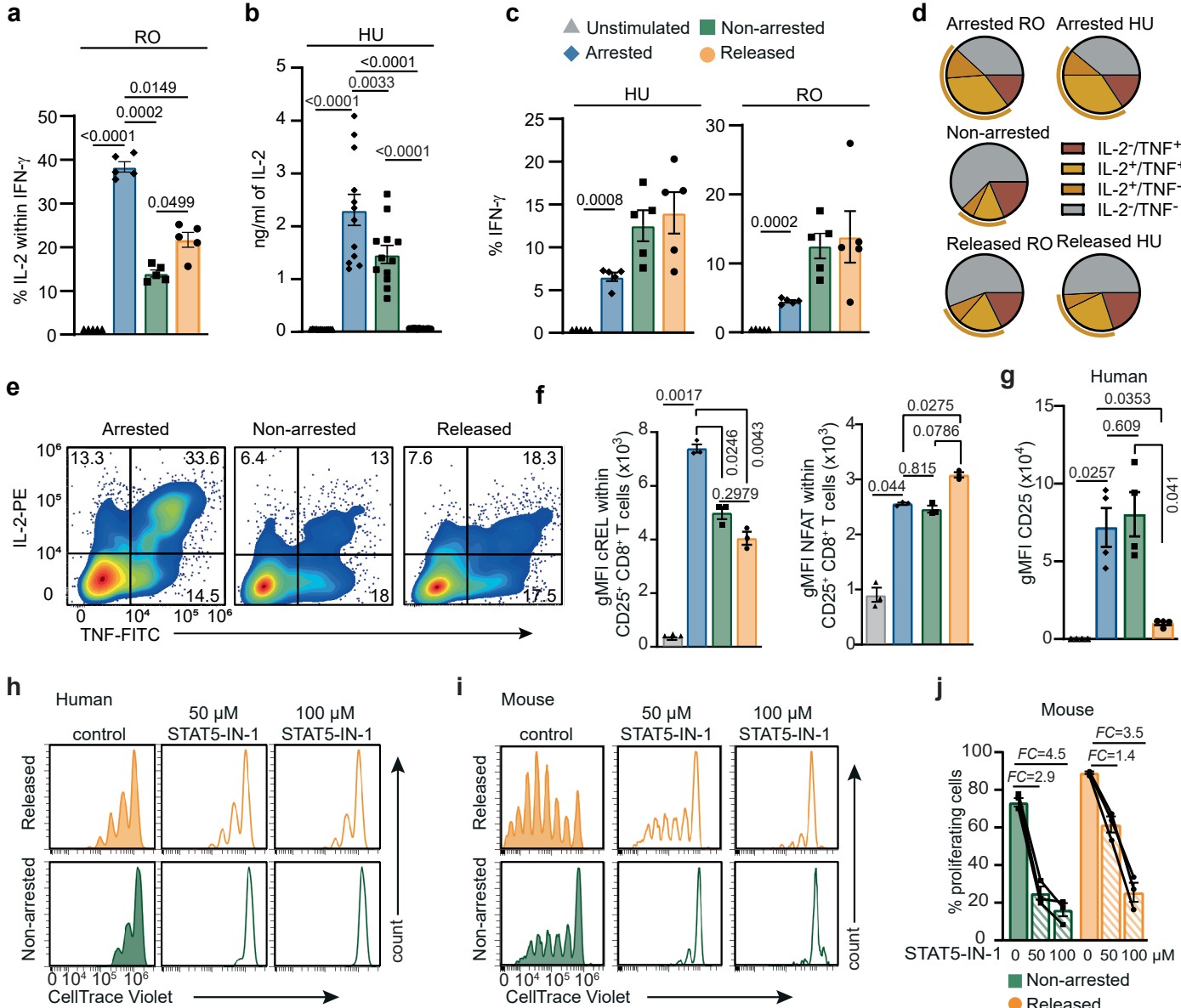

**Extended Data Fig. 9 | Increased IL-2 production and signaling contribute to enhanced proliferation in released CD8⁺ T cells. a**, Percentage of IL-2 producing cells (mean ± SEM) among unstimulated IFN-γ⁺ mouse CD8⁺ T cells, or ex vivo-stimulated IFN-γ⁺ mouse CD8⁺ T cells that were either left untreated (non-arrested), arrested in the cell cycle using RO-3306 (arrested), or released from RO-3306-induced arrest (n=5 mice). **b**, IL-2 concentration (mean ± SEM) in the supernatant of unstimulated human CD8⁺ T cells, or ex vivo–stimulated human CD8⁺ T cells that were either left untreated (non-arrested), arrested in the cell cycle using hydroxyurea (HU), or released from HU-induced arrest (released) (n=12 donors). **c**, Percentage of IFN-γ-producing mouse CD8⁺ T cells under the same conditions as in (a), with cell cycle inhibition by HU or RO-3306 (n=5 mice). **d**, Pie charts showing the average distribution of IL-2⁻/TNF⁺, IL-2⁺/TNF⁺, IL-2⁺/TNF⁻, and IL-2⁻/TNF⁻ subsets among IFN-γ⁺ CD8⁺ T cells following HU (top) or RO-3306 (bottom) treatment. The yellow line demarcates the total IL-2-producing fraction.

**e**, Contour plots showing expression patterns of IL-2 and TNF within the IFN-γ⁺CD8⁺ T cell population; merged data from 5 mice per condition. **f**, Expression of cREL (left) and NFAT (right) (gMFI ± SEM) within CD25⁺ mouse CD8⁺ T cells under unstimulated, and stimulated settings (HU-arrested, non-arrested and released from HU arrest) (n=3 mice). **g**, gMFI (± SEM) of CD25 expression in human CD8⁺ T cells that were unstimulated or ex vivo– stimulated and either non-arrested, HU-arrested or released (n=4 donors). **h, i**, Representative CellTrace Violet histograms (i,j) of ex vivo-stimulated human (h) and mouse (i) CD8⁺ T cells (non-arrested and HU-released), with or without addition of the STAT5 inhibitor STAT5-IN-1 supplemented during proliferation. **j**, Percentage (mean ± SEM) proliferating mouse CD8⁺ T cells (n=3 mice) under the same conditions as in (i). Fold-change (FC) is indicated. Statistical comparisons were performed using repeated measures ANOVA with Sidak's multiple comparisons (a-c, f, g); P values are shown on the graphs.

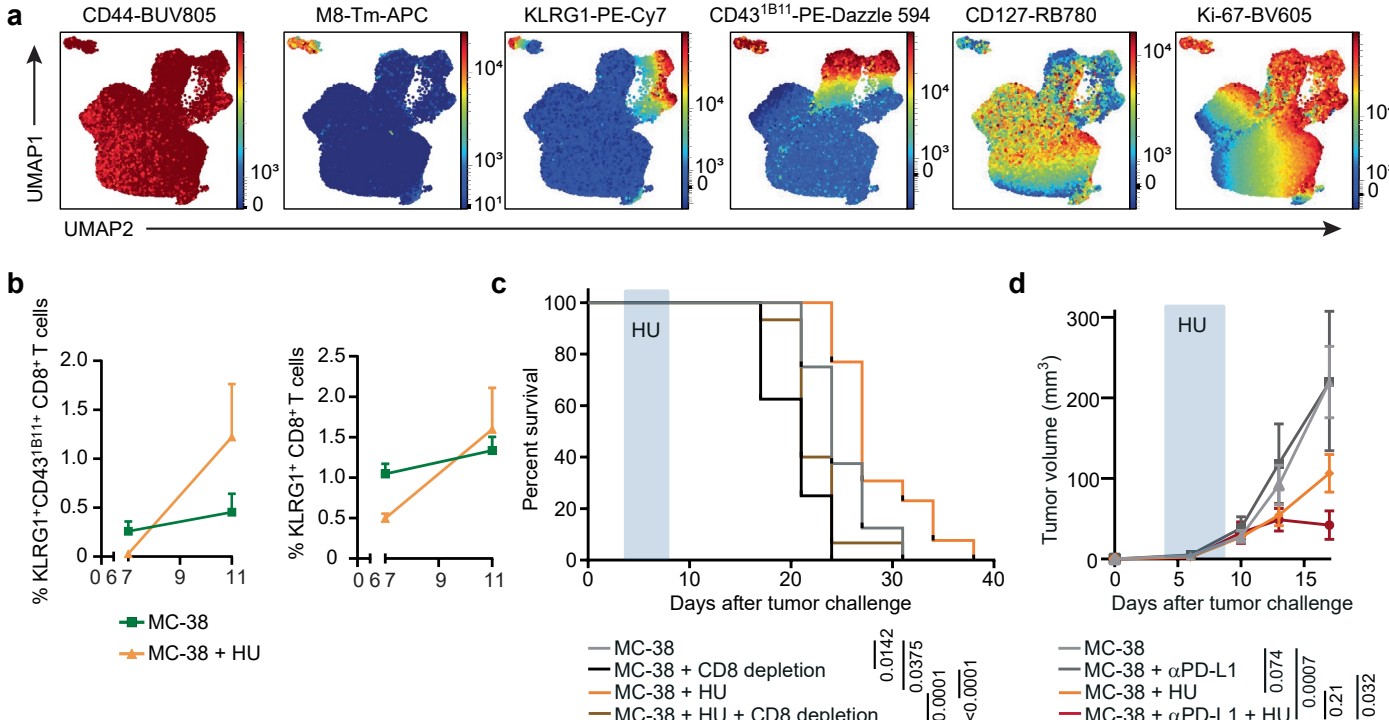

**Extended Data Fig. 10 | CD8⁺ T cells are required for tumor control following temporal cell cycle arrest. a**, UMAP plots of blood-derived CD44⁺CD8⁺ T cells from untreated (*n*=8) and HU treated (day 4-7, *n*=15) MC-38 tumor-bearing mice, analyzed on day 11 post-tumor challenge. Following down-sampling to 40.000 CD44⁺CD8⁺ T cells per group, UMAP embedding was performed. Expression of CD44, M8-tetramer KLRG1, CD43[1B11], CD127 and Ki-67 is overlaid on the UMAP (shown in Fig. 7b). **b**, Percentage (mean + SEM) of circulating KLRG1⁺CD43[1B11+] (left) and KLRG1⁺ (right) CD8⁺ T cells in blood from untreated (n=8) and HU-treated (day 4-7, *n*=15) MC-38 tumor bearing mice. **c**, Kaplan-Meier survival curves of MC-38-tumor bearing mice: untreated (*n*=8), CD8⁺ T cell-depleted (*n*=8), treated with HU (day 4-7, *n*=13), and HU-treated with concurrent CD8⁺ T cell depletion (*n*=15). Statistical comparisons for survival were performed with the log-rank Mantel-Cox test. **d**, Tumor volume over time (mean ± SEM) in MC-38 tumor-bearing mice: untreated (*n*=18), treated with HU (day 4-7, *n*=22), αPD-L1 treated (day 10/14/17, *n*=9), or the combination treatment (*n*=10). Statistical comparisons for tumor outgrowth were performed using the Kruskal-Wallis test followed by Dunn's multiple comparisons test; *P* values are shown on the graphs.

# Reporting Summary

## Statistics

For all statistical analyses, confirm that the following items are present in the figure legend, table legend, main text, or Methods section.

| n/a | Confirmed | |
|---|---|---|
| ☐ | ☒ | The exact sample size (*n*) for each experimental group/condition, given as a discrete number and unit of measurement |
| ☐ | ☒ | A statement on whether measurements were taken from distinct samples or whether the same sample was measured repeatedly |
| ☐ | ☒ | The statistical test(s) used AND whether they are one- or two-sided *Only common tests should be described solely by name; describe more complex techniques in the Methods section.* |
| ☒ | ☐ | A description of all covariates tested |
| ☐ | ☒ | A description of any assumptions or corrections, such as tests of normality and adjustment for multiple comparisons |
| ☐ | ☒ | A full description of the statistical parameters including central tendency (e.g. means) or other basic estimates (e.g. regression coefficient) AND variation (e.g. standard deviation) or associated estimates of uncertainty (e.g. confidence intervals) |
| ☐ | ☒ | For null hypothesis testing, the test statistic (e.g. *F*, *t*, *r*) with confidence intervals, effect sizes, degrees of freedom and *P* value noted *Give P values as exact values whenever suitable.* |
| ☒ | ☐ | For Bayesian analysis, information on the choice of priors and Markov chain Monte Carlo settings |
| ☒ | ☐ | For hierarchical and complex designs, identification of the appropriate level for tests and full reporting of outcomes |
| ☒ | ☐ | Estimates of effect sizes (e.g. Cohen's *d*, Pearson's *r*), indicating how they were calculated |

*Our web collection on statistics for biologists contains articles on many of the points above.*

## Software and code

Policy information about availability of computer code

| Data collection | BD FACSAria & BD Fortessa (BD Biosciences), Cytek Aurora (CYTEK), EVOS® FL Auto Imaging system (ThermoFisher), SpectraMax i3x Multi-Mode Microplate Reader (Molecular Devices), AMERSHAM Imager 600 (Bimedis), Novaseq 6000 (Illumina), General Metabolics' high-throughput non-targeted metabolomics platform (consisting of an Agilent 1260 Infinity II LC pump coupled to a Gerstel MPS autosampler (CTC Analytics, Zwingen, Switzerland) and an Agilent 6550 Series Quadrupole TOF mass spectrometer (Agilent, Santa Clara, CA, USA) with Dual AJS ESI source operating in negative mode), Exploris480 mass spectrometer (Thermo), iMarkTM microplate reader (Biorad), Vectra system (Akoya Biosciences. |
|---|---|
| Data analysis | FlowJo  (version 10), Morpheus software, OMIQ.ai, ImageJ, BIOWDL RNAseq pipeline v3.0.0, Ingenuity Pathway Analysis, Qlucore Omics Explorer (version 3.7), MATLAB, Graphpad Prism (version 10.2.3), QuPath (v0.3.1),RStudio (version 2024.04.2), R (version 4.4.1) and EdgeR (version 4.2.2). |

Flow cytometry
Samples were arcsinh-transformed with relevant cofactors and then pre-gated on FSC/SSC, singlet, live, CD8+/CD3+/CD44+, after which 1842 of these gated cells were subsampled. Dimension-reduction was conducted using the UMAP algorithm (15 neighbors, minimum distance of 0.4). UMAPs were computed using the following markers: Sca-1, CD98, Glut1, Ki-67, CD25, CD127, PKM, CD43, KLRG1, Tetramer, and G6PD. For immune profiling of ex vivo CD8+ T cells, opt-SNE analysis was performed using the OMIQ.ai platform. Samples were arcsinh-transformed with relevant cofactors and then pre-gated on FSC/SSC, singlet, live, CD8+/CD3+, after which 545 of these gated cells were subsampled. Dimension-reduction was conducted using the opt-SNE algorithm (1000 iterations, perplexity 30, Theta 0.5 and random seed 4644) and computed using the following markers: CXCR3, PD-1, CD62L, CD25, LAG-3, CD127, and EOMES. Morpheus software (MA, USA, https://software.broadinstitute.org/morpheus) was used to create heatmaps. Tumor immune filtrates containing more than 5% B cells after perfusion were excluded from analysis.

RNA sequencing

The extracted RNA was converted into cDNA by using the SuperScript® III First-Strand Synthesis kit (Invitrogen). To block cDNA synthesis of ribosomal RNA, the extracted RNA was treated with QIAseq® FastSelectTM -rRNA HMR Removal Kit (Qiagen). Second strand DNA synthesis was performed by using the Klenow Fragment exo-polymerase kit (Invitrogen). IDT adapters containing an 8bp UMI sequence and enrichment was performed for the 150-250 bp fragments by using the Kapa Hyperprep Kit. The quality and concentration of the fragmented and enriched library was verified with the Agilent 2100 Bioanalyzer by using the High Sensitivity DNA kit (Agilent Technologies). PCR products were purified by using AmpureXP Beads (Beckman CoulterTM). First, pooling of the samples was checked by shallow sequencing on the MiSeq and afterwards the samples were run on the Novaseq 6000 (Illumina).

RNAseq reads were processed using the opensource BIOWDL RNAseq pipeline v3.0.0 (https://zenodo.org/records/3713261) developed at the LUMC. This pipeline performs FASTQ preprocessing (including quality control, quality trimming, and adapter clipping), alignment, read quantification, and optionally transcript assembly. FastQC (v0.11.7) was used for checking raw read QC. Adapter clipping was performed using Cutadapt (v2.4) with the default settings. RNAseq reads' alignment was performed using STAR (v2.7.3a) on mouse reference genome GRCm38. umi_tools (v0.5.5) was used to remove duplicates identified by UMIs.

Differential expression analysis on the datasets of Low et al.,(re-activation program of memory CD8 T cells), and the arrested versus released CD8+ T cells is done using RStudio (version 2024.04.2), R (version 4.4.1) and EdgeR (version 4.2.2). In EdgeR. the normalization method Trimmed-mean-of-M values "TMM" is used. Differentially Expressed Gene(s) (DEGs) are filtered with an FDR of < 0.05. The overlapping differentially expressed genes between both datasets (1519 genes) were visualized in a heatmap after per-gene z-scaling based on log2CPM of counts. Clustering is based on agglomerative hierarchical clustering using Euclidean distance (unsupervised), with four clusters.

Polar Metabolites

Mass spectrometry data processing and analysis were performed in MATLAB (The MathWorks, Natick, MA, USA) using functions from the Bioinformatics, Statistics, Database, and Parallel Computing toolboxes. Peak picking was performed once per sample on the total profile spectrum generated by summing all scans over time, using wavelet decomposition from the Bioinformatics toolbox. Peaks below 5,000 ion counts in the summed spectrum were excluded to avoid low-abundance features unlikely to yield meaningful insights. Centroid lists from individual samples were merged into a single matrix by binning accurate masses within instrument-specific resolution tolerances.

A list of expected ions, including deprotonated, fluorinated, and major adduct forms, was generated from HMDB v4.0. All molecular formulas matching the measured masses within 0.001 Da were enumerated. Because this method does not include chromatographic separation or extensive $MS^2$ characterization, compounds with identical molecular formulas cannot be distinguished; thus, annotation confidence corresponds to Level 4. In practice, confidence for primary metabolic intermediates is higher due to their abundance in biological extracts. The resulting matrix contains the intensity of each mass peak in each sample, and a refined common m/z value was calculated using the weighted average of independently centroided values.

Phosphoproteomics data analysis and statistics

RAW files were searched against the Homo sapiens UniProt database (2023) using MaxQuant73 with default settings, adding Phospho(STY) as a variable modification and enabling FAIMS as appropriate. A maximum of four missed cleavages was allowed. Phosphosite intensities were imported into Perseus74, $log_2$-transformed, and filtered to remove contaminants and reverse hits. Sites quantified in at least one sample per condition were retained, and missing values were imputed from a downshifted normal distribution (width 0.3×σ, downshift 1.8×σ). Z-scores were calculated after mean centering to normalize technical variation while preserving donor-specific biological differences. Student's t-test was applied to non-arrested and HU-released conditions, and responsive phosphosites were defined by p < 0.05 and >1.5-fold change.

For manuscripts utilizing custom algorithms or software that are central to the research but not yet described in published literature, software must be made available to editors and reviewers. We strongly encourage code deposition in a community repository (e.g. GitHub). See the Nature Portfolio guidelines for submitting code & software for further information.

# Data

Policy information about availability of data

All manuscripts must include a data availability statement. This statement should provide the following information, where applicable:

- Accession codes, unique identifiers, or web links for publicly available datasets
- A description of any restrictions on data availability
- For clinical datasets or third party data, please ensure that the statement adheres to our policy

The mass spectrometry proteomics data have been deposited to the ProteomeXchange Consortium via the PRIDE partner repository with the dataset identifier PXD055517.
Reviewer account details:
 Username: reviewer_pxd055517@ebi.ac.uk
 Password: sXDvMdGjaUva

The RNAseq data have been deposited to the Gene Expression Omnibus (GEO).
To review GEO accession GSE277143:
Go to https://eur03.safelinks.protection.outlook.com/?url=https%3A%2F%2Fwww.ncbi.nlm.nih.gov%2Fgeo%2Fquery%2Facc.cgi%3Facc%3DGSE277143&data=05%7C02%7CT.C.van_der_sluis%40lumc.nl%7C6ba8084ee11d4c1ebbad08dcd88f2038%7Cc4048c4fdd544cbd80495457aacd2fb8%7C0%7C0%7C638623356452354578%7CUnknown%7CTWFpbGZsb3d8eyJWIjoiMC4wLjAwMDAiLCJQIjoiV2luMzIiLCJBTiI6Ik1haWwiLCJXVCI6Mn0%3D%7C0%7C%7C%7C&sdata=W1a%2FhMrrz1KWvjs5cacRwUAF9NYOnU8JSE8nXMMkXGo%3D&reserved=0
Enter token "wnurcqmetlcrbuj" into the box.

# Research involving human participants, their data, or biological material

Policy information about studies with human participants or human data. See also policy information about sex, gender (identity/presentation), and sexual orientation and race, ethnicity and racism.

Reporting on sex and gender | Human PBMCs were isolated from buffy coats obtained from Sanquin (Amsterdam, Netherlands). Donors at Sanquin include

| Reporting on sex and gender | all sexes and genders; samples were used without stratification. For the NEOLBC study, only female patients with breast cancer were included; all patient samples are therefore from females. |
|---|---|
| Reporting on race, ethnicity, or other socially relevant groupings | Human PBMCs were isolated from buffy coats obtained from Sanquin (Amsterdam, Netherlands). Donors at Sanquin include diverse races, ethnicities, and other socially relevant groupings; samples were used without stratification. |
| Population characteristics | Not applicable. |
| Recruitment | Recruitment criteria for the NEOLBC study are described in the corresponding manuscript (de Groot, Clin. Cancer Res. 2025). |
| Ethics oversight | The NEOLBC trial (NCT03283384) was conducted in accordance with the Declaration of Helsinki and approved by the Medical Ethical Committee of the LUMC. |

Note that full information on the approval of the study protocol must also be provided in the manuscript.

# Field-specific reporting

Please select the one below that is the best fit for your research. If you are not sure, read the appropriate sections before making your selection.

☒ Life sciences ☐ Behavioural & social sciences ☐ Ecological, evolutionary & environmental sciences

For a reference copy of the document with all sections, see nature.com/documents/nr-reporting-summary-flat.pdf

# Life sciences study design

All studies must disclose on these points even when the disclosure is negative.

| Sample size | Sample sizes for in vivo experiments (4–12 mice per group) were determined using G*Power or Power and Sample Size software and approved by the institutional statistician, providing 80% power at $\alpha = 0.05$. Sample sizes for ex vivo experiments were based on prior work in our laboratory to ensure sufficient numbers per group for informative results and statistical analysis. |
|---|---|
| Data exclusions | No animals or data points were excluded from the analyses, except one sample of the clinical data because the sample had insufficient cellularity (<25 cells/mm²), which was below the predefined analysis threshold. |
| Replication | All experiments were independently repeated at least 2–3 times with similar results, including ex vivo CD8* T cell assays and in vivo tumor models. Representative images and flow plots illustrate typical outcomes. No findings failed replication. Reproducibility was supported by consistent statistical results in independent experiments and biological replicates. |
| Randomization | Mice were randomized before the start of each experiment, and in tumor studies they were additionally randomized based on tumor size. Male and female animals were matched for age and sex. |
| Blinding | Blinding was not performed for the in vivo experiments; however, for selected in vitro assays, the experimenter acquiring the data was blinded to treatment, yielding results consistent with the non-blinded experiments. The researcher responsible for staining and analyzing clinical samples was blinded to sample identity. |

# Reporting for specific materials, systems and methods

We require information from authors about some types of materials, experimental systems and methods used in many studies. Here, indicate whether each material, system or method listed is relevant to your study. If you are not sure if a list item applies to your research, read the appropriate section before selecting a response.

## Materials & experimental systems

| n/a | Involved in the study |
|---|---|
| ☐ | ☒ Antibodies |
| ☐ | ☒ Eukaryotic cell lines |
| ☒ | ☐ Palaeontology and archaeology |
| ☐ | ☒ Animals and other organisms |
| ☒ | ☐ Clinical data |
| ☒ | ☐ Dual use research of concern |
| ☒ | ☐ Plants |

## Methods

| n/a | Involved in the study |
|---|---|
| ☒ | ☐ ChIP-seq |
| ☐ | ☒ Flow cytometry |
| ☒ | ☐ MRI-based neuroimaging |

## Antibodies

| Antibodies used | Mouse Flow cytometry:<br>anti-CD122 APC; clone  TM-b1 / Thermo Fisher / cat#17-1222-82 / RRID;AB_11151706 |
|---|---|

anti-CD127 Biotin; clone A7R34 / Thermo Fisher / cat#13-1271-85 / RRID;AB_466589
anti-CD127 RB780; clone A7R34 / BD Biosciences / cat#569066 / RRID;AB_3684748
anti-CD127 PE-Cy5; clone A7R34 / Biolegend / cat#135015 / RRID;AB_1937262
anti-CD25 PE/Cy7; clone PC61.5 / Thermo Fisher / cat#25-0251-82 / RRID;AB_469608
anti-CD25 PE; clone PC61 / BD Biosciences / cat#553866 / RRID;AB_395101
anti-CD25 APC; clone PC61.5 / Thermo Fisher / cat#17-0251-81 / RRID;AB_469365
anti-CD3 FITC; clone 145-2C11 / Thermo Fisher / cat#11-0031-85 / RRID;AB_464883
anti-CD3 PerCP-Cy5.5; clone 145-2C11 / Biolegend / cat#100327 / RRID;AB_893320
anti-CD3 BV785; clone 17A2 / Biolegend / cat#100231 / RRID;AB_11218805
anti-CD43 PE-Dazzle594; clone 1B11 / Biolegend / cat#121225 / RRID;AB_2687245
anti-CD44 IM7; clone BV785 / Biolegend / cat#103059 / RRID;AB_2571953
anti-CD44 eFluor450; clone IM7 / Thermo Fisher / cat#48-0441-82 / RRID;AB_1272246
anti-CD44 BUV805; clone IM7 / BD Biosciences / cat#741921 / RRID;AB_2871234
anti-CD45.1 PE; clone A20 / BD Biosciences / cat#553776 / RRID;AB_395044
anti-CD45.1 PE; clone A20 / BD Biosciences / cat#553776 / RRID;AB_395044
anti-CD62L PE; clone MEL-14 / BD Biosciences / cat#553151 / RRID;AB_394666
anti-CD69 eFluor450; clone H1.2F3 / Thermo Fisher / cat#48-0691-82 / RRID;AB_10719430
anti-CD8 FITC; clone 53-6.7 / Biolegend / cat#100706 / RRID;AB_312754
anti-CD8 APC; clone 53-6.7 / Biolegend / cat#100712 / RRID;AB_312751
anti-CD8 PerCP-Cy5.5; clone 53-6.7 / Biolegend / cat#100734 / RRID;AB_2075238
anti-CD8 APC-Fire750; clone 53-6.7 / Biolegend / cat#100766 / RRID;AB_2572113
anti-CD8 BUV395; clone 53-6.7 / BD Biosciences / cat#563786 / RRID;AB_2732919
anti-CD98 BUV615; clone RL388 / BD Biosciences / cat#752897 / RRID;AB_2916793
anti-CREL PE; clone 1RELAH5 / eBioscience / cat#12-6111-80 / RRID;AB_11042978
anti-CXCR3 PE; clone CXCR3-173 / Thermo Fisher / cat#12-1831-80 / RRID;AB_1210735
anti-EOMES PeCy7; clone Dan11mag / Thermo Fisher / cat#25-4875-82 / RRID;AB_2573454
anti-FDFT1 DyLight550; clone OT12F10 / Novus Biologicals / cat#NBP270715R / RRID;-
anti-G6PD APC-Cy7; clone EPR20668 / Abcam / cat#ab210702 / RRID;AB_2923527
anti-GLUT1 AF405; clone EPR3915 / Abcam / cat#ab252403 / RRID;AB_2783877
anti-Granzyme B PE; clone GB11 / Thermo Fisher / cat#12-8899-41 / RRID;AB_1659718
anti-ID2 eFluor450; clone ILCID2 / Thermo Fisher / cat#48-9475-82 / RRID;AB_2735053
anti-IFNy APC; clone XMG1.2 / Thermo Fisher / cat#17-7311-82 / RRID;AB_469504
anti-IL-2 PE; clone JES6-5H4 / Thermo Fisher / cat#12-7021-82 / RRID;AB_466150
anti-Ki67 BV605; clone 16A8 / Biolegend / cat#652413 / RRID;AB_2562664
anti-KLRG1 BV786; clone 2F1 / Biolegend / cat#138429 / RRID;AB_2629749
anti-KLRG1 BV605; clone 2F1 / Biolegend / cat#138419 / RRID;AB_2563357
anti-KLRG1 PeCy7; clone 2F1 / Biolegend / cat#138416 / RRID;AB_2561736
anti-LAG3 PE; clone C9B7W / Biolegend / cat#125208 / RRID;AB_2133343
anti-NFATc1 AF488; clone 7A6 / Biolegend / cat#649603 / RRID;AB_2561822
anti-PD1 FITC; clone RMP1-30 / Thermo Fisher / cat#11-9981-82 / RRID;AB_465467
anti-PD1 BV605; clone 29F.1A12 / Biolegend / cat#135220 / RRID;AB_2562616
anti-PKM PE; clone EPR10138(B) / Abcam / cat#ab210448 / RRID;AB_2941747
anti-pS6 unconjugated; clone D68F8 / Cell Signaling Technology / cat#4858 / RRID;AB_916156
anti-pSTAT5 PE; clone SRBCZX / Thermo Fisher / cat#12-9010-42 / RRID;AB_2572671
anti-Sca-1 BUV496; clone D7 / BD Biosciences / cat#750169 / RRID;AB_2874374
anti-TNF FITC; clone MP6-XT22 / Biolegend / cat#506304 / RRID;AB_315425

Human Flow cytometry
anti-ACC1 PE-Cy7; clone EPR23235-147 / Abcam / cat#ab269273 / RRID;AB_3665505
anti-Aldolase AF647; clone EPR23181-39 / Abcam / cat#ab275162 / RRID;AB_3099487
anti-ATP5a DyLight488; clone EPR13030(B) / Abcam / cat#ab176569 / RRID;AB_2801536
anti-CD25 BV711; clone M-A251 / Biolegend / cat#356138 / RRID;AB_2632781
anti-CD25 BV421; clone M-A251 / BioLegend / cat#356113 / RRID;AB_2562163
anti-CD25 PE-Cy7; clone M-A251 / BD Biosciences / cat#557741 / RRID;AB_396847
anti-CD3 Amcyan; clone SK7 / BD Biosciences / cat#339197 / RRID;AB_647355
anti-CD3 AlexaFluor700; clone UCHT1 / BD Biosciences / cat#557943 / RRID;AB_396952
anti-CD8 Pacific Blue; clone RPA-T8 / BD Biosciences / cat#558207 / RRID;AB_397058
anti-CD8 PeCy7; clone SK1 / Biolegend / cat#344712 / RRID;AB_2044008
anti-CD8 FITC; clone SK1 / BioLegend / cat#344704 / RRID;AB_1877178
anti-CD8 AF488; clone SK1 / Biolegend / cat#344716 / RRID;AB_10549301
anti-CD8 APC; clone SK1 / Biolegend / cat#344722 / RRID;AB_2075388
anti-CD8 BUV805; clone SK1 / BD Biosciences / cat#612889 / RRID;AB_2833078
anti-CD8 APC-Cy7; clone SK1 / BD Biosciences / cat#348813 / RRID;AB_2868857
anti-CD98 BUV395; clone UM7F8 / BD Biosciences / cat#744508 / RRID;AB_2742283

Human + Mouse Flow cytometry (cross-reactive)
anti-CPT1a PE-Cy5; clone EPR21843-71-2F / Abcam / cat#ab137040 / RRID;AB_2884996
anti-FDFT1 DyLight550; clone OTI2F10 / Novus Biologicals / cat#NBP2-70715R / RRID; -
anti-G6PD APC-Cy7; clone EPR20668 / Abcam / cat#ab210702 / RRID;AB_2923527
anti-GLUT1 AF405; clone EPR3915 / Abcam / cat#ab252403 / RRID;AB_2783877
anti-PKM PE; clone EPR10138(B) / Abcam / cat#ab210448 / RRID;AB_2941747
anti-pS6 (Ser235/236) unconjugated; clone D68F8 / Cell Signaling Technology / cat#4858 / RRID;AB_916156
anti-SDHA AlexaFluor647; clone EPR9043(B) / Abcam / cat#ab310057 / RRID; -
anti-γH2AX (Ser139) PE; clone N1-431 / BD Biosciences / cat#562377 / RRID;AB_2737611

Human IF antibodies
anti-CD3 -1;1000 - labelled with Opal 540 (1;200); clone EPR449E / Abcam / cat#ab52959 / RRID;-
anti-CD8 - 1;500 - labelled with Opal 520 (1;100); clone D8A8Y / Cell Signaling Technology / cat#85336 / RRID;AB_2800052
anti-Glut1 - 1;2000 - labelled with Opal Opal 570 (1;400); clone EPR3915 / Abcam / cat#ab252403 / RRID;AB_10903230
anti-Purified anti-Pan-Cytokeratin - 1;2000 - labelled with Opal Antibody Opal 650 (1;800); clone AE1/AE3 / Biolegend / cat#914204 / RRID;AB_2616960

Additional antibodies:
anti-Foxk1 unconjugated; clone; - / Cell Signaling Technology / cat# 12025S / RRID;AB_2797801
anti-Anti-Histone H3 (tri methyl K9) unconjugated; clone polyclonal / abcam / cat#ab8898 / RRID;AB_306848
anti-α-Rabbit IgG (H+L) HRP; clone polyclonal / Thermo Fisher/Invitrogen / cat# G-21234 / RRID;AB_2536530

Viability Stain:
zombie aqua / Biolegend / cat#423102
zombie NIR / Biolegend / cat#423106
7-AAD / ThermoFisher/ cat#A1310

| Validation | All commercially available antibodies have been validated by the respective suppliers. Anti-CD8 (clone 2.43), used for in vivo CD8+ T-cell depletion, was validated by measuring the reduction of CD8+ T cells in blood. Anti-PD-L1 (clone MIH-5), used for in vivo blockade of PD-1/PD-L1 interactions, was validated in a competition assay using fluorescently labeled MIH-5 for PD-L1 binding. |
| --- | --- |

# Eukaryotic cell lines

Policy information about cell lines and Sex and Gender in Research

| Cell line source(s) | TC-1 (mouse lung, RRID: CVCL_4699). Obtained from T.C. Wu.<br>MC-38 (RRID:CVCL_B288).<br>(E.G7-OVA (RRID:CVCL_3505).<br>MIH5 hybridoma (MIH5 (RRID:CVCL_CW88). Obtained from M. Azuma. |
| --- | --- |
| Authentication | TC-1 was authenticated by the in vivo recognition of E7 specific CD8 T cells. MC-38 was authenticated by parallel experiments where T cells with multiple known MC-38 specificities were infiltrated in these tumors. E.G7-OVA was authenticated by the in vivo response to (activated) SIINFEKL specific CD8 T cells. MIH-5 antibodies obtained from hybridoma cultures were tested by flow cytometry in a competitive binding assay for PD-L1. |
| Mycoplasma contamination | All cell lines were routinely tested negative for Mycoplasma via PCR. |
| Commonly misidentified lines<br>(See ICLAC register) | The cell lines used are not reported as known misidentified cell lines by the International Cell line Authentication Committee. |

# Animals and other research organisms

Policy information about studies involving animals; ARRIVE guidelines recommended for reporting animal research, and Sex and Gender in Research

| Laboratory animals | C57BL/6J mice were obtained from Charles River (L'Arbresle, France) and Janvier labs (Le Genest-Saint-Isle, France), and maintained in the animal facility of Leiden University Medical Centre (LUMC). OT-I mice (B6.Cg-PtprcaTg(TcraTcrb)1100Mjb), IL-2GFP-reporter mice (B6.Il2em1Lumc) (generated by the Transgenic Facility Leiden; Supplementary Fig. 2), IL-2flox/flox CreERT2)55, and Raptorflox/flox (B6.Cg-Rptortm2.1LexTg(Itgax-cre)1-1Reiz/J crossed in house with C57BL/6-Tg(Cd8a-cre)1Itan) were bred and maintained in the LUMC animal facility.<br><br>Animals were housed in individually ventilated cages under specific-pathogen free conditions at the animal facility at the Leiden University Medical Center (LUMC) at 20°C -22°C, a humidity of 45-65% RV and a light cycle of 6:30h-7:00h sunrise, 07:00h-18:00h daytime and 18:00h-18:30h sunset. All mice were 6-14 weeks of age. All animals received standard chow (SDS RM3 diet; DS801203G10R) |
| --- | --- |
| Wild animals | No wild animals were used in this study |
| Reporting on sex | Sex has not been considered in the design of the study. |
| Field-collected samples | The study does not involve samples collected in the field. |
| Ethics oversight | All animal experiments were approved by national (CCD) and local committees (Animal Welfare Body Leiden and Animal Experiment Committee Leiden) and performed under permit numbers AVD116002015271, AVD11600202013796, AVD11600202417987, and AVD1160020186804. |

Note that full information on the approval of the study protocol must also be provided in the manuscript.

# Plants

Seed stocks

NA

Novel plant genotypes

NA

Authentication

NA

# Flow Cytometry

## Plots

Confirm that:

☒ The axis labels state the marker and fluorochrome used (e.g. CD4-FITC).

☒ The axis scales are clearly visible. Include numbers along axes only for bottom left plot of group (a 'group' is an analysis of identical markers).

☐ All plots are contour plots with outliers or pseudocolor plots.

☒ A numerical value for number of cells or percentage (with statistics) is provided.

## Methodology

Sample preparation

Described in methods.

Instrument

Cytek Aurora (3L and 5L setup) and BD Fortessa flow cytometer (BD Biosciences) were used for analysis of cells. BD FACSAria was used for cell sorting for RNAsequencing.

Software

Flowjo software (TreeStar) and OMIQ data analysis software.

Cell population abundance

For RNA sequencing, spleens were isolated and a single cell suspension was obtained. CD8 T cells were enriched with magentic beads, CFSE labeled and stimulated as described in the methods. Purity of enriched cells was confirmed by flow cytometry. For sorting, lymphocytes were gated by FSC-A/SSC-A, then singlets were selected by FSC-A/FSC-H and SSC-A/SSC-H, exclusion of dead cells with 7AAD.

Gating strategy

For protein and or metabolic dye expression/intensity analysis, lymphocytes were gated by FSC-A/SSC-A, then singlets were selected by FSC-A/FSC-H, exclusion of dead cells by use of Zombie dyes or 7-AAD. Protein marker gating is based on the negative population from unstimulated cells.
Tetramer positive and KLRG1 postive cells were selected as followed: lymphocytes were gated by FSC-A/SSC-A, then singlets were selected by FSC-A/FSC-H, exclusion of dead cells by use of Zombie dyes or 7-AAD. Next cells CD3+/CD8+ cells were gated and Tetramer/CD44 double positive cells or KLRG1/CD44 double positive cells were gated and plotted. A gating strategy is provided as Supplementary Fig. 1.

☒ Tick this box to confirm that a figure exemplifying the gating strategy is provided in the Supplementary Information.

