## [Peer Review File · Nature Immunology]

Cell cycle arrest enhances CD8+ T cell effector function by potentiating glucose metabolism and IL-2 signaling

Corresponding Author: Dr Ramon Arens

Version 0:

Reviewer comments:

Reviewer #1

(Remarks to the Author)

van Haften et al. investigated the effects of cell cycle inhibitors on CD8+ T cell function. This study is important and timely. Inhibitors targeting cell cycle progression and checkpoints are approved for use in the treatment of cancer, and immune cells are critical in clearance of some solid tumours. Consistent with recent work on the CDK4/6 inhibitors abemaciclib (Goel et al. Nature 2017) and palbociclib (Lelliott et al., Heckler et al., in Cancer Discovery 2021), the researchers find that cell cycle inhibition during CD8+ T cell activation modulates immune phenotype by increasing glucose and amino acid metabolism prior to cell cycle entry. This effect was observed in human and mouse CD8+ T cells in vitro. The authors also demonstrate that targeting cell cycle progression and cell cycle checkpoints can enhance immune responses by increasing the fraction of antigen-specific CD8+ T cells and tumour cell clearance in vivo. However, I have major conceptual and technical concerns with the manuscript, as described below.

Major concerns:

1. To what extent are the phenotypes driven by DNA damage? Hydroxyurea induces replication stress and replication stress-induced DNA damage. The researchers should characterize the level of DNA damage in treated CD8+ T cells, particularly in the context of the various cell cycle inhibitors used in Figure 1. Indeed, the DNA damage responsive gene, CDKN1A (p21) is induced in "Released" and "Arrested" compared to "Non-arrested" cells (Fig. 2e)
2. Failure to 'blast' (Ext. Fig. 1c) suggests that CD8+ T cells treated with cell cycle inhibitors do not activate properly. Could this be off-target effects of the drugs used?
3. I am concerned that time in culture is a confounding factor in the comparisons between "Non-arrested cells" and "Released cells", which have been in culture for 60 and 120 hours, respectively. "For example, Fig. 4 e-g shows the differential effect of disabling mTOR signaling in "Non-arrested" versus "Released" cells. It is known that mTOR inhibition delays cell cycle entry following TCR activation but does not significantly delay cell cycle progression in CTLs (Howden et al. Nature Immunology 2019).
4. To what extent are the in vivo responses of HU specific to CD8 T cells? The graph shown in Fig.6e (and figure caption) is misleading because in the Methods section the tumor bearing animals were pre-treated with hydroxyurea, thus it is not possible to exclude a non-T cell response.

Misc:

1. Fig. 2b – typo in plot title
2. Rationale should be provided for why hydroxyurea was chosen for the remaining experiments, and not the other cell cycle inhibitors
3. Fig. 1 in vivo experiments. Does HU treatment result in reduced T cell numbers overall? If so, the fraction of antigen-specific CD8s will look apparently larger as shown in the Figure, but potentially not for the reason proposed (enhanced proliferation by antigen-specific CD8s).

4. What is the viability following HU treatment? The image in Ext. Fig. 1d suggests extensive cell death

Reviewer #2

(Remarks to the Author)

This manuscript sets an ambitious goal to assess the affect of uncoupling T cell programming from proliferation, via the use of T cells that have been cell cycle arrested in vitro using inhibitors of specific stages of the cell cycle. The premise of the study is important as it is difficult to disentangle proliferation from T cell activation programs. It is important to note that the authors primarily study T cell activation and differentiation programs upon re-entry into the cell cycle which again initiates proliferation. It seems that the authors actually find cell cycle re-entry dependent phenotypes (more coupled). Nonetheless, the study is timely and novel. Some main points should be addressed to fully interpret the conclusions.

1. The authors are studying T cells where cell cycle arrest as effectively been temporally synchronized and then released. So, many of these effects are in T cells with re-entry into the cell cycle (after a proliferation pause). Here the cells have initiated the cell proliferation program, so the interpretation might not be that the T cell phenotypes are due to uncoupling from cell proliferation, but instead, could be due to a temporary pause, followed by a burst in S phase or some alteration of the cell cycle and proliferation. When cells re-enter the cell cycle, why does the rate of proliferation increase in vitro? The authors' own data show that there is enhanced proliferation after a pause, making it difficult to 'uncouple proliferation from the T cell programs' Mechanistically – what accounts for the increased proliferation? Do the cells have different stages of the cell cycle compared to cells that have not been 'paused'? Are there differences in apoptosis?
2. It is interesting that metabolic genes are differentially expressed in cell cycle paused cells compared to non-arrested cells, but without careful examination of the drivers of increased cell cycle after the pause, it is difficult to fully interpret and conclude that this is a new mechanism. Could the increased cell proliferation also be enhancing metabolism? Are the differences just due to differences in the timing?
3. Overall the metabolic observations are correlative. Without more direct experiments and genetic manipulations to show essentiality, the authors should tone down their conclusions that the T cells stockpile nutrients to mediate increased proliferation or T cell programs after re-entry.
4. There is a large literature studying the means by which cancer cells and primary cells re-enter the cell cycle. Is this difference in signaling observed in T cells similar? It would be helpful to add a section to the discussion.
5. It is interesting that IL-2 decreases in proliferating cells, but is increased in proliferating cells that have been cell cycle arrested. Is there a direct link between re-entry into the cell cycle and IL-2? Does any aspect of cell cycle regulatory complexes affect IL-2?
6. What is the specific role of cholesterol metabolism in cell cycle re-entry? Does the cell cycle directly regulate cholesterol?
7. The immune profiling seems preliminary and limited – for instance, how are other cytokines affected?
8. Are the programs observed in arrested cells comparable to memory immune cells that have been re-activated? This seems like the more reasonable comparison and should also be experimentally tested and discussed.

Decision Letter:

19th Dec 2024

Dear Dr Arens,

As you are aware, your Article, "Cell cycle inhibition empowers CD8+ T cells by potentiating glucose metabolism" has now been seen by 2 referees and while they find your work of considerable potential interest, they have raised substantial concerns that must be addressed. We have also looked over your response to these concerns, thanks for sending me that document. I am happy to say that we are satisfied by your revision plan, and hence we would be very interested in considering a revised version.

If you choose to revise your manuscript taking into account all reviewer and editor comments, please highlight all changes in the manuscript text file in Microsoft Word format.

* If you have not done so already please begin to revise your manuscript so that it conforms to our Article format instructions at <http://www.nature.com/ni/authors/index.html>. Refer also to any guidelines provided in this letter.

The Reporting Summary can be found here:
<https://www.nature.com/documents/nr-reporting-summary.pdf>

Extended Data figures and tables are online-only (appearing in the online PDF and full-text HTML version of the paper), peer-reviewed display items that provide essential background to the Article but are not included in the printed version of the paper due to space constraints or being of interest only to a few specialists. A maximum of ten Extended Data display items (figures and tables) is typically permitted. When re-submitting your manuscript, please ensure that any supplementary figures and tables that are more critical to the manuscript's conclusions are converted to Extended data to increase these data's visibility.

Link Redacted

If you wish to submit a suitably revised manuscript we would hope to receive it within 6 months. If you cannot send it within this time, please let us know. We will be happy to consider your revision so long as nothing similar has been accepted for publication at Nature Immunology or published elsewhere.

Nature Immunology is committed to improving transparency in authorship. As part of our efforts in this direction, we are now requesting that all authors identified as 'corresponding author' on published papers create and link their Open Researcher and Contributor Identifier (ORCID) with their account on the Manuscript Tracking System (MTS), prior to acceptance. ORCID helps the scientific community achieve unambiguous attribution of all scholarly contributions. You can create and link your ORCID from the home page of the MTS by clicking on 'Modify my Springer Nature account'. For more information please visit www.springernature.com/orcid.

Thank you for the opportunity to review your work.

Sincerely,

Nick Bernard, PhD
Senior Editor
Nature Immunology

Reviewers' Comments:

Reviewer #1 (Remarks to the Author):

van Haften et al. investigated the effects of cell cycle inhibitors on CD8+ T cell function. This study is important and timely. Inhibitors targeting cell cycle progression and checkpoints are approved for use in the treatment of cancer, and immune cells are critical in clearance of some solid tumours. Consistent with recent work on the CDK4/6 inhibitors abemaciclib (Goel et al. Nature 2017) and palbociclib (Lelliott et al., Heckler et al., in Cancer Discovery 2021), the researchers find that cell cycle inhibition during CD8+ T cell activation modulates immune phenotype by increasing glucose and amino acid metabolism prior to cell cycle entry. This effect was observed in human and mouse CD8+ T cells in vitro. The authors also demonstrate that targeting cell cycle progression and cell cycle checkpoints can enhance immune responses by increasing the fraction of antigen-specific CD8+ T cells and tumour cell clearance in vivo. However, I have major conceptual and technical concerns with the manuscript, as described below.

Major concerns:

1. To what extent are the phenotypes driven by DNA damage? Hydroxyurea induces replication stress and replication stress-induced DNA damage. The researchers should characterize the level of DNA damage in treated CD8+ T cells, particularly in the context of the various cell cycle inhibitors used in Figure 1. Indeed, the DNA damage responsive gene, CDKN1A (p21) is induced in “Released” and “Arrested” compared to “Non-arrested” cells (Fig. 2e)

2. Failure to ‘blast’ (Ext. Fig. 1c) suggests that CD8+ T cells treated with cell cycle inhibitors do not activate properly. Could this be off-target effects of the drugs used?

3. I am concerned that time in culture is a confounding factor in the comparisons between “Non-arrested cells” and “Released cells”, which have been in culture for 60 and 120 hours, respectively. “For example, Fig. 4 e-g shows the differential effect of disabling mTOR signaling in “Non-arrested” versus “Released” cells. It is known that mTOR inhibition delays cell cycle entry following TCR activation but does not significantly delay cell cycle progression in CTLs (Howden et al. Nature Immunology 2019).

4. To what extent are the in vivo responses of HU specific to CD8 T cells? The graph shown in Fig.6e (and figure caption) is misleading because in the Methods section the tumor bearing animals were pre-treated with hydroxyurea, thus it is not possible to exclude a non-T cell response.

Misc:

1. Fig. 2b – typo in plot title

2. Rationale should be provided for why hydroxyurea was chosen for the remaining experiments, and not the other cell cycle inhibitors

3. Fig. 1 in vivo experiments. Does HU treatment result in reduced T cell numbers overall? If so, the fraction of antigen-specific CD8s will look apparently larger as shown in the Figure, but potentially not for the reason proposed (enhanced proliferation by antigen-specific CD8s).

4. What is the viability following HU treatment? The image in Ext. Fig. 1d suggests extensive cell death

Reviewer #2 (Remarks to the Author):

This manuscript sets an ambitious goal to assess the affect of uncoupling T cell programming from proliferation, via the use of T cells that have been cell cycle arrested in vitro using inhibitors of specific stages of the cell cycle. The premise of the study is important as it is difficult to disentangle proliferation from T cell activation programs. It is important to note that the authors primarily study T cell activation and differentiation programs upon re-entry into the cell cycle which again initiates proliferation. It seems that the authors actually find cell cycle re-entry dependent phenotypes (more coupled). Nonetheless, the study is timely and novel. Some main points should be addressed to fully interpret the conclusions.

1. The authors are studying T cells where cell cycle arrest as effectively been temporally synchronized and then released. So, many of these effects are in T cells with re-entry into the cell cycle (after a proliferation pause). Here the cells have initiated the cell proliferation program, so the interpretation might not be that the T cell phenotypes are due to uncoupling from cell proliferation, but instead, could be due to a temporary pause, followed by a burst in S phase or some alteration of the cell cycle and proliferation. When cells re-enter the cell cycle, why does the rate of proliferation increase in vitro? The authors’ own data show that there is enhanced proliferation after a pause, making it difficult to ‘uncouple proliferation from the T cell programs’ Mechanistically – what accounts for the increased proliferation? Do the cells have different stages of the cell cycle compared to cells that have not been ‘paused’? Are there differences in apoptosis?

2. It is interesting that metabolic genes are differentially expressed in cell cycle paused cells compared to non-arrested cells, but without careful examination of the drivers of increased cell cycle after the pause, it is difficult to fully interpret and conclude that this is a new mechanism. Could the increased cell proliferation also be enhancing metabolism? Are the differences just due to differences in the timing?

3. Overall the metabolic observations are correlative. Without more direct experiments and genetic manipulations to show essentiality, the authors should tone down their conclusions that the T cells stockpile nutrients to mediate increased proliferation or T cell programs after re-entry.

4. There is a large literature studying the means by which cancer cells and primary cells re-enter the cell cycle. Is this difference in signaling observed in T cells similar? It would be helpful to add a section to the discussion.

5. It is interesting that IL-2 decreases in proliferating cells, but is increased in proliferating cells that have been cell cycle arrested. Is there a direct link between re-entry into the cell cycle and IL-2? Does any aspect of cell cycle regulatory complexes affect IL-2?

6. What is the specific role of cholesterol metabolism in cell cycle re-entry? Does the cell cycle directly regulate cholesterol?

7. The immune profiling seems preliminary and limited – for instance, how are other cytokines affected?

8. Are the programs observed in arrested cells comparable to memory immune cells that have been re-activated? This seems like the more reasonable comparison and should also be experimentally tested and discussed.

Version 1:

Reviewer comments:

Reviewer #1

(Remarks to the Author)

The authors have not addressed the first two concerns I have with the study.

On point 1: hydroxyurea is an anti-cancer agent that functions by inhibiting ribonucleotide reductase, thereby reducing dNTPs, promoting replication stress and replication stress-induced DNA damage, in addition to oxidative damage, that is exacerbated by prolonged treatment (e.g. as reviewed in Musialek 2021). Therefore, a significant amount of cellular stress would be expected in the 60-hour treatment time in the experiments shown. Perhaps the mechanisms of action of hydroxyurea are different in CD8 T cells. But this is a controversial claim that requires substantial evidence.

Y15 phosphorylation on CDK1/2 is a poor choice to discriminate DNA damage from cell cycle control by phosphoproteomics. T14 and Y15 are difficult to discriminate by phosphoproteomics and indeed, phosphorylation on both T14 and Y15 can occur, which would reduce the intensity for the monophosphorylated Y15 peptide shown in the extended data. Biologically, this means Y15 can still be phosphorylated, even if there is a decrease in the Y15 monophosphorylated form, because the chemical species has shifted to the doubly phosphorylated form.

These differences could also be due to a difference in the cell cycle position of cells. For example, if "Released" cells were mostly in G2 phase (following multiple rounds of division), then these cells would be expected to have lower Y15 phosphorylation compared to "Non-arrested", and therefore not a reporter of an absence of DNA damage.

A direct measure of DNA damage is therefore crucial.

On point 2: Data on whether cells 'blast' (such as scatter measurements by flow cytometry) should be shown for all the cell cycle inhibitors shown in Figure 1. The authors' reply only indicate data for hydroxyurea.

On another point, in response to reviewer 2, the authors cite papers that do not appear to support their claims, e.g. "while IL-2 regulates CDK expression and activity, CDKs in turn modulate IL-2 production (Huleatt et al., 2003; Appleman et al., 2000)"

The Huleatt paper shows that p27 is crucial for arrest following IL-2 starvation.

The Appleman paper shows that IL-2 is crucial for activation of D-CDK4/6 and E-CDK2 complexes and downregulation of p27.

These data in these studies do not support the idea that CDKs modulate IL-2 production. What they do show is that IL-2 withdrawal or CD28 deletion impacts cell cycle progression by reduced CDK activity. Not the other way around.

Reviewer #3

(Remarks to the Author)

This manuscript provides a thorough mechanistic understanding of the "bounce-back" phenomenon whereby activated CD8 T cells that are paused in cell cycle are able to recover faster than their un-paused counterparts. The clinical relevance of this finding cannot be overstated, given the extensive use of CDK4/6 inhibitors and other cell cycle inhibitors for cancer therapy. It is extremely reassuring to know that T cells are able to recover quickly from intermittent periods of proliferative arrest, and that they spend the arrested time building up glycogen and other metabolic stores needed for rapid growth.

Minor comments:

To my knowledge, the first report of the bounce-back phenomenon and increased IL-2 production based on cell cycle state was 20 years ago in JI (PMID: 15778358). Obviously, the current authors have greatly expanded on this early finding, but the original paper should be cited.

Figure 7i is lacking a color key for the immunofluorescence image.

Decision Letter:

8th Oct 2025

Dear Dr Arens,

As you are already aware, your Article "Cell cycle arrest empowers CD8+ T cells by potentiating glucose metabolism and IL-2 signaling" has now been seen by 2 referees.

Reviewer 2 did not supply a report so we added a third and mediating reviewer. There are some minor fixes needed, particularly reviewer 1 comments about needing to add a more direct measure of DNA damage. Having now looked over your Author Response to these comments I can say that we are satisfied by this short revision plan.

We therefore invite you to revise your manuscript taking into account all reviewer and editor comments. Please highlight all changes in the manuscript text file in Microsoft Word format.

* If you have not done so already please begin to revise your manuscript so that it conforms to our Article format instructions at <http://www.nature.com/ni/authors/index.html>. Refer also to any guidelines provided in this letter.

* Please include a revised version of any required reporting checklist. It will be available to referees to aid in their evaluation of the manuscript goes back for peer review. They are available here:

Reporting summary:

Please note, Extended Data figures and tables are online-only (appearing in the online PDF and full-text HTML version of the paper), peer-reviewed display items that provide essential background to the Article but are not included in the printed version of the paper due to space constraints or being of interest only to a few specialists. A maximum of ten Extended Data display items (figures and tables) is typically permitted. When re-submitting your manuscript, please ensure that any supplementary figures and tables that are more critical to the manuscript's conclusions are converted to Extended data to increase these data's visibility.

Link Redacted

We hope to receive your revised manuscript within two weeks. If you cannot send it within this time, please let us know. We will be happy to consider your revision so long as nothing similar has been accepted for publication at Nature Immunology or published elsewhere.

Nature Immunology is committed to improving transparency in authorship. As part of our efforts in this direction, we are now requesting that all authors identified as 'corresponding author' on published papers create and link their Open Researcher and Contributor Identifier (ORCID) with their account on the Manuscript Tracking System (MTS), prior to acceptance. ORCID helps the scientific community achieve unambiguous attribution of all scholarly contributions. You can create and link your ORCID from the home page of the MTS by clicking on 'Modify my Springer Nature account'. For more information please visit www.springernature.com/orcid.

Sincerely,

Nick Bernard, PhD
Senior Editor

Referee expertise:

Reviewers' Comments:

Reviewer #1 (Remarks to the Author):

The authors have not addressed the first two concerns I have with the study.

On point 1: hydroxyurea is an anti-cancer agent that functions by inhibiting ribonucleotide reductase, thereby reducing dNTPs, promoting replication stress and replication stress-induced DNA damage, in addition to oxidative damage, that is exacerbated by prolonged treatment (e.g. as reviewed in Musialek 2021). Therefore, a significant amount of cellular stress would be expected in the 60-hour treatment time in the experiments shown. Perhaps the mechanisms of action of hydroxyurea are different in CD8 T cells. But this is a controversial claim that requires substantial evidence.

Y15 phosphorylation on CDK1/2 is a poor choice to discriminate DNA damage from cell cycle control by phosphoproteomics. T14 and Y15 are difficult to discriminate by phosphoproteomics and indeed, phosphorylation on both T14 and Y15 can occur, which would reduce the intensity for the monophosphorylated Y15 peptide shown in the extended data. Biologically, this means Y15 can still be phosphorylated, even if there is a decrease in the Y15 monophosphorylated form, because the chemical species has shifted to the doubly phosphorylated form.

These differences could also be due to a difference in the cell cycle position of cells. For example, if "Released" cells were mostly in G2 phase (following multiple rounds of division), then these cells would be expected to have lower Y15 phosphorylation compared to "Non-arrested", and therefore not a reporter of an absence of DNA damage.

A direct measure of DNA damage is therefore crucial.

On point 2: Data on whether cells 'blast' (such as scatter measurements by flow cytometry) should be shown for all the cell cycle inhibitors shown in Figure 1. The authors' reply only indicate data for hydroxyurea.

On another point, in response to reviewer 2, the authors cite papers that do not appear to support their claims, e.g. "while IL-2 regulates CDK expression and activity, CDKs in turn modulate IL-2 production (Huleatt et al., 2003; Appleman et al., 2000)"

The Huleatt paper shows that p27 is crucial for arrest following IL-2 starvation.

The Appleman paper shows that IL-2 is crucial for activation of D-CDK4/6 and E-CDK2 complexes and downregulation of p27.

These data in these studies do not support the idea that CDKs modulate IL-2 production. What they do show is that IL-2 withdrawal or CD28 deletion impacts cell cycle progression by reduced CDK activity. Not the other way around.

Reviewer #3 (Remarks to the Author):

This manuscript provides a thorough mechanistic understanding of the "bounce-back" phenomenon whereby activated CD8 T cells that are paused in cell cycle are able to recover faster than their un-paused counterparts. The clinical relevance of this finding cannot be overstated, given the extensive use of CDK4/6 inhibitors and other cell cycle inhibitors for cancer therapy. It is extremely reassuring to know that T cells are able to recover quickly from intermittent periods of proliferative arrest, and that they spend the arrested time building up glycogen and other metabolic stores needed for rapid growth.

Minor comments:

To my knowledge, the first report of the bounce-back phenomenon and increased IL-2 production based on cell cycle state was 20 years ago in JI (PMID: 15778358). Obviously, the current authors have greatly expanded on this early finding, but the original paper should be cited.

Figure 7i is lacking a color key for the immunofluorescence image.

Version 2:

Reviewer comments:

Reviewer #1

(Remarks to the Author)

The reviewers have addressed my concerns. As highlighted in my original review, it is my opinion that this study is highly timely and will be of broad interest. The latest experiments included will be key to aid readers (especially in the cell cycle field) in interpreting the main conclusions.

Decision Letter:

Our ref: NI-A38581B

10th Nov 2025

Dear Dr. Arens,

Thank you for submitting your revised manuscript "Cell cycle arrest empowers CD8+ T cells by potentiating glucose metabolism and IL-2 signaling" (NI-A38581B). It has now been seen again by reviewer 1 only, and their comments are below. The reviewer finds that the paper has improved in revision, and therefore we'll be happy in principle to publish it in Nature Immunology, pending minor revisions to comply with our editorial and formatting guidelines.

We will now perform detailed checks on your paper and will send you a checklist detailing our editorial and formatting requirements in about a week. Please do not upload the final materials and make any revisions until you receive this additional information from us.

If you had not uploaded a Word file for the current version of the manuscript, we will need one before beginning the editing process; please email that to immunology@us.nature.com at your earliest convenience.

Thank you again for your interest in Nature Immunology Please do not hesitate to contact me if you have any questions.

Sincerely,

Nick Bernard, PhD
Senior Editor
Nature Immunology

Reviewer #1 (Remarks to the Author):

The reviewers have addressed my concerns. As highlighted in my original review, it is my opinion that this study is highly timely and will be of broad interest. The latest experiments included will be key to aid readers (especially in the cell cycle field) in interpreting the main conclusions.

Response to reviewers; point-by-point reply

We thank the referees for their time to review our manuscript thoroughly providing constructive critique and suggestions.

Reviewers' comments: Reviewer #1 (Remarks to the Author)

van Haften et al. investigated the effects of cell cycle inhibitors on CD8+ T cell function. This study is important and timely. Inhibitors targeting cell cycle progression and checkpoints are approved for use in the treatment of cancer, and immune cells are critical in clearance of some solid tumours. Consistent with recent work on the CDK4/6 inhibitors abemaciclib (Goel et al. Nature 2017) and palbociclib (Lelliott et al., Heckler et al., in Cancer Discovery 2021), the researchers find that cell cycle inhibition during CD8+ T cell activation modulates immune phenotype by increasing glucose and amino acid metabolism prior to cell cycle entry. This effect was observed in human and mouse CD8+ T cells in vitro. The authors also demonstrate that targeting cell cycle progression and cell cycle checkpoints can enhance immune responses by increasing the fraction of antigen-specific CD8+ T cells and tumour cell clearance in vivo. However, I have major conceptual and technical concerns with the manuscript, as described below.

Major concerns:

1. To what extent are the phenotypes driven by DNA damage? Hydroxyurea induces replication stress and replication stress-induced DNA damage. The researchers should characterize the level of DNA damage in treated CD8+ T cells, particularly in the context of the various cell cycle inhibitors used in Figure 1. Indeed, the DNA damage responsive gene, CDKN1A (p21) is induced in “Released” and “Arrested” compared to “Non-arrested” cells (Fig. 2e)

Reply: In our transcriptomic and phospho-proteomic analysis the DNA damage pathway was not among the top pathways that were differential among the arrested compared to non-arrested. However, we agree it will be informative to add the DNA damage data (Extended Data Fig. 4 and 7) given that cell cycle inhibitors can induce the DNA damage pathway. We have added the following text in the manuscript:

Page 8: The DNA damage response (DDR) pathway was not identified as differentially regulated, consistent with the absence of key DDR gene transcripts specifically associated with either the arrested or released state (**Extended Data Fig. 4**).

Page 12: Consistent with the transcriptome data, DDR-associated phosphosites were detectable but did not exhibit the characteristic phosphorylation pattern indicative of DNA damage or replication stress, such as activation of core DDR proteins including ATM, ATR, PRKDC, CHEK1/2, and components of the FANC pathway (**Extended Data Fig. 7a**). Kinase phosphosite analysis revealed a coordinated cell cycle progression pattern, with reduced inhibitory CDK2_Y15 phosphorylation in released cells concurrent with enhanced phosphorylation of multiple cell cycle-regulatory kinases (**Extended Data Fig. 7b**).

The reviewer raises an important point about CDKN1A (p21) transcriptional induction and potential DNA damage contributions. While CDKN1A has dual functions in both cell cycle regulation and DNA damage response, our phosphoproteomics analysis reveals a pattern consistent with coordinated cell cycle programming. Kinase phosphosites revealed a coordinated cell cycle progression pattern, marked by reduced inhibitory CDK2_Y15 phosphorylation and concurrent activation of multiple cell cycle-regulatory kinases in released cells. Though some DDR-associated phosphosites were detected, they did not display the characteristic pattern of DNA damage or replication stress responses. The

integrated data suggests these changes primarily reflect cell cycle programming rather than DNA damage responses, consistent with our observation of similar phenotypes using different cell cycle inhibitors.

2. Failure to 'blast' (Ext. Fig. 1c) suggests that CD8+ T cells treated with cell cycle inhibitors do not activate properly. Could this be off-target effects of the drugs used?

Reply: Based on the activation markers such as CD25 (Figure 6g, 6h) but also on CD69 (data not shown in previous version but shown in the revised version: Fig. 1g), pS6 (Figure 5c, 5d) and CD62L down-regulation (Fig. 1g, Extended Data Fig. 1d) we concluded that the cell-cycle arrested cells are activated substantially. Also on basis of the percentage of blasted cells, the arrested cells seem to be activated. Besides the addition of the CD69 data we provided also the % of blasted cells to illustrate this better (Extended Data Fig. 1f).

3. I am concerned that time in culture is a confounding factor in the comparisons between "Non-arrested cells" and "Released cells", which have been in culture for 60 and 120 hours, respectively. "For example, Fig. 4 e-g shows the differential effect of disabling mTOR signaling in "Non-arrested" versus "Released" cells. It is known that mTOR inhibition delays cell cycle entry following TCR activation but does not significantly delay cell cycle progression in CTLs (Howden et al. Nature Immunology 2019).

Reply: We agree with the reviewer that time in culture is an important factor to consider. We have therefore also provided in depth in vivo kinetics of transient arrested CD8 T cells (Fig. 1h, Fig. 7e), which mimics the ex vivo studies.

With respect to the mTOR signaling, we have added extensively traced mTOR signaling by pS6 levels over time, and found that non-arrested cells remained to have high pS6 levels whereas released cells had reduced levels, already observed 24 hours after release till at least 60 h after release (Extended Data Fig. 8a).

The following text was added: Kinetic analysis showed that non-arrested cells remained high in pS6 levels (up to at least 120h post stimulation) while after cell cycle arrest the released cells showed decreased levels, already noticeable 24 hours after release (**Extended Data Fig. 8a**).

The data with Raptor (mTORC1) deficient CD8 T cells and rapamycin confirms these findings (Figure 5e-h).

4. To what extent are the in vivo responses of HU specific to CD8 T cells? The graph shown in Fig.6e (and figure caption) is misleading because in the Methods section the tumor bearing animals were pre-treated with hydroxyurea, thus it is not possible to exclude a non-T cell response.

Reply: The description of experiment Figure 6e (in the revised experiment Figure 7j) in the Methods was unfortunately not correct. The graph and legend were correct. We apologize for this error in the Methods section. Thus, not the mice but only the OT-I cells were pre-treated with HU (before transfer). We have corrected the Methods section accordingly.

This data shows an exclusive T-cell mediated effect. Nevertheless, we appreciate the concern regarding possible non-T cell effects of HU treatment when used in vivo, such as the experiment shown in 6f

(current Figure 7k). Hence, we have performed an experiment in which CD8 T cells are depleted to study therapy-mediated effects on tumor control. We found that HU-mediated effects on tumor growth inhibition is dependent on CD8 T cells (Extended Data Fig. 10c).

Misc:

1. Fig. 2b – typo in plot title

Reply: We apologies for the typo. We have corrected this in the figure (Arested> Arrested).

2. Rationale should be provided for why hydroxyurea was chosen for the remaining experiments, and not the other cell cycle inhibitors

Reply: We have consistently observed that HU provided an overall better survival of the cells of the cells during ex vivo treatment compared to other cell cycle inhibitors. We have added this data to the manuscript (Extended Data Fig. 1b). Moreover, HU can easily be used in vivo while e.g. RO-3306 is complicated. The following text was added in the manuscript:

Page 5: HU consistently provided the best overall CD8⁺ T cell survival compared to other cell cycle inhibitors (**Extended Data Fig 1b**).

3. Fig. 1 in vivo experiments. Does HU treatment result in reduced T cell numbers overall? If so, the fraction of antigen-specific CD8s will look apparently larger as shown in the Figure, but potentially not for the reason proposed (enhanced proliferation by antigen-specific CD8s).

Reply: We thank the reviewer for addressing this topic. We have analyzed the influence of HU (and Topo) on the total CD8 T cells (using the datasets of Fig. 1h). The data show that on day 6 and day 10 post vaccination the cell cycle inhibitors (provided for 4 days after vaccination) do not reduce the percentage of CD8 T cells in the blood. We have included this data (Extended Data Fig. 2b) and added the following sentence (on page 6):

Moreover, the treatment itself did not result in a reduction the total population of total CD8⁺ T cells, indicating a specific effect on proliferating antigen-specific CD8⁺ T cells (**Extended data Fig. 2b**).

Moreover, the studies concerning the adoptive T cell transfers in tumor-bearing mice indicate that HU enhances antigen-specific CD8 T cell proliferation after treatment (Figure 7). Thus, HU treatment seems to specifically act on activated/proliferating CD8 T cells.

4. What is the viability following HU treatment? The image in Ext. Fig. 1d suggests extensive cell death

Reply: We appreciate this concern as an impact on viability may be expected from cell cycle inhibition. HU treatment, however, resulted in a relative modest additional cell death during culture. With the HU dose used in our experiments, the cell viability in the HU-arrested setting was approximately 10-15% lower compared to non-arrested cells. We have this data included in Extended Fig. 1b (related to Question 2, we have also provided a comparison with RO).

Reviewer #2

(Remarks to the Author)

This manuscript sets an ambitious goal to assess the affect of uncoupling T cell programming from proliferation, via the use of T cells that have been cell cycle arrested in vitro using inhibitors of specific stages of the cell cycle. The premise of the study is important as it is difficult to disentangle proliferation from T cell activation programs. It is important to note that the authors primarily study T cell activation and differentiation programs upon re-entry into the cell cycle which again initiates proliferation. It seems that the authors actually find cell cycle re-entry dependent phenotypes (more coupled). Nonetheless, the study is timely and novel. Some main points should be addressed to fully interpret the conclusions.

1. The authors are studying T cells where cell cycle arrest as effectively been temporally synchronized and then released. So, many of these effects are in T cells with re-entry into the cell cycle (after a proliferation pause). Here the cells have initiated the cell proliferation program, so the interpretation might not be that the T cell phenotypes are due to uncoupling from cell proliferation, but instead, could be due to a temporary pause, followed by a burst in S phase or some alteration of the cell cycle and proliferation. When cells re-enter the cell cycle, why does the rate of proliferation increase in vitro? The authors' own data show that there is enhanced proliferation after a pause, making it difficult to 'uncouple proliferation from the T cell programs' Mechanistically – what accounts for the increased proliferation? Do the cells have different stages of the cell cycle compared to cells that have not been 'paused'? Are there differences in apoptosis?

Reply: The reviewer brings up here an important remark which we have aimed to address with new experiments as follows.

Our data indicates that for the increased proliferation after transient cell cycle blockade, the rewiring of the metabolism leading to stock-piling of nutrients and enhanced glycolysis is important as well as the induction of IL-2. In our ex vivo experiments we used CD8 T cells that are first activated and subsequently inhibited before the cells can undergo cell cycle progression. To clearly show that during cell cycle inhibition, the glucose metabolism is already rewired we have combined Edu/FxCycle, indicating G1/G0, S and G2/M cell cycle phases with staining for PKM (Extended Data Fig. 6d). Our data showed that the arrested cells are in the G1/G0 phase or halted in the S phase (mainly in the low DNA content fraction (early S)) while non-arrested cells and released cells are also found in the G2/M phase and in the S phase (both low and high DNA content). In particular, PKM is already upregulated in the S phase of the arrested cells. we added the following text:

Increased PKM expression was specifically observed during the early S phase in HU-arrested cells, which displayed only low DNA content as determined by FxCycle staining (**Extended Data Fig. 6d**). Because HU blocks progression beyond the S phase, no cells advanced into G2/M under arrested conditions. Upon release from HU, however, cells progressed through the cell cycle, and displayed elevated PKM levels in both early and late S phases as well as in G2/M. Notably, PKM expression remained low in the G0/G1 phase across arrested, non-arrested and released conditions, highlighting a cell cycle–linked regulation of PKM that is associated with DNA replication and mitotic entry.

Are there differences in apoptosis? As mentioned to reviewer 1 we have included data on apoptosis of the CD8 T cells during cell cycle blockade (Extended Data Fig 1b).

2. It is interesting that metabolic genes are differentially expressed in cell cycle paused cells compared to non-arrested cells, but without careful examination of the drivers of increased cell cycle after the pause, it is difficult to fully interpret and conclude that this is a new mechanism. Could the increased cell proliferation also be enhancing metabolism? Are the differences just due to differences in the timing?

Reply: The connection between proliferation and metabolism has been addressed in reply to the previous question, where we show the link between glucose metabolism and cell proliferation. To address the questions related to the timing we have performed several experiments. We have performed an experiment with shorter cell cycle blockade (12h instead of 60h). The shorter cell cycle arrest allows less time for stock-piling and other metabolic adaptations. The effects on proliferation were also less pronounced. We have added the following in the revised manuscript:

Notably, a shorter period of cell cycle arrest (12h), permitting less time for stockpiling nutrients such as glucose, enhanced proliferation upon release but to a lesser extent (as compared to 60h), highlighting the functional relevance of metabolic preconditioning during arrest (**Extended Data Fig. 6f**).

In addition (see also response to referee 1 Q3), we have more extensively traced mTOR signaling by pS6 levels over time. We found that non-arrested cells remained to have high pS6 levels whereas released cells had reduced levels, already observed 24 hours after release till at least 60 h after release (Extended Data Fig. 8a).

The following text was added: Kinetic analysis showed that non-arrested cells remained high in pS6 levels (up to at least 120h post stimulation) while after cell cycle arrest the released cells showed decreased levels, already noticeable 24 hours after release (**Extended Data Fig. 8a**).

3. Overall the metabolic observations are correlative. Without more direct experiments and genetic manipulations to show essentiality, the authors should tone down their conclusions that the T cells stockpile nutrients to mediate increased proliferation or T cell programs after re-entry.

Reply: The occurrence of stock-piling is shown independently by the polar metabolite data (Figure 3) and by the glycogen storage experiment (Figure 3f, 3g). To extend this data and to address the link between stock piling and increased proliferation we have performed an experiment with an inhibitor of glycogen phosphorylase (CP91149). Using this inhibitor we were able to directly show the importance of glucose storage and proliferation.

The following text was added to the revised manuscript: Restraining glycogen breakdown by selective inhibition of glycogen phosphorylase³⁶ using CP91149 impaired proliferation in a dose-dependent manner, indicating that cell-intrinsic glycogenolysis and glycolytic activity supports proliferation following transient cell cycle arrest (**Fig. 3h**).

In addition, as also discussed in the previous question, reducing the time for stock piling also reduces the effects on proliferation (Extended Data Fig. 6f).

4. There is a large literature studying the means by which cancer cells and primary cells re-enter the cell cycle. Is this difference in signaling observed in T cells similar? It would be helpful to add a section to the discussion.

Reply: We appreciate the reviewer's insightful question. We will expand the Discussion to address this point. The effects of temporal cell cycle blockade on other cell types, including tumor cells, certainly warrant further investigation. However, given the unique role of IL-2 in our system, we consider it most likely that CD8 T cells are particularly sensitive and respond differently compared to other cells. Supporting this idea, we have analyzed a clinical cohort in which CD8⁺ T cells infiltrating tumors upregulated GLUT1 following treatment with the CDK4/6 inhibitor Ribociclib. We have now added these data to the manuscript (Fig. 7g–i), as we consider they provide additional strength and clinical relevance to our findings.

The added discussion text: While our study focused on CD8⁺ T cells, it is important to consider that other immune cells and tumor cells may also undergo metabolic rewiring upon cell cycle blockade. However, a critical distinction is that CD8⁺ T cells produce large amounts of IL-2 during arrest, a feature not shared by most other cell types, and this cytokine appears central to their enhanced proliferative response upon release. This suggests that CD8⁺ T cells may uniquely exploit temporal arrest to stockpile nutrients, reprogram metabolism, and prime for IL-2–driven expansion, whereas tumor cells may primarily experience growth restriction.

The clinical cohort text: Next, we evaluated whether transient cell cycle blockade affects CD8⁺ T cell metabolism in a clinical setting. In biopsies from patients with stage II/III breast cancer, the frequency of GLUT1-expressing CD8⁺ T cells increased following intermittent ribociclib treatment compared to baseline levels during letrozole monotherapy (**Fig. 7g–i**).

5. It is interesting that IL-2 decreases in proliferating cells, but is increased in proliferating cells that have been cell cycle arrested. Is there a direct link between re-entry into the cell cycle and IL-2? Does any aspect of cell cycle regulatory complexes affect IL-2?

Reply: We agree that there is a well-established link between IL-2 signaling and cell cycle progression. IL-2 directly influences the cell cycle by promoting degradation of the CDK inhibitor p27^{Kip1}, thereby increasing CDK2 activity and enabling progression through G1 phase. Interestingly, this relationship is bidirectional: while IL-2 regulates CDK expression and activity, CDKs in turn modulate IL-2 production (Huleatt et al., 2003; Appleman et al., 2000). Moreover, there are also links between IL-2 and metabolic priming.

In the discussion we have combined these topics as follows: IL-2 signaling is closely linked to cell cycle progression in CD8⁺ T cells. IL-2 promotes degradation of the CDK inhibitor p27^{Kip1}, leading to increased CDK2 activity and enabling progression through G1 phase. This relationship is bidirectional: IL-2 regulates CDK expression and activity, while CDKs in turn modulate IL-2 production. Beyond its role in cell cycle control, IL-2 also fuels metabolic programming, for example, by inducing upregulation of GLUT1 expression to support glucose uptake. This establishes a synergistic feedback loop whereby metabolically primed CD8⁺ T cells upregulate IL-2 receptor expression, thereby increasing their sensitivity to IL-2 signaling. Together, these mechanisms highlight the intricate crosstalk between IL-2 signaling, cell cycle machinery, and metabolic priming, which jointly may contribute to the enhanced proliferative capacity of CD8⁺ T cells following transient cell cycle arrest. However, the dual role of IL-2 in supporting both effector T cells and regulatory T cells (Tregs) poses a potential limitation, as IL-2-driven Treg expansion could suppress anti-tumor immunity. Therapeutic strategies that selectively bias IL-2 signaling toward effector T cells, such as IL-2 muteins or IL-2/antibody complexes, may help overcome this hurdle and improve clinical outcomes.

6. What is the specific role of cholesterol metabolism in cell cycle re-entry? Does the cell cycle directly regulate cholesterol?

Reply: Cholesterol metabolism plays a central role in proliferation, as cholesterol and lipids are required for membrane biogenesis, and energy homeostasis (Kuzu et al. 2016; Bietz et al. 2017). Moreover, certain signals for proliferation are transmitted through cholesterol-enriched lipid rafts. The cell cycle itself also regulates cholesterol metabolism by synchronizing enzyme activity with membrane expansion and by controlling cholesterol synthesis enzymes, highlighting a bidirectional relationship.

Consistent with this, our transcriptomic analysis showed that cholesterol metabolism is induced after stimulation combined with cell cycle blockade and further increased during release (Extended data Figure 3). We validated these findings by measuring FDFT1 expression *ex vivo* and *in vivo* (previously Extended data Figure 6c, 6d, and 6e, now revised as Figure 4c, 4d, 4e). Functionally, blockade of cholesterol synthesis by atorvastatin reduced proliferation in the non-arrested and released setting, supporting the importance of cholesterol metabolism for proliferation (previous version Extended Data Fig.6, revised version Fig. 4e). Moreover, we performed an additional experiment using zaragozic acid, a direct inhibitor of FDFT1, acting downstream of atorvastatin. The results were indicated that predominantly the non-arrested cells were affected, which may relate to the higher FDFT1 levels in the released cells (revised version Fig. 4e).

The text is as follows: FDFT1 expression was further increased in released CD8⁺ T cells and exceeded levels observed in non-arrested cells. Restricting FDFT1 using zaragozic acid impaired the proliferation of non-arrested cells but not of released cells, which may be related to their higher FDFT1 levels (**Fig. 4e**). Inhibition of the rate-limiting enzyme HMG-CoA reductase with atorvastatin suppressed proliferation in both non-arrested cells and released cells (**Fig. 4e**).

Given the central role of cholesterol metabolism, we have moved these data to the main Figures to emphasize their importance.

We have also high-lighted this in the discussion (page 19): Moreover, cholesterol biosynthesis, essential for proliferation through its role in membrane biogenesis and lipid raft-mediated signaling platforms, was elevated during cell cycle arrest and further increased upon release. Consistent with their higher FDFT1 expression, released cells showed greater resilience to functional inhibition, highlighting FDFT1-driven metabolic priming as a mechanism supporting robust proliferation after transient cell cycle blockade.

7. The immune profiling seems preliminary and limited – for instance, how are other cytokines affected?

Reply: We agree that the immune profiling could be expanded. In the current manuscript, we focused on IL-2 as it was prominently affected by cell cycle blockade at the transcriptional level (see Extended Data Fig. 3). We have performed new experiments to assess other key cytokines produced by effector CD8 T cells.

We have added the following text regarding the cytokine data (p14/15): The frequency of IFN- γ -producing CD8⁺ T cells also increased during arrest but remained lower than in non-arrested

or released CD8⁺ T cells (**Extended data Fig. 9c**). A substantial fraction of IL-2-producing CD8⁺ T cells co-expressed TNF (**Extended data Fig. 9d-e**), indicating enhanced cytokine polyfunctionality, which together with elevated autocrine IL-2 levels on a per-cell basis are hallmarks of memory T cells with superior expansion potential.

And we have further strengthened the IL-2 data by showing that the IL-2 production on a per-cell basis is enhanced in the arrested cells. Text page 14: Moreover, the IL-2 production on a per-cell basis was enhanced in arrested cells (**Fig. 6b**), which coincided with high amounts of IL-2 in the supernatant (**Fig. 6c, Extended Data Fig. 9b**).

In addition, we have expanded the immune profiling with respect to cell surface/intracellular molecules.

See page 6, ex vivo: Although downmodulation of CD62L and CD127 expression was modest, arrested CD8⁺ T cells upregulated the early activation marker CD69 (**Fig. 1g, Extended Data Fig. 1c-e**). Expression of EOMES was also induced in arrested CD8⁺ T cells, which further increased upon release, corresponding to the expression of granzyme B and CXCR3. The inhibitory receptors PD-1 and LAG-3 were moderately upregulated by arrested CD8⁺ T cells, with stronger expression observed in non-arrested and released cells (**Fig. 1g, Extended Data Fig. 1c-d**).

See page 16, in vivo: HU treatment induced increased expression of GLUT1, PKM, G6PD and CD98 in blood-circulating memory/effector CD8⁺ T cells of MC-38 tumor-bearing mice (**Fig. 7a-c**). Phenotypically, these cells exhibited high KLRG1 and CD43^{1BB} expression (**Extended Data Fig. 10a-b**).

8. Are the programs observed in arrested cells comparable to memory immune cells that have been re-activated? This seems like the more reasonable comparison and should also be experimentally tested and discussed.

Reply: We have addressed this interesting question by comparing our transcriptomic data set of the arrested/released cells with a transcriptomic data set of re-activated memory immune cells (Extended Data Fig.5). The comparisons suggest that arrested CD8⁺ T cells mirror the transcriptional profile of resting memory T cells, whereas released CD8⁺ T cells display gene expression patterns characteristic of reactivated memory T cells.

The text added to the revised manuscript is as follows.

To assess whether the transcriptional programs of the arrested and released CD8⁺ T cells resemble those of resting and reactivated memory CD8⁺ T cells, we compared our mRNA-sequencing dataset to a recently published dataset profiling lymph node and tissue-resident memory CD8⁺ T cells in both resting and reactivated states. Using the same EdgeR pipeline (FDR < 0.05), we identified 1,519 overlapping differentially expressed genes (**Extended Data Fig. 5**). While certain genes were similarly upregulated (such as *Ii2*) or downregulated in both arrested and reactivated cells, many changes, including those in glycolytic and effector genes (*Pkm*, *Aldoa*, *Pgam1*, *Gapdh*, *Eno1*, *Gzmb*), were shared between the released and reactivated conditions. These findings indicate that arrested CD8⁺ T cells largely mirror the transcriptional profile of resting memory T cells, whereas released CD8⁺ T cells acquire gene expression patterns characteristic of reactivated memory T cells, suggesting that memory-like features including re-activation properties are already imprinted during the arrested state.

Response to reviewers; point-by-point reply

Reviewer #1

The authors have not addressed the first two concerns I have with the study.

On point 1: hydroxyurea is an anti-cancer agent that functions by inhibiting ribonucleotide reductase, thereby reducing dNTPs, promoting replication stress and replication stress-induced DNA damage, in addition to oxidative damage, that is exacerbated by prolonged treatment (e.g. as reviewed in Musialek 2021). Therefore, a significant amount of cellular stress would be expected in the 60-hour treatment time in the experiments shown. Perhaps the mechanisms of action of hydroxyurea are different in CD8 T cells. But this is a controversial claim that requires substantial evidence. Y15 phosphorylation on CDK1/2 is a poor choice to discriminate DNA damage from cell cycle control by phosphoproteomics. T14 and Y15 are difficult to discriminate by phosphoproteomics and indeed, phosphorylation on both T14 and Y15 can occur, which would reduce the intensity for the monophosphorylated Y15 peptide shown in the extended data. Biologically, this means Y15 can still be phosphorylated, even if there is a decrease in the Y15 monophosphorylated form, because the chemical species has shifted to the doubly phosphorylated form. These differences could also be due to a difference in the cell cycle position of cells. For example, if “Released” cells were mostly in G2 phase (following multiple rounds of division), then these cells would be expected to have lower Y15 phosphorylation compared to “Non-arrested”, and therefore not a reporter of an absence of DNA damage.

A direct measure of DNA damage is therefore crucial.

Reply: We thank the reviewer for emphasizing the importance of directly assessing DNA damage. In response, we have performed experiments to quantify double-strand DNA breaks (DSBs) by flow cytometric detection of γ -H2AX (phosphorylated Histone H2A at serine 139, pSer139), a well-established marker of DNA damage (Hoare et al., J Hepatol. 2013). This approach enables sensitive, single-cell analysis of DSBs. These measurements were performed with human T cells treated with or without cell cycle inhibitors (HU, palbociclib and ribociclib) under unstimulated, arrested, non-arrested, and released conditions.

The results are summarized in the manuscript as follows: “To evaluate directly whether transient cell cycle inhibition induces DNA damage, we assessed γ -H2AX expression, a marker of DNA double-strand breaks (DSBs), in CD8⁺ T cells (Hoare 2013). HU-arrested cells showed elevated γ -H2AX at 60 h, consistent with replication stress-associated DNA damage caused by stalled replication forks (Musialek 2021), an effect that is exacerbated by prolonged HU exposure. Upon release from HU, γ -H2AX levels were reduced, likely reflecting DNA repair. In contrast, treatment with palbociclib or ribociclib did not induce detectable DNA damage, and released cells exhibited only a modest increase in γ -H2AX expression, (**Extended Data Fig. 8a-b**). Non-arrested cells displayed low γ -H2AX expression at 24 h, which increased after 60 h of stimulation, consistent with replication-associated stress during continuous proliferation. Together, these data show that transient cell cycle inhibition and subsequent release do not cause substantial or lasting DNA damage.”

We have removed the mention of CDK2_Y15 phosphorylation from the main text to avoid any misinterpretation. We emphasize that we did not use this site as a readout of DNA damage, and our conclusions are based on the absence of coordinated DDR signatures across canonical phosphosites (Extended Data Fig. 7a).

On point 2: Data on whether cells ‘blast’ (such as scatter measurements by flow cytometry) should be shown for all the cell cycle inhibitors shown in Figure 1. The authors’ reply only indicate data for hydroxyurea.

Reply: We have added scatter measurements for the other cell cycle inhibitors, and below the results are shown. We have include these results in the manuscript (Extended Data Figure 1h).

On another point, in response to reviewer 2, the authors cite papers that do not appear to support their claims, e.g.

“while IL-2 regulates CDK expression and activity, CDKs in turn modulate IL-2 production (Huleatt et al., 2003; Appleman et al., 2000)”

The Huleatt paper shows that p27 is crucial for arrest following IL-2 starvation.

The Appleman paper shows that IL-2 is crucial for activation of D-CDK4/6 and E-CDK2 complexes and downregulation of p27.

These data in these studies do not support the idea that CDKs modulate IL-2 production. What they do show is that IL-2 withdrawal or CD28 deletion impacts cell cycle progression by reduced CDK activity. Not the other way around.

Reply: We thank the reviewer for this helpful clarification. We agree that the cited papers (Huleatt et al., 2003; Appleman et al., 2000) primarily demonstrate that IL-2 regulates CDK expression and activity, rather than the reverse. To support the statement that CDKs can, in turn, modulate IL-2 production and signaling, we will add relevant references showing that CDK activity contributes to regulation of IL-2 production and signaling (e.g., Boussiotis et al., 2000; Chunder et al., 2012; Mohapatra et al., 2001).

-p27kip1 functions as an energy factor inhibiting interleukin 2 transcription and clonal expansion of alloreactive human and mouse helper T lymphocytes.

Boussiotis VA, Freeman GJ, Taylor PA, Berezovskaya A, Grass I, Blazar BR, Nadler LM. Nat Med. 2000;6(3):290-7. doi: 10.1038/73144.

-Cyclin-dependent kinase 2 controls peripheral immune tolerance.

Chunder N, Wang L, Chen C, Hancock WW, Wells AD. J Immunol. 2012 Dec 15;189(12):5659-66. doi: 10.4049/jimmunol.1202313. Epub 2012 Nov 7.

-Interdependence of cdk2 activation and interleukin-2Ralpha accumulation in T cells.

Mohapatra S, Pledger WJ. J Biol Chem. 2001;276(24):21984-9. doi: 10.1074/jbc.M100037200.

Reviewer #3

This manuscript provides a thorough mechanistic understanding of the “bounce-back” phenomenon whereby activated CD8 T cells that are paused in cell cycle are able to recover faster than their un-paused counterparts. The clinical relevance of this finding cannot be overstated, given the extensive use of CDK4/6 inhibitors and other cell cycle inhibitors for cancer therapy. It is extremely reassuring to know that T cells are able to recover quickly from intermittent periods of proliferative arrest, and that they spend the arrested time building up glycogen and other metabolic stores needed for rapid growth.

Minor comments:

To my knowledge, the first report of the bounce-back phenomenon and increased IL-2 production based on cell cycle state was 20 years ago in JI (PMID: 15778358). Obviously, the current authors have greatly expanded on this early finding, but the original paper should be cited.

Reply: We have cited the paper (Munitic et al., J Immunol. 2005) in the discussion. The authors showed that discontinued stimulated CD4⁺ mouse T cells results in similar proliferation as compared to continuous stimulation, involving rapid production of IL-2 by responsive G1 T cells upon restimulation.

Figure 7i is lacking a color key for the immunofluorescence image.

Reply: We have added a color key for Figure 7i.